# KaSA: Knowledge-Aware Singular-Value Adaptation of Large Language Models

**Fan Wang**[*♡], **Juyong Jiang**[*♡], **Chansung Park**[*♠], **Sunghun Kim**[†♡♣], **Jing Tang**[†♡♣]

[♡]The Hong Kong University of Science and Technology (Guangzhou)
[♠]Electronics and Telecommunications Research Institute
[♣]The Hong Kong University of Science and Technology
{csfanwang,csjuyongjiang,deep.diver.csp}@gmail.com
{hunkim,jingtang}@ust.hk

## Abstract

The increasing sizes of large language models (LLMs) result in significant computational overhead and memory usage when adapting these models to specific tasks or domains. Various parameter-efficient fine-tuning (PEFT) methods have been devised to mitigate these challenges by training a small set of parameters for the task-specific updates of the model weights. Among PEFT methods, LoRA stands out for its simplicity and efficiency, inspiring the development of a series of variants. However, LoRA and its successors disregard the knowledge that is noisy or irrelevant to the targeted task, detrimentally impacting model performance and leading to suboptimality. To address this limitation, we introduce Knowledge-aware Singular-value Adaptation (KaSA), a PEFT method that leverages singular value decomposition (SVD) with knowledge-aware singular values to dynamically activate knowledge based on its relevance to the task at hand. We conduct extensive experiments across a range of LLMs on tasks spanning natural language understanding (NLU), generation (NLG), instruction following, and commonsense reasoning. The experimental results demonstrate that KaSA consistently outperforms FFT and 14 popular PEFT baselines across 16 benchmarks and 4 synthetic datasets, underscoring our method's efficacy and adaptability. The source code of our method is available at https://github.com/juyongjiang/KaSA.

## 1 Introduction

Large language models (LLMs) pretrained on massive general domain data have shown remarkable generalization ability, facilitating their application across diverse tasks (Zhao et al., 2023; Touvron et al., 2023b; OpenAI, 2023; Yoo et al., 2024; Jiang et al., 2024). The adaptation of these pretrained language models (PLMs) to specific downstream tasks generally involves full fine-tuning (FFT), where all model parameters are updated and distinct replicas of the parameters are saved for each task (Guo et al., 2021; Mao et al., 2022; Gao et al., 2024). However, the increasing size of LLMs significantly raises the computational and memory costs associated with FFT, making FFT impractical in resource-constrained environments (Lester et al., 2021; Cai et al., 2024; Meng et al., 2024). Consequently, a surge of parameter-efficient fine-tuning (PEFT) methods (Zaken et al., 2021; Li & Liang, 2021; Hu et al., 2021; Liu et al., 2023; Pfeiffer et al., 2021; Houlsby et al., 2019; Liu et al., 2024) have emerged, aiming to reduce the computational and memory costs by only updating a small set of parameters while fixing the base model (Mao et al., 2022; Lialin et al., 2023).

Notably, LoRA (Hu et al., 2021) is popular for its simplicity and effectiveness (Wang et al., 2024a; Liu et al., 2024; Gao et al., 2024). It reparameterizes the task-specific update $\Delta \mathbf{W} \in \mathbb{R}^{n \times m}$ with a couple of low-rank matrices, $\mathbf{A}$ and $\mathbf{B}$, while keeping the base model $\mathbf{W}^{(0)} \in \mathbb{R}^{n \times m}$ unchanged during fine-tuning. Without loss of generality, we suppose $n \geq m$ to simplify the notation. The fine-tuning process of LoRA can be formally expressed as $\mathbf{W}^{(0)} + \Delta \mathbf{W} = \mathbf{W}^{(0)} + \frac{\alpha}{r}\mathbf{B}\mathbf{A}$, where

---

[*]Equal contributors: Fan Wang, Juyong Jiang, and Chansung Park.
[†]Corresponding authors: Sunghun Kim and Jing Tang.

$\mathbf{B} \in \mathbb{R}^{n \times r}$, $\mathbf{A} \in \mathbb{R}^{r \times m}$, $\alpha$ is a scaling constant, and the rank $r \ll m$. A significant advantage of LoRA is its practicality in integrating the low-rank matrices back into the base model, thereby preserving the model architecture and avoiding additional inference latency (Hu et al., 2021; Han et al., 2024; Meng et al., 2024).

Despite LoRA's success, its initialization strategy, which employs random Gaussian noise for $\mathbf{A}$ and zeros for $\mathbf{B}$, creates an unguided subspace for the trainable parameters, causing slow convergence and suboptimal performance (Meng et al., 2024; Wang et al., 2024a). To address this problem, PiSSA (Meng et al., 2024) and MiLoRA (Wang et al., 2024a) use singular value decomposition (SVD) for optimizing initialization. SVD can factorize a matrix into three distinct matrices ($\mathbf{U}$, $\boldsymbol{\Sigma}$, $\mathbf{V}$), where $\mathbf{U}$ and $\mathbf{V}$ are semi-orthogonal matrices, and $\boldsymbol{\Sigma}$ is a diagonal matrix containing singular values sorted in descending order. In particular, the magnitude of singular values represents the importance of parametric knowledge encapsulated in their corresponding singular vectors, with large values indicating important world knowledge and small values indicating noisy or long-tail knowledge (Yan et al., 2021; Wang et al., 2024a; Yang et al., 2023; Sharma et al., 2023). PiSSA and MiLoRA apply SVD to decompose the base model into two components: the principal components correlated with major singular values, and the residual components associated with minor singular values. Specifically, PiSSA fine-tunes the low-rank matrices, $\mathbf{B}$ and $\mathbf{A}$, initialized with principal components, while preserving the residual components frozen, resulting in faster convergence and improved model performance (Meng et al., 2024). In contrast, MiLoRA focuses on fine-tuning $\mathbf{B}$ and $\mathbf{A}$ initialized with the minor singular value components, while fixing the principal components, aiming to boost performance and alleviate world knowledge forgetting (Wang et al., 2024a).

However, PiSSA and MiLoRA disregard two issues that can detrimentally affect model performance. Firstly, a portion of the task-specific updates targets the weight changes of the noisy knowledge encoded in the base model, potentially leading to suboptimal performance. Secondly, the low-rank matrices, whether initialized with the principal or residual components, inherit knowledge from the base model. These components may include information that is irrelevant to the specific downstream task, leading to conflicts within the parametric knowledge and degrading the model's representational capability.

To address these problems, we propose a PEFT method, named *KaSA* (Knowledge-aware Singular-value Adaptation), which leverages SVD with knowledge-aware singular values to dynamically activate parametric knowledge according to its relevance to downstream tasks. Specifically, KaSA begins by performing knowledge-based SVD truncation to the base model $\mathbf{W}^{(0)}$ for removing the minor singular components $\mathbf{W}_{noise} \in \mathbb{R}^{n \times m}$ that contain noisy and long-tail knowledge (Gu et al., 2024; Wang et al., 2024b; Meng et al., 2024). This process results in an SVD-truncated model $\mathbf{W}_{world} \in \mathbb{R}^{n \times m}$ that retains essential world knowledge. To maintain a consistent representational space between $\mathbf{W}_{world}$ and its task-specific updates $\Delta \mathbf{W}$, KaSA reparameterizes $\Delta \mathbf{W}$ in the SVD form, $\Delta \mathbf{W} = \Delta \mathbf{U} \Delta \boldsymbol{\Sigma} \Delta \mathbf{V}^{\top}$, where $\Delta \boldsymbol{\Sigma}$ comprises knowledge-aware singular values $(\Delta \sigma_1, ..., \Delta \sigma_r)$. The singular-value adaptation presents dual benefits: 1) reparameterizing the task-specific updates in SVD form ensures that these updates and $\mathbf{W}_{world}$ share the same representational space, thereby preserving knowledge consistency; 2) the knowledge-aware singular values learn to activate the parametric knowledge according to its pertinence to particular downstream tasks, reducing the intervention of irrelevant knowledge, consequently enhancing model performance.

We conduct extensive experiments to fine-tune LLMs of varying sizes and architectures across various tasks, including natural language understanding (NLU), natural language generation (NLG), instruction following, and commonsense reasoning. Substantial experimental results demonstrate that KaSA consistently outperforms FFT and 14 existing popular PEFT baselines across different LLMs on 16 benchmarks and 4 synthetic datasets, highlighting its efficacy and adaptability. In summary, the key contributions in our work are as follows:

- We propose a novel PEFT method, KaSA, which leverages SVD with knowledge-aware singular values to activate parametric knowledge based on its relevance to downstream tasks, achieving superior performance over FFT and existing prevailing PEFT techniques across various tasks.

- KaSA features a linear framework allowing seamless integration of the singular value adaptation module with the SVD truncated model, inducing no inference latency. Furthermore,

our method supports training distinct adaptation modules for different tasks, all sharing a single base model, thereby reducing the storage needs for task-switching.

- We conduct extensive experiments on NLU, NLG, instruction following, and common-sense reasoning tasks using various LLMs on well-known benchmarks. Our KaSA consistently outperforms FFT and 14 PEFT baselines across different benchmarks and synthetic datasets, demonstrating its efficacy and adaptability.

- We make all high-quality synthetic instruction-following datasets generated by GPT4o publicly available [1], enabling the community to enhance the functionality of PEFT and support future research endeavors.

## 2    RELATED WORK

### 2.1    PARAMETER-EFFICIENT FINE-TUNING

The increasing LLM scale presents significant challenges to efficiently adapting these models to specific tasks (Lialin et al., 2023; Zhao et al., 2023). In response, a surge of PEFT methods has emerged, reducing the computation burden by updating a minimal set of parameters during fine-tuning (Mao et al., 2022; Karimi Mahabadi et al., 2021; Han et al., 2024). PEFT methods can be generally categorized into selective, additive, and re-parameterized methods (Ding et al., 2022; Lialin et al., 2023; Xu et al., 2023). Selective methods (Zaken et al., 2021; Sung et al., 2021; Guo et al., 2021; He et al., 2023) train a predetermined set of the model's existing parameters while keeping the rest of the model intact. Additive methods (Houlsby et al., 2019; He et al., 2022a; Li & Liang, 2021; Liu et al., 2023; Lester et al., 2021) introduce extra modules or parameters to fine-tune and maintain the original base model frozen. Reparametrized methods (Hu et al., 2021; Dettmers et al., 2023; Zhang et al., 2022; Valipour et al., 2023; Liu et al., 2024) reparameterize the model's weight updates into an equivalent low-rank form for fine-tuning. Among reparameterized approaches, LoRA stands out for its simple yet efficient mechanism of employing two low-rank matrices to approximate task-specific updates. The fine-tuned LoRA matrices can be integrated with the base model, ensuring no inference latency. LoRA has inspired a series of variants, each targeting specific improvements. For instance, DyLoRA (Valipour et al., 2023) trains the low-rank matrices across a spectrum of ranks by sorting the representation learned at different ranks during training, shortening the training time. QLoRA (Dettmers et al., 2023) combines 4-bit quantization with LoRA for enhanced resource efficiency. DoRA (Liu et al., 2024) decomposes the base model into magnitude and direction components for fine-tuning, reducing the number of trainable parameters and improving performance over LoRA. Our method, KaSA, diverges from these reparametrized methods by employing a knowledge-aware SVD structure, enhancing the fine-tuning efficacy further.

### 2.2    SINGULAR VALUE DECOMPOSITION IN NATURAL LANGUAGE PROCESSING

SVD plays a crucial role in various domains, such as model compression (Yuan et al., 2023; Wang et al., 2024b; Hsu et al., 2021; Chen et al., 2021), dimensionality reduction of word embeddings (Tanwar et al., 2018; Shyamasundar & Rani, 2016), and latent semantic structure analysis (Deerwester et al., 1990; Kou & Peng, 2015; Horasan et al., 2019). In the rapidly growing realm of LLMs, SVD emerges as a promising, yet relatively underexplored, technique for PEFT. A series of SVD-based PEFT methods exploit the relationship between SVD and matrix rank to ascertain optimal ranks for specific downstream tasks. For example, AdaLoRA (Zhang et al., 2022) employs SVD to reparameterize task-specific updates and adaptively determines the suitable rank through importance scoring, thus improving the model performance and parameter efficiency. SARA (Gu et al., 2024) conducts SVD at the initialization phase to identify the appropriate rank for each layer, thereby maintaining the benefits of LoRA and boosting performance. PiSSA (Meng et al., 2024) and MiLoRA (Wang et al., 2024a), as mentioned in Section 1, utilize SVD to optimize LoRA's initialization. Specifically, PiSSA (Meng et al., 2024) only fine-tunes the low-rank matrices initialized with the principal components associated with a few largest singular values, while preserving the residual frozen. This initialization strategy facilitates faster convergence and enhanced performance. Conversely, MiLoRA (Wang et al., 2024a) fine-tunes the minor components associated with minimal singular values, enhancing model performance while preserving the model's world knowledge.

---

[1] https://huggingface.co/llama-duo

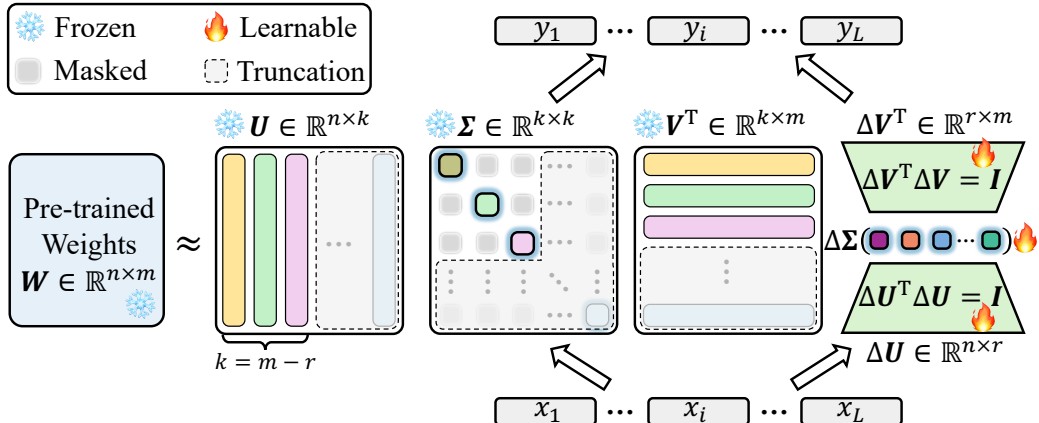

Figure 1: The architecture of our proposed KaSA encompasses two stages: (**Left**) knowledge-based SVD truncation to remove the noisy knowledge from the base model; (**Right**) knowledge-aware singular-value adaptation to adjust singular values that dynamically activate knowledge across $\Delta \mathbf{W}$ model parameters based on its relevance to downstream tasks.

Unlike these methods, our method emphasizes the adaptive adjustment of singular values, allowing nuanced and dynamic activation of parametric knowledge based on its importance to downstream tasks.

## 3 METHODOLOGY

In this section, we commence with modeling the general PEFT process and training objective in Section 3.1. We subsequently provide a detailed introduction of KaSA in Section 3.2, followed by the description of its training objective in Section 3.3.

### 3.1 PROBLEM STATEMENT

Before introducing KaSA, it is essential to delineate and model the process and objective of PEFT for LLMs based on the Transformer architecture (Vaswani, 2017). Fundamentally, PEFT is the process of training a pretrained model to a targeted task using a task-specific dataset. It aims to minimize the divergence between the predicted probability distribution of the fine-tuned model and the actual distribution of the training data, while only modifying a small set of parameters.

Consider a pretrained model $\mathbf{W}^{(0)}$, initially parameterized by $\Theta_0$. To adapt this model to a particular task, we employ PEFT using a dataset $D = \{(x_l, y_l)\}_{l=1}^{Q}$ comprising $Q$ input-output instances. The PEFT process utilizes a limited set of parameters, denoted as $\boldsymbol{\Psi}$, to learn the task-specific update $\triangle\Theta$, ensuring that $|\boldsymbol{\Psi}| \ll |\Theta_0|$. This results in a fine-tuned model $\mathbf{W}$, parameterized by $\Theta_0 + \triangle\Theta(\boldsymbol{\Psi})$. The objective is to align the predicted probability distribution of $\mathbf{W}$ with the actual distribution of training data, thereby enhancing the fine-tuned model's task performance. The primary objective of PEFT is thus centered on the optimization of $\boldsymbol{\Psi}$:

$$\mathcal{L}_1(\boldsymbol{\Psi}) = \sum_{(x,y)\in D} \sum_{t=1}^{|y|} -\log(P_{\Theta_0+\triangle\Theta(\boldsymbol{\Psi})}(y_t|x, y_{<t})) \tag{1}$$

### 3.2 KNOWLEDGE-AWARE SINGULAR-VALUE ADAPTATION

As depicted in Fig.1, KaSA encompasses two primary stages: 1) the knowledge-based SVD truncation, which removes the noisy knowledge from the base model; and 2) knowledge-aware singular-value adaptation, which involves adjustment of singular values that dynamically activate parametric knowledge based on its relevance to the targeted task.

KaSA begins with a knowledge-based SVD truncation to the base model $\mathbf{W}^{(0)} \in \mathbb{R}^{n\times m}$. For simplicity of denotation, we suppose $n \geq m$. This process factories $\mathbf{W}^{(0)}$ using SVD and subsequently

truncates the minor singular components $\mathbf{W}_{noise} \in \mathbb{R}^{n \times m}$, removing noisy and long-tail knowledge and resulting in a lower-rank model $\mathbf{W}_{world} \in \mathbb{R}^{n \times m}$. We use this refined model $\mathbf{W}_{world}$ to approximate the base model, making the adaptation of $\mathbf{W}^{(0)}$ to be resembled by that of $\mathbf{W}_{world}$:

$$\mathbf{W} = \mathbf{W}^{(0)} + \Delta\mathbf{W} = \mathbf{U}\mathbf{\Sigma}\mathbf{V}^\top + \Delta(\mathbf{U}\mathbf{\Sigma}\mathbf{V}^\top) = \sum_{i=1}^{m} u_i\sigma_i v_i^\top + \sum_{i=1}^{m} \Delta(u_i\sigma_i v_i^\top) \tag{2}$$

$$= (\mathbf{W}_{world} + \mathbf{W}_{noise}) + (\Delta\mathbf{W}_{world} + \Delta\mathbf{W}_{noise}) \tag{3}$$

$$= (\sum_{i=1}^{m-r} u_i\sigma_i v_i^\top + \sum_{i=1}^{r} u_i\sigma_i v_i^\top) + (\sum_{i=1}^{m-r} \Delta(u_i\sigma_i v_i^\top) + \sum_{i=1}^{r} \Delta(u_i\sigma_i v_i^\top)) \tag{4}$$

$$\approx \mathbf{W}_{world} + \Delta\mathbf{W}_{world} = \sum_{i=1}^{m-r} u_i\sigma_i v_i^\top + \sum_{i=1}^{m-r} \Delta(u_i\sigma_i v_i^\top) \tag{5}$$

where $\mathbf{U} \in \mathbb{R}^{n \times m}$, $\mathbf{V} \in \mathbb{R}^{m \times m}$, and $\mathbf{V}^\top$ is the transpose of $\mathbf{V}$. $\mathbf{U} = [u_1, ..., u_m]$ and $\mathbf{V} = [v_1, ..., v_m]$ are the corresponding matrices containing left and right singular vectors, respectively. The diagonal matrix $\mathbf{\Sigma} \in \mathbb{R}^{m \times m}$ contains positive singular values $(\sigma_1, ..., \sigma_m)$ sorted from high to low $(\sigma_1 \geq \sigma_2 \geq \cdots \geq \sigma_m \geq 0)$. The hyperparameter $r$ represents the number of truncated minor singular values, with $r \ll m$. $\mathbf{U}$ and $\mathbf{V}$ are semi-orthogonal, satisfying that:

$$\mathbf{U}^\top\mathbf{U} = \mathbf{V}^\top\mathbf{V} = \mathbf{I}_m \tag{6}$$

where the identity matrix $\mathbf{I}_m \in \mathbb{R}^{m \times m}$. Following the knowledge-based SVD truncation, we employ the knowledge-aware singular-value adaptation, which reparameterizes the task-specific updates of $\mathbf{W}_{world}$ in the SVD form with knowledge-aware singular values. Therefore, the weight of a model fine-tuned with KaSA can be formally expressed as:

$$\mathbf{W} = \mathbf{W}^{(0)} + \Delta\mathbf{W} \approx \mathbf{W}_{world} + \eta\Delta\mathbf{U}\Delta\mathbf{\Sigma}\Delta\mathbf{V}^\top = \sum_{i=1}^{m-r} u_i(\sigma_i)v_i^\top + \eta\sum_{j=1}^{r} \Delta u_j(\Delta\sigma_j)\Delta v_j^\top$$

$$\text{s.t.} \quad \Delta\mathbf{U}^\top\Delta\mathbf{U} = \Delta\mathbf{V}^\top\Delta\mathbf{V} = \mathbf{I}_r \tag{7}$$

where $\mathbf{I}_r \in \mathbb{R}^{r \times r}$, $\eta > 0$ is a constant scaler, the diagonal matrix $\Delta\mathbf{\Sigma} \in \mathbb{R}^{r \times r}$ comprising learnable knowledge-aware singular values $(\Delta\sigma_1, ..., \Delta\sigma_r)$. The matrices $\Delta\mathbf{U}$ and $\Delta\mathbf{V}$ are semi-orthogonal, ensuring that the updates retain necessary structural properties.

## 3.3 TRAINING OBJECTIVE

FFT typically serves as a comparative performance upper bound for PEFT methods (Valipour et al., 2023). Consequently, we expect that the performance of the model fine-tuned with KaSA will closely approximate that of FFT. We denote the FFT model as $\mathbf{W}_{fft} = \mathbf{W}^{(0)} + \Delta\mathbf{W}$. We impose a regularization $\|\mathbf{W}_{fft} - \mathbf{W}_{world}\|_F$, represented by the Frobenius norm, to constrain the task-specific updates. Based on the properties of Frobenius norms, we can further explore the boundary of the task-specific updates:

$$\|\mathbf{W}_{fft}\|_F + \|\mathbf{W}_{world}\|_F \geq \|\mathbf{W}_{fft} - \mathbf{W}_{world}\|_F \geq \|\Delta\mathbf{U}\Delta\mathbf{\Sigma}\Delta\mathbf{V}^\top\|_F = \|\sum_{j=1}^{r} \Delta u_j(\Delta\sigma_j)\Delta v_j^\top\|_F \tag{8}$$

To stabilize the model training and extend the searching space, we introduce $\mathcal{L}_2$ to minimize the lower boundary of $\|\mathbf{W}_{fft} - \mathbf{W}_{world}\|_F$:

$$\mathcal{L}_2(\Delta\mathbf{\Sigma}) = \|\Delta\mathbf{U}\Delta\mathbf{\Sigma}\Delta\mathbf{V}^\top\|_F^2 \tag{9}$$

According to the Eckart–Young–Mirsky theorem (Eckart & Young, 1936), $\mathcal{L}_2$ is reformulated as:

$$\mathcal{L}_2(\Delta\mathbf{\Sigma}) = \|\Delta\mathbf{U}\Delta\mathbf{\Sigma}\Delta\mathbf{V}^\top\|_F^2 = \|\sum_{j=1}^{r} \Delta u_j(\Delta\sigma_j)\Delta v_j^\top\|_F^2 = \sum_{j=1}^{r} (\Delta\sigma_j)^2 \tag{10}$$

Our method proposes knowledge-aware singular-value adaptation, which reparameterizes the task-specific update in the SVD form and guides $\Delta \mathbf{U}$ and $\Delta \mathbf{V}$ to conform to orthogonality. Given this, we introduce $\mathcal{L}_3$ to constrain $\Delta \mathbf{U}$ and $\Delta \mathbf{V}$ adhere to orthogonality, such that:

$$\mathcal{L}_3(\boldsymbol{\Psi}) = \left\| \Delta \mathbf{U}^\top \Delta \mathbf{U} - \mathbf{I}_r \right\|_F + \left\| \Delta \mathbf{V}^\top \Delta \mathbf{V} - \mathbf{I}_r \right\|_F \tag{11}$$

Overall, our methods leverage $\mathcal{L}_1$, $\mathcal{L}_2$, and $\mathcal{L}_3$ to serve jointly for optimizing the model's task performance while adhering to SVD structure. For adjusting $\mathcal{L}_2$ and $\mathcal{L}_3$, we introduce $\beta > 0$ and $\gamma > 0$ as their corresponding scalers. The overall training objective of KaSA can be expressed as:

$$\mathcal{L}(\boldsymbol{\Psi}, \Delta\boldsymbol{\Sigma}) = \min_{\boldsymbol{\Psi}, \Delta\boldsymbol{\Sigma}} (\mathcal{L}_1(\boldsymbol{\Psi}, \Delta\boldsymbol{\Sigma}) + \beta \mathcal{L}_2(\Delta\boldsymbol{\Sigma}) + \gamma \mathcal{L}_3(\boldsymbol{\Psi})) \tag{12}$$

We present the PyTorch-style pseudocode for KaSA and its training objective in Appendix A.

## 4 EXPERIMENTS

In this section, we evaluate KaSA's effectiveness across different downstream tasks, including natural language understanding (NLU), natural language generation (NLG) (see Appendix F.2), instruction following, and commonsense reasoning. For NLU tasks, we evaluate KaSA with RoBERTa (Liu et al., 2021) and DeBERTaV3 (He et al., 2022b) on the GLUE (Wang et al., 2018) benchmark. For NLG tasks, we assess our method with GPT-2 (Radford et al., 2019) on the E2E NLG Challenge (Novikova et al., 2017) benchmark. We further assess the instruction following performance using well-known LLMs, including LLaMA3 8B (Meta, 2024), Mistal 7B (Jiang et al., 2023), Gemma 7B (Gemma Team, 2024), and LLaMA2 13B (Touvron et al., 2023b). These models are fine-tuned with different PEFT methods using four synthetic datasets generated by GPT4o, each tailored to summarization, classification, coding, and closed QA. GPT4o is then employed as a judge to evaluate the fine-tuned models' performance, assigning scores on a scale of 10. We also follow (Kopiczko et al., 2023) and (Gao et al., 2024) to fine-tune the four models on the Alpaca dataset (Taori et al., 2023b) and report evaluation results on MT-Bench, with GPT4 serving as the judge, yielding scores within 10. Additionally, we substantiate KaSA's generality by fine-tuning LLaMA2 7B and LLaMA3 8B models on the Commonsense170K dataset (Hu et al., 2023), which includes training sets from eight commonsense reasoning datasets, and evaluating them on individual test sets of these constituent datasets. Finally, we conduct ablation studies to investigate the impacts of different components, budget parameter scalability, and the distribution of knowledge-aware singular values across various layers. All experiments are conducted on NVIDIA A100-SXM4 (80GB) GPUs, except for the NLU experiments, which are conducted on NVIDIA GeForce RTX 3090 (24GB) GPUs.

### 4.1 BASELINES

We compare KaSA with FFT and 14 PEFT baselines to substantiate its efficiency and robustness:
• **Adapter-based methods** We consider four representative Adapter tuning methods as baselines: 1) Adapter$^H$ (Houlsby et al., 2019); 2) Adapter$^D$ (Rücklé et al., 2021); 3) Adapter$^L$ (Lin et al., 2020); and 4) Adapter$^P$ (Pfeiffer et al., 2021).
• **LoRA-based methods** We select LoRA and its variants: 1) LoRA (Hu et al., 2021); 2) DyLoRA (Valipour et al., 2023); 3) VeRA (Kopiczko et al., 2023); and 4) DoRA (Liu et al., 2024).
• **SVD-based methods** Considering that our method is associated with SVD, we chose SVD-based PEFT baselines: 1) AdaLoRA (Zhang et al., 2022); 2) PiSSA (Meng et al., 2024); 3) MiLoRA (Wang et al., 2024a); 4) SARA (Gu et al., 2024); and 5) CorDA (Yang et al., 2024).
• **Other methods** Apart from the aforementioned baselines, we also consider other important fine-tuning methods: 1) FFT; and 2) BitFit (Zaken et al., 2021).
To ensure a fair comparison with these baselines, we meticulously replicate the experimental configurations as described in previous studies (Hu et al., 2021; Zhang et al., 2022; Gu et al., 2024). Introductions of the baselines and comprehensive details of the experimental setup are provided in Appendix B and Appendix E, respectively.

### 4.2 NATURAL LANGUAGE UNDERSTANDING

**Models and Datasets.** For NLU tasks, our method involves fine-tuning foundation models such as RoBERTa-base (125M), RoBERTa-large (355M) (Liu et al., 2021), and DeBERTaV3-base (He

Table 1: Performance of RoBERTa-base (RoB$_{base}$) and RoBERTa-large (RoB$_{large}$) with different adaptation methods on 6 datasets of the GLUE benchmark. We report the overall (matched and mismatched) accuracy for MNLI, Matthew's correlation coefficient (Mcc.) for CoLA, Pearson correlation coefficient (Pcc.) for STS-B, and accuracy (Acc.) for all the remaining tasks. We report the average result of five runs with different random seeds. The best results for each dataset are shown in **bold**. Higher is better for all metrics.

| Model(Method) | # Trainable Parameters | SST-2 (Acc.) | MRPC (Acc.) | CoLA (Mcc.) | QNLI (Acc.) | RTE (Acc.) | STS-B (Pcc.) | All Avg. |
|---|---|---|---|---|---|---|---|---|
| RoB$_{base}$(FFT) | 125.0M | 94.8 | 90.2 | 63.6 | 92.8 | 78.7 | 91.2 | 85.2 |
| RoB$_{base}$(BitFit) | 0.1M | 93.7 | **92.7** | 62.0 | 91.8 | 81.5 | 90.8 | 85.4 |
| RoB$_{base}$(Adpt$^D$) | 0.3M | 94.2 | 88.5 | 60.8 | 93.1 | 71.5 | 89.7 | 83.0 |
| RoB$_{base}$(Adpt$^D$) | 0.9M | 94.7 | 88.4 | 62.6 | 93.0 | 75.9 | 90.3 | 84.2 |
| RoB$_{base}$(LoRA) | 0.3M | 95.1 | 89.7 | 63.4 | **93.3** | 78.4 | **91.5** | 85.2 |
| RoB$_{base}$(AdaLoRA) | 0.3M | 94.5 | 88.7 | 62.0 | 93.1 | 81.0 | 90.5 | 85.0 |
| RoB$_{base}$(DyLoRA) | 0.3M | 94.3 | 89.5 | 61.1 | 92.2 | 78.7 | 91.1 | 84.5 |
| RoB$_{base}$(PiSSA) | 0.3M | 95.0 | 88.2 | 65.5 | 92.0 | 75.1 | 90.4 | 84.4 |
| RoB$_{base}$(MiLoRA) | 0.3M | 94.6 | 88.7 | 63.1 | 92.8 | 80.5 | 91.3 | 85.2 |
| RoB$_{base}$(KaSA) | 0.3M | **95.2** | 90.7 | **65.8** | 93.3 | **81.6** | 91.1 | **86.3** |
| RoB$_{large}$(FFT) | 355.0M | 96.4 | 90.9 | 68.0 | 94.7 | 86.6 | 92.4 | 88.2 |
| RoB$_{large}$(Adpt$^P$) | 3.0M | 96.1 | 90.2 | 68.3 | 94.8 | 83.8 | 92.1 | 87.6 |
| RoB$_{large}$(Adpt$^P$) | 0.8M | 96.6 | 89.7 | 67.8 | 94.8 | 80.1 | 91.9 | 86.8 |
| RoB$_{large}$(Adpt$^H$) | 6.0M | 96.2 | 88.7 | 66.5 | 94.7 | 83.4 | 91.0 | 86.8 |
| RoB$_{large}$(Adpt$^H$) | 0.8M | 96.3 | 87.7 | 66.3 | 94.7 | 72.9 | 91.5 | 84.9 |
| RoB$_{large}$(LoRA) | 0.8M | 96.2 | 90.2 | 68.2 | 94.8 | 85.2 | 92.3 | 87.8 |
| RoB$_{large}$(KaSA) | 0.8M | **96.9** | **91.2** | **69.4** | **94.9** | **88.8** | **92.5** | **89.0** |

et al., 2022b) using the GLUE (General Language Understanding Evaluation) benchmark (Wang et al., 2018). The GLUE benchmark encompasses a wide array of datasets designed to test various aspects of NLU, including question answering, natural language inference, sentiment analysis, and textual entailment. In this context, our evaluation is conducted across 6 datasets from the GLUE: SST-2, MRPC, CoLA, QNLI, RTE, and STS-B. Detailed statistical information about the GLUE benchmark can be found in Appendix C.1.

**Implementation Details.** Basically, we follow the experimental setup applied in (Hu et al., 2021; Zhang et al., 2022) to ensure a fair comparison. We randomly initialize the knowledge-aware singular values without bias, which only introduces negligible $r$ coefficients in each layer. For all evaluated datasets in GLUE, we meticulously tune the hyperparameters, including the learning rates $lr \in [1\text{E-}5, 1\text{E-}3]$, the rank of SVD truncation $k \in \{1, 2, 4, 8, 16, 32, 64, 128\}$, and two trade-off loss coefficients $\beta \in [1\text{E-}5, 1]$ and $\gamma \in [1\text{E-}5, 1]$. The results we present are the median outcomes from 5 runs, each conducted with a distinct random seed. To maintain fair trainable parameters, we fine-tune the `query` and `value` weights in each Transformer block and set a rank $r = 8$ across all datasets. More detailed hyperparameters are presented in Appendix E.1.

**Main Results.** Table 1 presents the performance of RoBERTa-base and RoBERTa-large models fine-tuned using our KaSA in contrast to PEFT baselines. KaSA achieves the best performance across all datasets except MRPC and STS-B for the RoBERTa-base model. Notably, KaSA registers the highest average performances for both RoBERTa models: 86.3% for RoBERTa-base and 89.0% for RoBERTa-large. This underscores the effectiveness, adaptability, and scalability of our proposed approach. In a significant comparison with FFT, our KaSA, which utilizes merely up to 0.24% (approximately 0.3M/125.0M) of trainable parameters, outperforms FFT in 13 out of 14 scenarios and matches its performance on the STS-B dataset for the RoBERTa-base model. The results from DeBERTaV3-base are presented in Appendix F.1.

## 4.3 INSTRUCTION FOLLOWING

**Models and Datasets.** To validate KaSA's adaptability and versatility, we extend our experiments to include instruction tuning of LLaMA3 8B (Meta, 2024), Mistral 7B (Jiang et al., 2023), Gemma 7B (Gemma Team, 2024), and LLaMA2 13B (Touvron et al., 2023b). We fine-tune the models using four synthetic instruction-following datasets produced by GPT4o, each containing 128K samples, covering tasks such as summarization, classification, coding, and closed QA. Additionally, we fine-

Table 2: Instruction following evaluation results with average scores for the most popular LLMs fine-tuned on the 128k synthetic datasets and the Alpaca dataset, and evaluated by GPT4o and GPT4 with the scores within 10 on test subsets and MT-Bench, respectively.

| Model | Method | # Trainable Parameters | Classification | Summarization | Coding | Closed QA | MT-Bench |
|---|---|---|---|---|---|---|---|
| Gemma 7B | w/o FT | - | 2.41 | 2.28 | 3.07 | 2.95 | 2.56 |
| | FFT | 8.54B | 5.58 | 7.78 | 7.61 | **8.88** | 4.69 |
| | LoRA | 3.21M | 5.98 | 7.29 | 7.75 | 8.18 | 4.32 |
| | PiSSA | 3.21M | 6.23 | 7.88 | 7.80 | 8.22 | 4.66 |
| | MiLoRA | 3.21M | 6.30 | 7.62 | 7.71 | 8.27 | 4.53 |
| | **KaSA** | 3.22M | **6.88** | **7.92** | **8.01** | 8.69 | **4.97** |
| Mistral 7B | w/o FT | - | 2.31 | 2.81 | 2.32 | 3.02 | 1.16 |
| | FFT | 7.25B | **6.73** | **7.18** | **7.53** | **8.75** | 4.22 |
| | LoRA | 3.40M | 5.07 | 5.72 | 6.17 | 7.39 | 4.18 |
| | PiSSA | 3.40M | 5.46 | 5.86 | 6.41 | 7.24 | 4.24 |
| | MiLoRA | 3.40M | 5.33 | 5.89 | 6.52 | 7.28 | 4.29 |
| | **KaSA** | 3.41M | 5.72 | 6.82 | 6.74 | 7.75 | **4.58** |
| LLaMA3 8B | w/o FT | - | 2.04 | 2.03 | 2.86 | 3.33 | 3.11 |
| | FFT | 8.03B | 5.44 | 7.80 | 7.59 | **8.90** | 4.11 |
| | LoRA | 3.40M | 6.12 | 7.20 | 7.37 | 6.02 | 4.19 |
| | PiSSA | 3.40M | 6.35 | 7.31 | 7.59 | 6.18 | 4.26 |
| | MiLoRA | 3.40M | 6.37 | 7.61 | 7.65 | 6.39 | 4.32 |
| | **KaSA** | 3.41M | **6.55** | **7.83** | **7.89** | 6.81 | **4.71** |
| LLaMA2 13B | w/o FT | - | 1.00 | 1.08 | 1.01 | 1.27 | 1.01 |
| | FFT | 13.02B | 5.86 | **7.93** | 7.88 | **8.97** | 4.37 |
| | LoRA | 6.55M | 6.23 | 7.38 | 7.54 | 6.25 | 4.43 |
| | PiSSA | 6.55M | 6.47 | 7.45 | 7.83 | 6.54 | 4.39 |
| | MiLoRA | 6.55M | 6.45 | 7.63 | 7.85 | 6.82 | 4.51 |
| | **KaSA** | 6.56M | **6.86** | 7.92 | **8.09** | 7.12 | **4.95** |

tune using the Alpaca dataset (Taori et al., 2023b) and report the evaluation results on MT-Bench (Zheng et al., 2023), with GPT4 serving as the judge, yielding scores within 10. The detailed processing and statistical information of the synthetic datasets, Alpaca, and MT-Bench are presented in Appendix C.3 and C.4, respectively.

**Implementation Details.** Following the experimental setup in (Park et al., 2024), we use the summarization, classification, coding, and closed QA subsets from the "No Robots" (Rajani et al., 2023) dataset as seeds to create distinct synthetic datasets via GPT4o. We fine-tune the mentioned LLMs using these datasets and then prompt each fine-tuned model to generate four responses based on prompts sampled from the test subsets of the seed dataset. To ensure fair comparisons, we maintain a consistent fine-tuning and inference configuration across all fine-tuned models. We subsequently use GPT4o as a judge to apply single-answer grading strategies to evaluate the response quality of the fine-tuned LLMs on a scale from 1 to 10. For the Alpaca dataset, we fine-tune the specified models and prompt them to generate responses to questions from MT-Bench, with GPT4 serving as a judge, assigning scores within 10. Detailed prompts for data synthesis and performance evaluation, along with hyperparameter settings, are presented in Appendix C.3, D, and E.3, respectively.

**Main Results.** In Table 2, the results show that KaSA consistently surpasses LoRA, PiSSA, and MiLoRA across four 128k synthetic datasets, regardless of the model used. Notably, Gemma 7B and LLaMA3 8B, fine-tuned with KaSA, even surpass FFT in the classification, summarization, and coding datasets. In the evaluation using MT-Bench, KaSA consistently outperforms FFT and PEFT baselines on all models, showing remarkable efficacy. With significance tests showing ($p < 0.05$) in 9 out of 12 experimental settings on MT-Bench, KaSA demonstrates significant performance improvements over LoRA, PiSSA, and MiLoRA. These results further highlight the effectiveness, robustness, and adaptability of our method.

## 4.4 COMMONSENSE REASONING

**Models and Datasets.** Following (Wang et al., 2024a), we fine-tune the LLaMA2 7B (Touvron et al., 2023a) and the LLaMA3 8B (Meta, 2024) models using the Commonsense170K dataset, aiming to conduct a comprehensive evaluation across eight well-known commonsense reasoning tasks: BoolQ (Clark et al., 2019), PIQA (Bisk et al., 2020), SIQA (Sap et al., 2019), HellaSwag (Zellers et al., 2019), WinoGrande (Sakaguchi et al., 2021), ARC-e, ARC-c (Clark et al., 2018), and OBQA (Mihaylov et al., 2018).

Table 3: Performance comparison of LLaMA2 7B and LLaMA3 8B with different adaptation methods on eight commonsense reasoning datasets. The symbol † indicates that the results are taken from (Wang et al., 2024a). The best results are shown in **bold**. Higher is better for all tasks. ∗ denotes that the best results do not surpass ChatGPT.

| Model | Method | BoolQ | PIQA | SIQA | HellaSwag | WinoGrande | ARC-e | ARC-c | OBQA | Avg. |
|---|---|---|---|---|---|---|---|---|---|---|
| ChatGPT† | - | 73.1 | 85.4 | 68.5 | 78.5 | 66.1 | 89.8 | 79.9 | 74.8 | 77.0 |
| LLaMA2 7B | LoRA† | 69.8 | 79.9 | 79.5 | 83.6 | 82.6 | 79.8 | 64.7 | 81.0 | 77.6 |
| | PiSSA† | 67.6 | 78.1 | 78.4 | 76.6 | 78.0 | 75.8 | 60.2 | 75.6 | 73.8 |
| | MiLoRA† | 67.6 | 83.8 | 80.1 | 88.2 | 82.0 | 82.8 | 68.8 | 80.6 | 79.2 |
| | KaSA | **73.6** | **84.4**∗ | **80.2** | **91.5** | **84.5** | **84.7**∗ | **72.1**∗ | **81.2** | **81.5** |
| LLaMA3 8B | LoRA† | 70.8 | 85.2 | 79.9 | 91.7 | 84.3 | 84.2 | 71.2 | 79.0 | 80.8 |
| | PiSSA† | 67.1 | 81.1 | 77.2 | 83.6 | 78.9 | 77.7 | 63.2 | 74.6 | 75.4 |
| | MiLoRA† | 68.8 | 86.7 | 77.2 | 92.9 | **85.6** | 86.8 | 75.5 | 81.8 | 81.9 |
| | KaSA | **73.6** | **88.1** | **80.4** | **94.7** | 85.5 | **89.7**∗ | **79.4**∗ | **85.6** | **84.6** |

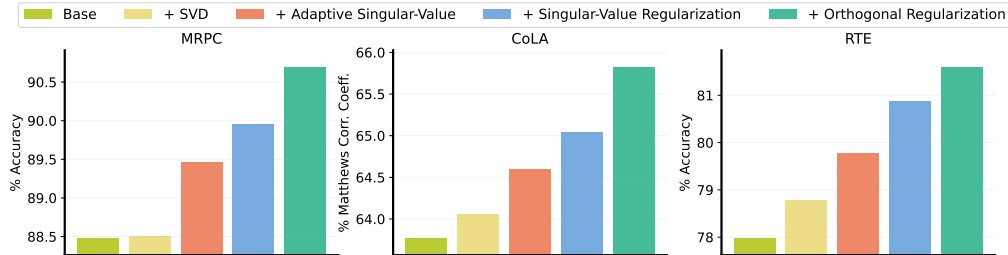

Figure 2: Components ablation study about knowledge-based SVD truncation, knowledge-aware singular value adaptation, singular value regularization $\mathcal{L}_2$, and orthogonal regularization $\mathcal{L}_3$ on MRPC, CoLA, and RTE datasets.

**Implementation Details.** To ensure a fair comparison, we implement our KaSA within the LLM-Adapters framework [2] (Hu et al., 2023), following MiLoRA (Wang et al., 2024a). We adhere strictly to the hyperparameter configurations for training and evaluation as specified by (Wang et al., 2024a) and (Hu et al., 2023), **without any tuning**, such as tuning the training epochs and learning rate. For detailed hyperparameters utilized, refer to Appendix E.4.

**Main Results.** As illustrated in Table 3, KaSA consistently surpasses all established baselines for both LLaMA2 7B and LLaMA3 8B across all eight benchmarks when using identical hyperparameter settings. Notably, KaSA achieves the highest average score, reflecting significant performance improvements across a diverse range of reasoning tasks. These results, obtained from rigorously controlled comparisons, align with our observations in NLU, NLG, and instruction following tasks. This consistency further corroborates the robustness and superiority of our method.

## 4.5 IN-DEPTH ANALYSIS

**Components Ablation Study.** Our method encompasses four principle components: knowledge-based SVD truncation, knowledge-aware singular value adaptation, singular value regularization $\mathcal{L}_2$, and orthogonal regularization $\mathcal{L}_3$. To examine the collective contributions of these components, we conduct ablation experiments on MRPC, CoLA, and RTE datasets from GLUE using the RoBERTa-base. Specifically, we compare KaSA with the following variants: (1) standard LoRA (as the base); (2) SVD truncation + LoRA; (3) SVD truncation + knowledge-aware singular-value adaptation; (4) SVD truncation + knowledge-aware singular-value adaptation + $\mathcal{L}_2$; (5) SVD truncation + knowledge-aware singular-value adaptation + $\mathcal{L}_2$ + $\mathcal{L}_3$. From the results in Figure 2, we observe that the model performances continually increase as more components are involved in the fine-tuning. The fifth bar in Figure 2 shows that variant (5), the full implementation of KaSA, achieves significant performance improvements across all three datasets. Conversely, excluding any of these components results in performance declines ranging from 2.05% to 3.25%, underscoring their collective importance in enhancing KaSA's effectiveness. Additional results of the components ablation study on SST-2, QNLI, and STS-B datasets are detailed in Appendix F.3.

---

[2]https://github.com/AGI-Edgerunners/LLM-Adapters

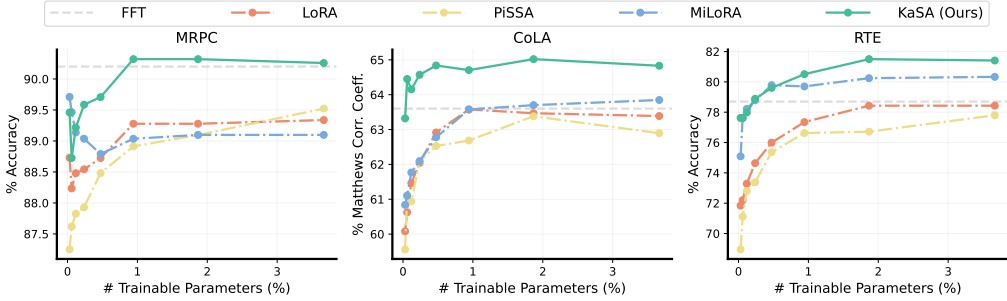

Figure 3: Budget parameter scalability of fine-tuning RoBERTa-base with LoRA, PiSSA, MiLoRA, and KaSA on MRPC, CoLA, and RTE datasets.

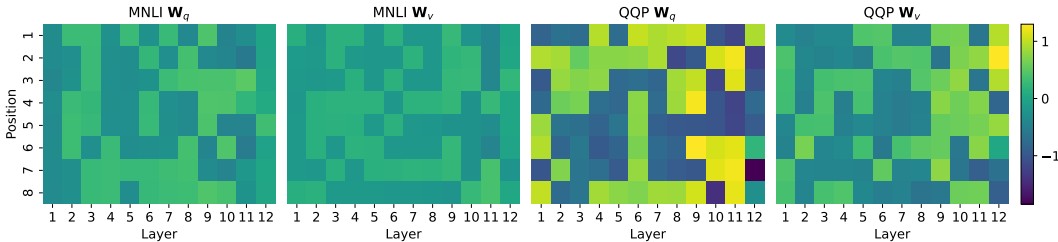

Figure 4: The final distribution of knowledge-aware singular values for $\mathbf{W}_q$ and $\mathbf{W}_v$ upon fine-tuning the RoBERTa-base model on the MNLI and QQP benchmarks. In this context, the $x$-axis corresponds to the layer index, and the $y$-axis denotes the position index. Each value signifies the relevance of the associated knowledge.

**Budget Parameter Scalability.** We compare the performance of fine-tuning RoBERTa-base with LoRA, PiSSA, MiLoRA, and KaSA across various scales of trainable parameters. Specifically, we employ these methods to the query and value weights of the transformer block and use a range of ranks $r = \{1, 2, 4, 8, 16, 32, 64, 128\}$ to control the parameter scales. Figure 3 shows that KaSA consistently outperforms LoRA, as well as the SVD-based baselines, at equivalent parameter scales across various datasets, indicating our method's efficacy and robustness. Moreover, we observe that enlarging trainable parameter scales does not invariably result in performance improvement. Notably, both methods peak in performance at $r = 8$, with KaSA enhancing LoRA by 1.96% on MRPC, 2.05% Mcc. on CoLA, and 2.53% Acc. on RTE.

**Knowledge-Aware Singular-Value.** The conventional FFT, which updates all parameters indiscriminately, often incorporates irrelevant or minimally contributory knowledge to the task at hand, leading to overfitting and a decline in model generalization capability (Valipour et al., 2023). To this end, we propose a novel knowledge-aware singular value module to adaptively activate the relevant task-specific knowledge. To validate our motivation, we visualize the knowledge-aware singular values of $\mathbf{W}_q$ and $\mathbf{W}_v$ when fine-tuning RoBERTa-base on the MNLI and QQP benchmarks, as depicted in Figure 4. We can clearly observe that different scales of singular values are allocated across different layers, indicating that it dynamically prioritizes knowledge across parameters.

## 5 CONCLUSION

In this paper, we introduce a PEFT method, KaSA, which incorporates SVD with knowledge-aware singular values for dynamic activation of parametric knowledge according to its relevance to the given tasks. KaSA commences with knowledge-based SVD truncation of minor singular value components to remove noisy knowledge within the base model. Subsequently, it reparameterizes task-specific updates in the SVD form, leveraging knowledge-aware singular values for dynamic knowledge activation according to relevance. Our extensive experiments on various LLMs across tasks in NLU, NLG, instruction following, and commonsense reasoning reveal that KaSA consistently surpasses FFT and a variety of prevailing PEFT baselines across well-known benchmarks and our synthetic datasets, highlighting the superiority of our method.

## ACKNOWLEDGMENTS

Jing Tang's work was partially supported by National Key R&D Program of China under Grant No. 2023YFF0725100 and No. 2024YFA1012701, by the National Natural Science Foundation of China (NSFC) under Grant No. 62402410 and No. U22B2060, by Guangdong Provincial Project (No. 2023QN10X025), by Guangdong Basic and Applied Basic Research Foundation under Grant No. 2023A1515110131, by Guangzhou Municipal Science and Technology Bureau under Grant No. 2023A03J0667 and No. 2024A04J4454, by Guangzhou Municipal Education Bureau (No. 2024312263), and by Guangzhou Municipality Big Data Intelligence Key Lab (No. 2023A03J0012), Guangzhou Industrial Information and Intelligent Key Laboratory Project (No. 2024A03J0628) and Guangzhou Municipal Key Laboratory of Financial Technology Cutting-Edge Research (No. 2024A03J0630). This work was also supported by IITP grant funded by the Korea government(MSIT)[RS-2023-00215959, Development of Access Agnostic wired and wireless integrated optical access technology].

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

## A  PSEUDOCODE FOR KASA

---

**Algorithm 1** PyTorch-style pseudocode for **KaSA**.

---

```
1  class KaSA(nn.Module):
2      def __init__(self,
3          rank: int = 8, # kasa rank
4          alpha: int = 16, # kasa alpha
5          base_layer: nn.Module # pre-trained layer
6      ):
7          # definitions
8          self.r = rank
9          self.alpha = alpha
10         self.scaling = alpha / rank
11         self.in_features, self.out_features = base_layer.in_features,
                base_layer.out_features
12
13         # Step 1: knowledge-based SVD truncation
14         self.svd_rank = self.in_features - self.r
15         U, S, Vh = torch.linalg.svd(base_layer.weight.data, full_matrices=False)
16         base_layer.weight.data = U[:, :self.svd_rank] @ torch.diag(S[:self.svd_rank]) @
                Vh[:self.svd_rank, :]
17         self.base_layer = base_layer
18
19         # Step 2: knowledge-aware singular-value adaptation
20         self.delta_v = nn.Linear(self.in_features, self.r, bias=False)
21         self.delta_sigma = nn.Parameter(torch.randn(self.r), requires_grad=True)
22         self.delta_u = nn.Linear(self.r, self.out_features, bias=False)
23
24     def forward(self, x: torch.Tensor):
25         # Step 3: merge W + Delta_W (Eq.7)
26         Delta_W = self.delta_u @ torch.diag(self.delta_sigma) @ self.delta_v
27         result = self.base_layer(x)
28         result = result + torch.einsum('ijk,kl->ijl', x, Delta_W) * self.scaling
29         return result
30
31 def regularization_loss(
32     model: nn.Module,
33     beta: float,
34     gamma: float
35 ):
36     # definitions
37     l2_loss = 0.0
38     l3_loss = 0.0
39     num_param = 0
40     for name, param in model.named_parameters():
41         if param.requires_grad:
42             # singular value regularization
43             if 'delta_sigma' in name:
44                 num_param += 1
45                 diag_norm = torch.sum(param ** 2)
46                 l2_loss += diag_norm
47             # orthogonal regularization
48             elif 'delta_v' in name or 'delta_u' in name:
49                 if 'delta_v' in name:
50                     matmul_result = torch.matmul(param.T, param)
51                 else:
52                     matmul_result = torch.matmul(param, param.T)
53
54                 I = torch.eye(matmul_result.size(0), device=matmul_result.device)
55                 diff_I = matmul_result - I
56                 matrix_loss = torch.norm(diff_I, p='fro')
57                 l3_loss += matrix_loss
58     auxi_loss = (beta * l2_loss + gamma * l3_loss) / num_param if num_param > 0 else 0.0
59     return auxi_loss
```

---

## B  BASELINES

To demonstrate its efficacy and robustness, we evaluate KaSA against FFT and multiple well-regarded PEFT baselines. The descriptions of our selective baselines are as follows:

- **Full fine-tuning (FFT)** initializes the base model with pre-trained weights and biases, updating all parameters during fine-tuning. Full fine-tuning typically serves as a comparative performance upper bound for PEFT methods (Valipour et al., 2023).

- **Bitfit** (Zaken et al., 2021) fine-tunes the bias vectors, leaving other model parameters unchanged.

- **Adapter tuning** integrates tunable adapter layers into Transformer blocks, featuring a pair of down-projection and up-projection matrices with a non-linear activation function in between. We compare four Adapter variants: **Adapter$^H$** (Houlsby et al., 2019) inserts adapter layers after the attention and the feed-forward block to fine-tune. **Adapter$^D$** (Rücklé et al., 2021) discards non-activated adapters to improve fine-tuning efficiency. **Adapter$^L$** (Lin et al., 2020) employs an efficient design, placing adapter layers after the MLP module and LayerNorm. **Adapter$^P$** (Pfeiffer et al., 2021) applies adapter after the feed-forward layer and employs a two-stage learning strategy to enhance multi-task performance.
- **LoRA** (Hu et al., 2021) only fine-tunes a pair of low-rank matrices to approximate the task-specific knowledge updates, effectively diminishing the number of trainable parameters.
- **AdaLoRA** (Zhang et al., 2022) reparameterizes task-specific knowledge updates in the SVD form and adaptively allocates the parameter budget through pruning the less important singular values.
- **DyLoRA** (Valipour et al., 2023) dynamically trains LoRA for a range of ranks, reducing the training time to find a fixed, optimal rank.
- **VeRA** (Kopiczko et al., 2023) employs learnable vectors to adapt a shared pair of frozen random matrices across layers to reduce the trainable parameters count.
- **DoRA** (Liu et al., 2024) decomposes the base model weights into magnitude and direction components for fine-tuning, reducing the number of trainable parameters.
- **PiSSA** (Meng et al., 2024) performs SVD to portion the base model into principal components with larger singular values and residual components with smaller ones, fine-tuning the low-rank matrices initialized with the principle components while keeping the residual components unchanged.
- **MiLoRA** (Wang et al., 2024a) also utilizes SVD for parameter initialization but diverges from PiSSA by fine-tuning low-rank matrices initialized with residual components and maintaining the principal ones unchanged.
- **SARA** (Gu et al., 2024) conducts SVD at the initialization stage to adaptively find the appropriate rank for each layer.
- **CorDA** (Yang et al., 2024) performs SVD on the base model, oriented by the covariance matrix that encodes the context of the target task. CorDA supports two fine-tuning modes: 1) initializing the tunable low-rank matrices with principal components for enhanced performance; and 2) freezing the principle components while using minor components to initialize tunable matrices, thereby preserving world knowledge.

## C  DETAILS OF BENCHMARK DATASETS

### C.1  GLUE BENCHMARK

For natural language understanding (NLU), we employ the GLUE benchmark (Wang et al., 2018), which is a widely used benchmark containing a collection of 8 NLU datasets, including CoLA, SST-2, MRPC, STS-B, QQP, MNLI, QNLI, and RTE. We present the statistical information of the GLUE benchmark in the table below.

### C.2  E2E NLG CHALLENGE

For natural language generation (NLG), we utilize the E2E (End-to-End) NLG Challenge dataset (Novikova et al., 2017), which is commonly used for the evaluation of natural language generation models. This dataset includes approximately 42k training samples, 4.6k validation samples, and 4.6k test samples from the restaurant domain. The E2E dataset involves evaluations across five metrics: BLEU, NIST, METEOR, ROUGE-L, and CIDEr. Detailed explanations of these metrics are as follows:

- **BLEU** (Bilingual Evaluation Understudy) evaluates the quality of machine-generated text by comparing it to one or more human-generated reference translations.

Table 4: Overview of task descriptions and dataset statistics within the GLUE benchmark.

| Corpus | Task | # Train | # Val | # Test | # Labels | Metrics | Domain |
|--------|------|---------|-------|--------|----------|---------|--------|
| | | | | Single-Sentence Tasks | | | |
| CoLA | Acceptability | 8.55k | 1.04k | 1.06k | 2 | Matthews Corr. | misc. |
| SST-2 | Sentiment | 67.3k | 872 | 1.82k | 2 | Accuracy | Movie reviews |
| | | | | Similarity and Paraphrase Tasks | | | |
| MRPC | Paraphrase | 3.67k | 408 | 1.73k | 2 | Accuracy/F1 | News |
| STS-B | Sentence similarity | 5.75k | 1.5k | 1.38k | 1 | Pearson/Spearman Corr. | misc. |
| QQP | Paraphrase | 364k | 40.4k | 391k | 2 | Accuracy/F1 | Social QA |
| | | | | Inference Tasks | | | |
| MNLI | NLI | 393k | 19.65k | 19.65k | 3 | Accuracy | misc. |
| QNLI | QA/NLI | 105k | 5.46k | 5.46k | 2 | Accuracy | Wikipedia |
| RTE | NLI | 2.49k | 277 | 3k | 2 | Accuracy | News & Wikipedia |

- **NIST** (National Institute of Standards and Technology) evaluates the quality of machine-generated text by calculating the similarity between a machine output and a reference text using weighted average of n-grams precision.

- **METEOR** (Metric for Evaluation of Translation with Explicit ORdering) measures the alignment between the machine-generated and reference texts by calculating a score based on the harmonic mean of precision and recall.

- **ROUGE-L** (Recall-Oriented Understudy for Gisting Evaluation) measures the longest common subsequence(LCS) between the machine output and the reference. It specifically focuses on the sequence of words, making it sensitive to the fluency and order of information in the generated text.

- **CIDEr** (Consensus-based Image Description) measures the similarity of the machine-generated text and the human-generated ground truth by considering both the n-gram overlap and the consensus among human annotators.

## C.3    SYNTHETIC DATASET

For instruction following tasks, we employ synthetic datasets generated using GPT4o, based on the foundational "No Robots" seed dataset (Rajani et al., 2023). Task-specific subsets, including summarization, classification, coding, and closed QA, serve as seeds for generating synthetic data through the framework proposed by (Park et al., 2024). Table 5 presents the volume of data samples and token-level statistical information for these task-specific synthetic subsets.

Table 5: Data volume and token-level statistics of the train and test synthetic datasets generated by GPT4o for each instruction-following task.

| Task | Split | Data Volume | | Token-level Statistics | | | |
|------|-------|------|-----------|-----|-----|------|------|
| | | Seed | Synthesis | Min | Max | Avg. | Std. |
| Summarization | Train | 395 | 128K | 10 | 2,386 | 95 | 53 |
| | Test | 25 | 100 | 148 | 1,150 | 426 | 245 |
| Classification | Train | 334 | 128K | 6 | 2,159 | 67 | 37 |
| | Test | 16 | 64 | 46 | 520 | 119 | 109 |
| Coding | Train | 334 | 128K | 9 | 6,518 | 151 | 84 |
| | Test | 16 | 64 | 49 | 821 | 317 | 189 |
| Closed QA | Train | 245 | 128K | 12 | 1,701 | 135 | 59 |
| | Test | 15 | 60 | 126 | 1,578 | 411 | 378 |

## C.4    ALPACA AND MT-BENCH

**Alpaca** (Taori et al., 2023a) is a well-known instruction dataset that contains 51k instruction-following demonstrations generated by text-davinci-003. These data are synthesized using an improved self-instruct method (Wang et al., 2023). The dataset is designed for instruction-tuning LLMs

to improve their ability to follow instructions. Each sample includes an instruction, an input (if applicable), and an output. A specific example is presented below.

```
{
  "instruction": "Create a classification task by clustering the given
      list of items.",
  "input": "Apples, oranges, bananas, strawberries, pineapples",
  "output": "Class 1: Apples, Oranges\nClass 2: Bananas,
      Strawberries\nClass 3: Pineapples",
  "text": "Below is an instruction that describes a task, paired with an
      input that provides further context. Write a response that
      appropriately completes the request.\n\n### Instruction:\nCreate a
      classification task by clustering the given list of items.\n\n###
      Input:\nApples, oranges, bananas, strawberries, pineapples\n\n###
      Response:\nClass 1: Apples, Oranges\nClass 2: Bananas,
      Strawberries\nClass 3: Pineapples"
}
```

The **instruction** describes the targeted task to be performed by the model. Each of the 52k instructions is unique. The **input** can represent the optional input to the task or serve as the additional context to the corresponding instruction. The **output** is the response generated by text-davinci-003 to the associated instruction. The **Text** is the formatted combination of the instruction, input, and output, using the prompt template for fine-tuning models.

**MT-bench** (Zheng et al., 2023) contains 80 predefined open-ended questions across diverse domains such as writing, roleplay, reasoning, math, coding, extraction, STEM, and humanities. These challenging questions are designed to automatically assess an LLM's instruction-following capabilities, with advanced service LLMs like GPT-4 acting as judges. Below is an example from MT-bench.

```
{
  "question_id": 101,
  "category": "reasoning",
  "turns": [
    "Imagine you are participating in a race with a group of people. If
        you have just overtaken the second person, what's your current
        position? Where is the person you just overtook?",
    "If the \"second person\" is changed to \"last person\" in the above
        question, what would the answer be?"
  ],
  "reference": [
    "You are in second place.",
    "Uncertain."
  ]
}
```

## C.5 COMMONSENSE REASONING

The Commonsense170K dataset (Hu et al., 2023) contains data samples from eight well-known commonsense reasoning tasks:

- **BoolQ** (Clark et al., 2019) dataset comprises 15,942 naturally occurring yes/no questions, generated in unprompted and unconstrained settings.

- **PIQA** (Bisk et al., 2020) dataset consists of samples structured as multiple-choice questions, each presenting a question with two possible solutions that require physical commonsense to answer.

- **SIQA** (Sap et al., 2019) dataset contains multiple-choice questions regarding the pragmatic implications of social events, which can measure LLMs' abilities to address social commonsense reasoning.

- **HellaSwag** (Zellers et al., 2019) dataset includes commonsense natural language inference questions, offering a context and multiple endings to complete it.

- **WinoGrande** (Sakaguchi et al., 2021) dataset is structured as a fill-in-the-blank task with two options, designed to test a model's ability to correctly solve the problem using commonsense reasoning.

- **ARC-e** and **ARC-c** are the Easy and Challenge Set of the ARC (Clark et al., 2018) dataset, which contains grade-school level, multiple-choice science questions. Notably, the Challenge Set includes questions answered incorrectly by both the retrieval-based algorithm and word co-occurrence algorithm.

- **OBQA** (Mihaylov et al., 2018) dataset contains multiple-choice elementary-level science questions requiring multi-step reasoning, use of additional common and provided science facts (open book), and rich text comprehension.

# D    PROMPT TEMPLATES

Following the typical practices of (Wang et al., 2023) and (Zheng et al., 2023), we leverage two specialized prompt templates: 1) one for generating synthetic datasets and 2) another for evaluating the outputs of fine-tuned LLMs. To be specific, Figure 5 presents the prompt template crafted for generating synthetic data aimed at the summarization task, whereas Figure 6 shows the prompt template for other tasks. We guide GPT4o in generating analogous data samples by using a reference example pair consisting of a prompt $instruction and its corresponding response $response from the training subset of the seed dataset. In addition, the template is designed to request multiple synthetic data samples in a single query, thus maximizing the efficiency of API use. On the other hand, Figure 7 shows the prompt template used for assessing the precision and similarity between the response $lm_response and $human_response given the same $instruction from the test subset of the seed dataset, where the $ symbol indicates a placeholder, designed to be substituted with actual data during the runtime. We only report the precision results in our experiments for the sake of brevity. Given the unique features of different downstream tasks, there is no optimal prompt template that universally applies. Therefore, the actual content of the prompt template is adjusted to align with the specific requirements of the task for which the synthetic dataset is being generated.

---

**Prompt of Data Synthesis for Summarization Task**

Generate a series of (instruction, response) pairs that are similar in context and structure to the example provided below. Each pair should consist of a concise instruction followed by an appropriate, detailed response. The instruction should pose a clear task or question, while the response should provide a comprehensive answer or solution that could be understood by someone with a basic understanding of the subject.

Example pair:
Instruction: $instruction
Response: $response

Your task is to generate more pairs that maintain this level of clarity and detail. The topic is $topic. Write a long text of instruction by yourself, then summarize the given instruction in a response. Ensure that the responses are informative and accurate, suitable for an educational context.

Store the generated pairs in JSON format, with each pair as an object within an array. Each object should have two key-value pairs: "instruction" and "response". For instance:

```
{
  "contents":
  [
    {"instruction": "text", "response": "text"},
    {"instruction": "text", "response": "text"},
    ...
  ]
}
```

Remember to maintain consistency in the format and ensure the generated pairs are diverse and cover a broad range of subjects. You must return the response in the asked format and you must not add any additional text in your response.

---

Figure 5: Prompt template of data synthesis for summarization tasks by GPT4o.

---

**Prompt of Data Synthesis for Classification, Coding, and Closed QA Tasks**

Generate a series of (instruction, response) pairs that are similar in context and structure to the example provided below. Each pair should consist of a concise instruction followed by an appropriate, detailed response. The instruction should pose a clear task or question, while the response should provide a comprehensive answer or solution that could be understood by someone with a basic understanding of the subject.

Example pair:
Instruction: $instruction
Response: $response

Your task is to generate more pairs that maintain this level of clarity and detail. The topic is $topic. Ensure that the responses are informative and accurate, suitable for an educational context.

Store the generated pairs in JSON format, with each pair as an object within an array. Each object should have two key-value pairs: "instruction" and "response". For instance:

```
{
  "contents":
  [
    {"instruction": "text", "response": "text"},
    {"instruction": "text", "response": "text"},
    …
  ]
}
```

Remember to maintain consistency in the format and ensure the generated pairs are diverse and cover a broad range of subjects. You must return the response in the asked format and you must not add any additional text in your response.

---

Figure 6: Prompt template of data synthesis for classification, coding, and closed QA tasks by GPT4o.

---

**Generated Text Assessment Prompt**

You are a meticulous evaluator assessing the quality of a response generated for a specific instruction. Your task is to assign a score between 1 and 10 (whole numbers only, no decimals) based on how well the response satisfies the requirements of the instruction. Consider the following criteria:

1. Completeness: Does the response fully address all aspects of the instruction?
2. Relevance: Is the response focused and aligned with the instruction's requirements?
3. Clarity: Is the response clear and easy to understand?

Provide a brief justification for your score, highlighting key strengths or weaknesses in the response. Output your evaluation in the following JSON format:
{"score": [integer score between 1 and 10], "justification": "[brief explanation of the score]"}

Instruction:
$instruction

Response:
$lm_response

Example Output:
```
{
    "score": 9,
    "justification": "The response is complete, relevant, and mostly clear, with minor areas for improvement in phrasing."
}
```

---

Figure 7: Prompt template to evaluate the fine-tuned model's response by GPT4o.

# E   TRAINING DETAILS

## E.1   NATURAL LANGUAGE UNDERSTANDING

For NLU tasks, we align with the experimental setup detailed in (Hu et al., 2021; Zhang et al., 2022) for a fair comparison. The detailed configurations of KaSA for RoBERTa-base, RoBERTa-large, and DeBERTaV3-base on the GLUE benchmark are depicted in Table 6 and Table 7, respectively. It is important to note that our adaptation process for the MRPC, RTE, and STS-B tasks begins with the pre-trained RoBERTa model, rather than a model that has already been adapted to MNLI. As a result, we fine-tune the models on all datasets starting from their original pre-trained weights. The results we present are the median results from 5 runs, each conducted with a distinct random seed.

Table 6: The hyperparameters we used for RoBERTa-base and RoBERTa-large on the GLUE benchmark.

| Model | Settings | MNLI | SST-2 | MRPC | CoLA | QNLI | QQP | RTE | STS-B |
|---|---|---|---|---|---|---|---|---|---|
| Common | Optimizer | | | | AdamW | | | | |
| | Warmup Ratio | | | | 0.06 | | | | |
| | LR Schedule | | | | Linear | | | | |
| RoBERTa$_{base}$ | Batch Size | 32 | 128 | 32 | 32 | 32 | 128 | 32 | 32 |
| | # Epochs | 100 | 100 | 100 | 100 | 10 | 100 | 100 | 40 |
| | Learning Rate | 5E-04 | 5E-04 | 4E-04 | 4E-04 | 4E-04 | 5E-04 | 4E-04 | 3E-04 |
| | Weight Decay | 0.0 | 0.0 | 0.0 | 0.0 | 0.0 | 0.0 | 0.0 | 0.0 |
| | KaSA Rank | | | | $r_{query} = r_{value} = 8$ | | | | |
| | KaSA $\alpha$ | | | | 16 | | | | |
| | KaSA $\beta$ | 2.4E-3 | 1E-04 | 1E-01 | 1E-04 | 1E-02 | 1E-4 | 2.4E-01 | 1E-04 |
| | KaSA $\gamma$ | 2.4E-4 | 1E-03 | 1E-03 | 1E-03 | 1E-05 | 1E-3 | 2.4E-04 | 1E-05 |
| | KaSA Dropout | 0.0 | 0.0 | 0.0 | 0.0 | 0.0 | 0.0 | 0.0 | 0.0 |
| | Max Seq. Len. | 512 | 512 | 512 | 512 | 512 | 512 | 512 | 512 |
| RoBERTa$_{large}$ | Batch Size | - | 64 | 32 | 32 | 8 | - | 32 | 32 |
| | # Epochs | - | 10 | 10 | 100 | 20 | - | 100 | 20 |
| | Learning Rate | - | 4E-04 | 3E-04 | 3E-04 | 4E-04 | - | 4E-04 | 3E-04 |
| | Weight Decay | - | 0.1 | 0.1 | 0.0 | 0.0 | - | 0.0 | 0.0 |
| | KaSA Rank | | | | $r_{query} = r_{value} = 8$ | | | | |
| | KaSA $\alpha$ | | | | 16 | | | | |
| | KaSA $\beta$ | - | 1E-04 | 1E-02 | 2.4E-01 | 1E-02 | - | 1E-04 | 1E-03 |
| | KaSA $\gamma$ | - | 1E-04 | 1E-02 | 2.4E-04 | 1E-03 | - | 1E-03 | 1E-02 |
| | KaSA Dropout | - | 0.0 | 0.0 | 0.0 | 0.0 | - | 0.0 | 0.0 |
| | Max Seq. Len. | - | 512 | 512 | 512 | 512 | - | 512 | 128 |

Table 7: The hyperparameters we used for DeBERTaV3-base on the GLUE benchmark.

| Model | Settings | SST-2 | MRPC | CoLA | QNLI | RTE | STS-B |
|---|---|---|---|---|---|---|---|
| | Optimizer | | | AdamW | | | |
| | Warmup Ratio | | | 0.06 | | | |
| | LR Scheduler | | | Linear | | | |
| DeBERTaV3-base | Batch size | 128 | 32 | 32 | 16 | 32 | 32 |
| | # Epochs | 10 | 10 | 100 | 20 | 100 | 20 |
| | Learning Rate | 5E-4 | 4E-4 | 4E-4 | 4E-4 | 5E-4 | 4E-4 |
| | Weight Decay | 0.0 | 0.0 | 0.0 | 0.0 | 0.0 | 0.0 |
| | KaSA Rank | | | $r_{query} = r_{value} = 8$ | | | |
| | KaSA $\alpha$ | | | 16 | | | |
| | KaSA $\beta$ | 1E-04 | 1.0 | 2.4E-01 | 1E-01 | 1E-04 | 1E-01 |
| | KaSA $\gamma$ | 1E-03 | 1.0 | 2.4E-04 | 1E-01 | 1E-03 | 1E-01 |
| | KaSA Dropout | 0.0 | 0.0 | 0.0 | 0.0 | 0.0 | 0.0 |
| | Max Seq. Len. | 512 | 512 | 64 | 512 | 512 | 512 |

## E.2 NATURAL LANGUAGE GENERATION

For NLG tasks, our KaSA adheres to the experimental setup outlined in (Hu et al., 2021; Gu et al., 2024) to ensure a fair comparison. The comprehensive configurations of KaSA for GPT-2 Medium and GPT-2 Large models on the E2E NLG Challenge benchmark are depicted in Table 8.

## E.3 INSTRUCTION FOLLOWING

For instruction following tasks, we adopt the framework proposed by (Park et al., 2024) to streamline the processes of data synthesis, fine-tuning, and evaluation. We fine-tune several of the most popular LLMs, including LLaMA3 8B, Mistal 7B, Gemma 7B, and LLaMA2 13B, using KaSA and different PEFT baselines to facilitate comparative analysis. Detailed hyperparameter configurations are provided in Table 9.

Table 8: The hyperparameters for GPT-2 on E2E NLG Challenge.

| Stage | Settings | Medium | Large |
|---|---|---|---|
| Training | Optimizer | AdamW | |
| | Weight Decay | 0.01 | 0.01 |
| | Dropout Prob | 0.1 | 0.1 |
| | Batch Size | 8 | |
| | # Epoch | 5 | |
| | Warmup Steps | 500 | |
| | LR Scheduler | Linear | |
| | Label Smooth | 0.1 | 0.1 |
| | Learning Rate | 2E-4 | |
| | KaSA Rank | $r_{query} = r_{value} = 4$ | |
| | KaSA $\alpha$ | 32 | |
| | KaSA $\beta$ | 1E-4 | |
| | KaSA $\gamma$ | 1E-3 | |
| Inference | Beam Size | 10 | |
| | Length Penalty | 0.9 | 0.8 |
| | no repeat ngram size | 4 | |

Table 9: Detailed configurations used for the instruction following task.

| Stage | Settings | Classification | Summarization | Coding | Closed QA | MT-Bench |
|---|---|---|---|---|---|---|
| Training | Optimizer | | | AdamW | | |
| | Batch Size | | Gemma 7B = 8, Mitral 7B = LLaMA3 8B = 16 | | | |
| | # Epoch | | | 1 | | |
| | Warmup Ratio | | | 0.1 | | |
| | Data Type | | | Bfloat16 | | |
| | LR Scheduler | | | Cosine | | |
| | Learning Rate | | | 2.0E-04 | | |
| | KaSA Rank | | | $r_{query} = r_{value} = 8$ | | |
| | KaSA $\alpha$ | | | 16 | | |
| | KaSA $\beta$ | | | 1E-4 | | |
| | KaSA $\gamma$ | | | 1E-3 | | |
| | KaSA Dropout | | | 0.05 | | |
| | Max Seq. Len. | | | 512 | | |
| Inference | Number of Beams | | | 10 | | |
| | Length Penalty | | | 0.8 | | |
| | No Repeat N-Gram Size | | | 4 | | |

### E.4 COMMONSENSE REASONING

We adhere strictly to the hyperparameter configurations for training and evaluation as specified by (Wang et al., 2024a) and (Hu et al., 2023), **without any tuning**. The specific hyperparameter configurations used are shown in Table 10.

## F ADDITIONAL EXPERIMENTAL RESULTS

### F.1 NATURAL LANGUAGE UNDERSTANDING ON DEBERTAV3-BASE

As demonstrated in Table 11, the DeBERTaV3-base results consistently surpass all baseline performances across the datasets, with the exception of STS-B, achieving the highest average performance of 88.72%. This further validates the efficacy of our method across different model architectures.

### F.2 NATURAL LANGUAGE GENERATION

**Models and Datasets.** For NLG tasks, we employ KaSA and other PEFT baselines to fine-tune both GPT-2 Medium (355M) and GPT-2 Large (774M) models (Radford et al., 2019) on the well-established E2E (End-to-End) NLG Challenge benchmark (Novikova et al., 2017), which focuses on restaurant domain information. The statistics of the E2E NLG Challenge benchmark and the evaluation metrics applied are detailed in C.2.

Table 10: The hyperparameter configurations for LLaMA2 7B and LLaMA3 8B on commonsense reasoning tasks. To ensure a fair comparison, these configurations remain consistent across LoRA, PiSSA, and MiLoRA, with the exception of the specific hyperparameters unique to KaSA, namely $\beta$ and $\gamma$, as well as PiSSA and MiLoRA, where $\alpha = 32$.

| Hyperparameters | Commonsense Reasoning | |
|---|---|---|
| | LLaMA2 7B | LLaMA3 8B |
| Optimizer | AdamW | |
| Batch Size | 16 | |
| # Epoch | 3 | |
| Warmup Steps | 100 | |
| LR Scheduler | Linear | |
| Learning Rate | 3E-4 | |
| KaSA Rank | 32 | |
| KaSA $\alpha$ | 64 | |
| Dropout Prob | 0.05 | |
| KaSA $\beta$ | 1E-2 | 1E-4 |
| KaSA $\gamma$ | 1E-3 | 1E-3 |
| Placement | query, key, value, MLP up, MLP down | |

Table 11: Performance of DeBERTaV3-base ($\text{DeB}_{v3}$) with different adaptation methods on 6 datasets of the GLUE benchmark. We report the average result of five runs with different random seeds. The best results for each dataset are shown in **bold**. Higher is better for all metrics.

| Model(Method) | # Trainable Parameters | SST-2 (Acc.) | MRPC (Acc.) | CoLA (Mcc.) | QNLI (Acc.) | RTE (Acc.) | STS-B (Pcc.) | All Avg. |
|---|---|---|---|---|---|---|---|---|
| $\text{DeB}_{v3}$(FFT) | 184.0M | 95.63 | 89.46 | 69.19 | 94.03 | 83.75 | 91.60 | 87.28 |
| $\text{DeB}_{v3}$(Adpt$^H$) | 0.6M | 95.30 | 89.22 | 67.87 | 93.76 | 85.56 | 91.30 | 87.17 |
| $\text{DeB}_{v3}$(Adpt$^P$) | 0.6M | 95.53 | 89.22 | 69.48 | 93.98 | 84.12 | 91.52 | 87.31 |
| $\text{DeB}_{v3}$(LoRA) | 0.3M | 94.95 | 89.71 | 68.71 | 94.03 | 85.56 | **91.68** | 87.44 |
| $\text{DeB}_{v3}$(AdaLoRA) | 0.3M | 95.80 | 90.44 | 70.04 | 94.49 | 87.36 | 91.63 | 88.29 |
| $\text{DeB}_{v3}$(PiSSA) | 0.3M | 95.30 | **91.42** | 70.29 | 93.59 | 84.84 | 91.37 | 87.80 |
| $\text{DeB}_{v3}$(MiLoRA) | 0.3M | 95.99 | 89.71 | 70.34 | 94.14 | 85.92 | 90.28 | 87.73 |
| $\text{DeB}_{v3}$(KaSA) | 0.3M | **96.22** | **91.42** | **70.41** | **94.55** | **88.09** | 91.62 | **88.72** |

**Implementation Details.** We adopt the experimental configurations delineated in (Hu et al., 2021; Gu et al., 2024) for the fine-tuning of `query` and `value` weights within each Transformer block, setting a rank of $r = 4$. The AdamW optimizer is employed, paired with a linear learning rate schedule over 5 epochs. The reported results represent the mean outcomes from 3 runs, each initialized with a distinct random seed, selecting the performance at the last epoch of each run for comparison. For further details on the hyperparameters utilized, refer to E.2.

**Main Results.** We present the performance comparison in Table 12. As can be seen, our method consistently outshines the baselines in language generation capabilities across various evaluated metrics. More specifically, regarding the GPT-2 Medium model, KaSA outperforms the baselines in 4 out of 5 metrics and achieves comparable performance (72.1 vs. 72.3) in the ROUGE-L metric with the top-performing baseline, SARA. In the GPT-2 Large model, KaSA surpasses the baselines across all metrics, further confirming its superior performance and scalability.

## F.3 Components Ablation Study on SST-2, QNLI, and STS-B

Figure 8 shows the results of ablation studies conducted on the SST-2, QNLI, and STS-B datasets. From the results, we observe that: 1) the model's performance consistently improves with the inclusion of additional components during fine-tuning; 2) excluding any of these components leads to a decline in performance. These findings align with that observed in Section 4.5, emphasizing the effectiveness of each designed principal component of KaSA in enhancing model performance.

Table 12: Performance of GPT-2 Medium and Large models with different adaptation methods on the E2E NLG Challenge. For all metrics, higher values indicate better performance. $^*$ indicates that the results are reported in prior works. Best results are shown in **bold**.

| Model(Method) | # Trainable Parameters | BLEU | NIST | METEOR | ROUGE-L | CIDEr |
|---|---|---|---|---|---|---|
| GPT-2$_{\text{Medium}}$(FFT$^*$) | 354.92M | 68.2 | 8.62 | 46.2 | 71.0 | 2.47 |
| GPT-2$_{\text{Medium}}$(Adpt$^L*$) | 0.37M | 66.3 | 8.41 | 45.0 | 69.8 | 2.40 |
| GPT-2$_{\text{Medium}}$(Adpt$^L*$) | 11.09M | 68.9 | 8.71 | 46.1 | 71.3 | 2.47 |
| GPT-2$_{\text{Medium}}$(Adpt$^H*$) | 11.09M | 67.3 | 8.50 | 46.0 | 70.7 | 2.44 |
| GPT-2$_{\text{Medium}}$(LoRA$^*$) | 0.35M | 70.4 | 8.85 | 46.8 | 71.8 | 2.53 |
| GPT-2$_{\text{Medium}}$(AdaLoRA) | 0.38M | 68.2 | 8.58 | 44.1 | 70.7 | 2.35 |
| GPT-2$_{\text{Medium}}$(DyLoRA) | 0.39M | 69.2 | 8.75 | 46.3 | 70.8 | 2.46 |
| GPT-2$_{\text{Medium}}$(VeRA) | 0.098M | 69.1 | 8.71 | 46.3 | 70.8 | 2.43 |
| GPT-2$_{\text{Medium}}$(SARA) | 0.33M | 70.4 | 8.84 | 46.7 | **72.3** | **2.55** |
| GPT-2$_{\text{Medium}}$(KaSA) | 0.35M | **70.6** | **8.86** | **46.9** | 72.1 | **2.55** |
| GPT-2$_{\text{Large}}$(FFT$^*$) | 774.03M | 68.5 | 8.78 | 46.0 | 69.9 | 2.45 |
| GPT-2$_{\text{Large}}$(Adpt$^L*$) | 0.88M | 69.1 | 8.68 | 46.3 | 71.4 | 2.49 |
| GPT-2$_{\text{Large}}$(Adpt$^L*$) | 23.00M | 68.9 | 8.70 | 46.1 | 71.3 | 2.45 |
| GPT-2$_{\text{Large}}$(LoRA$^*$) | 0.77M | 70.4 | 8.89 | 46.8 | **72.0** | 2.47 |
| GPT-2$_{\text{Large}}$(KaSA) | 0.77M | **70.5** | **8.90** | **47.0** | **72.0** | **2.50** |

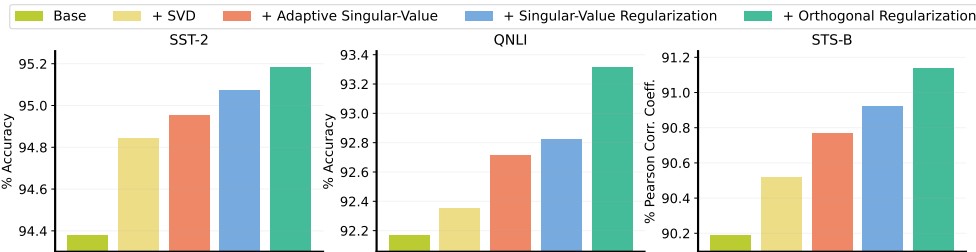

Figure 8: Components ablation study about knowledge-based SVD truncation, knowledge-aware singular value adaptation, singular value regularization $\mathcal{L}_2$, and orthogonal regularization $\mathcal{L}_3$ on SST-2, QNLI, and STS-B datasets.

## F.4 RANK $k$ OF KNOWLEDGE-BASED SVD TRUNCATION

As depicted in Section 1, components of the original base model weight matrix $\mathbf{W}^{(0)}$ associated with smaller singular values are identified to contain noise or less relevant information (Sharma et al., 2023; Wang et al., 2024a). This presence can adversely affect the convergence of model training and its overall efficacy. We propose the truncation of these components to refine the focus of the base model towards more pertinent knowledge domains, thereby mitigating the adverse impacts. Therefore, we delve into the impact of varying the rank (denoted as $k \in \{1, 2, 4, 8, 16, 32, 64, 128\}$) of SVD truncation on the model's performance, using RoBERTa-base on the MRPC, CoLA, and RTE datasets. As illustrated in Figure 9, an enhancement in model performance is observed as $k$ increases from 1 to 8. Conversely, an escalation in $k$ from 8 to 128 results in a decrement in performance. This observation highlights the criticality of identifying an optimal SVD truncation rank that achieves a delicate balance between incorporating world knowledge with large singular values and excluding disruptive noise information with smaller singular values, thereby optimizing model performance. The adaptive determination of the optimal SVD truncation rank emerges as a compelling avenue for future research.

## F.5 RANK $r$ OF KNOWLEDGE-AWARE SINGULAR-VALUE ADAPTATION

We explore the impact of different rank settings on performance across a range of tasks. Specifically, our analysis focuses on LoRA, MiLoRA, PiSSA, and KaSA, using ranks ranging from $r = \{1, 2, 4, 8, 16, 32, 64, 128\}$ on the CoLA, MRPC, and RTE datasets. As presented in Table 13, KaSA consistently surpasses the baselines across various rank settings in 92 out of 96 cases across the four datasets, highlighting the efficacy and robustness of our proposed method. To further our investigation, we increase the rank to 128 and compare KaSA with LoRA, DoRA (Liu et al., 2024),

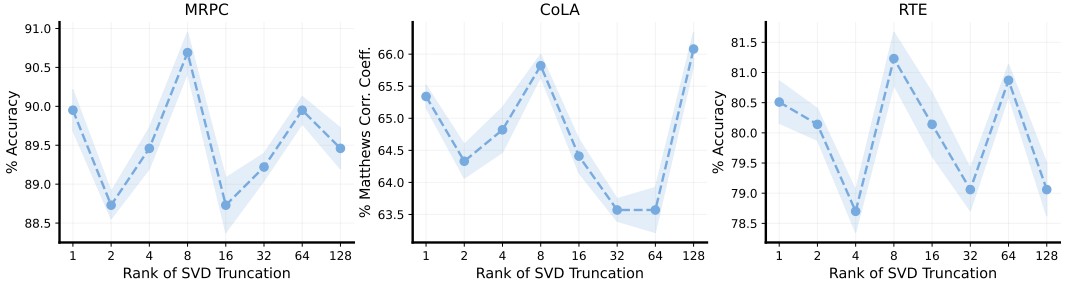

Figure 9: The impact of varying the rank of SVD truncation on the model's performance across three datasets.

Table 13: Performance comparison of LoRA and SVD-based baselines on CoLA, MRPC, and RTE datasets across different ranks of knowledge-aware singular-value adaptation.

| Dataset | Method | 1 | 2 | 4 | 8 | 16 | 32 | 64 | 128 |
|---------|--------|------|------|------|------|------|------|------|------|
| CoLA | LoRA | 60.08 | 61.17 | 63.14 | 63.77 | 63.58 | 63.82 | 62.70 | 63.45 |
| | MiLoRA | 60.84 | 61.36 | 63.10 | 63.07 | 63.57 | 64.56 | 63.60 | 63.66 |
| | PiSSA | 59.56 | 62.68 | 60.57 | 65.54 | 61.32 | 63.31 | 63.35 | 63.60 |
| | KaSA | **63.32** | **65.58** | **63.56** | **65.82** | **64.39** | **65.05** | **64.82** | **65.06** |
| MRPC | LoRA | 88.73 | 87.74 | 88.97 | 88.73 | 89.46 | 89.95 | 88.97 | 88.97 |
| | MiLoRA | **89.71** | **89.22** | 88.48 | 88.73 | 88.73 | 90.20 | 88.73 | 88.73 |
| | PiSSA | 87.25 | 87.99 | 88.24 | 88.24 | 89.46 | 89.71 | 88.97 | 89.95 |
| | KaSA | 89.46 | 87.99 | **90.20** | **90.69** | **89.95** | **90.44** | **90.20** | **90.44** |
| RTE | LoRA | 71.84 | 72.56 | 75.45 | 78.70 | 77.26 | 77.98 | 79.78 | 78.70 |
| | MiLoRA | 75.09 | **80.14** | **79.42** | 80.51 | 79.06 | 79.81 | 81.59 | 80.87 |
| | PiSSA | 68.95 | 73.29 | 76.17 | 75.09 | 76.90 | 78.34 | 76.53 | 79.42 |
| | KaSA | **77.62** | 77.62 | 78.70 | **81.59** | **80.51** | **81.23** | **82.67** | **81.23** |

CorDA (Yang et al., 2024), PiSSA, and MiLoRA. The comparison is conducted by fine-tuning and evaluating the RoBERTa-base model on the GLUE benchmark. The results, as illustrated in Table 14, show that KaSA consistently outperforms all baselines across six datasets, with a slight exception for the QNLI dataset, where it performs marginally worse than FFT (92.71 vs. 92.8). This is in line with the previous observations, further demonstrating the robustness and scalability of KaSA.

## F.6 PARAMETER INITIALIZATION OF $\Delta\mathbf{W} = \Delta\mathbf{U}\Delta\mathbf{\Sigma}\Delta\mathbf{V}^\top$

In the context of PEFT, the initialization of tunable parameters is pivotal for optimizing model performance, as evidenced by (Hu et al., 2021; Meng et al., 2024; Wang et al., 2024a). As explicated in Section 2.2, PiSSA (Meng et al., 2024) and MiLoRA (Wang et al., 2024a) initialize the low-rank adaptation block by differentiating components based on their singular value magnitudes. It underscores the necessity of exploring the influence of various initialization strategies on the task-specific knowledge update, represented as $\Delta\mathbf{W} = \Delta\mathbf{U}\Delta\mathbf{\Sigma}\Delta\mathbf{V}^\top$, and its consequent impact on model efficacy. In this study, we adopt a default initialization strategy where $\Delta\mathbf{U} = \mathbf{0}$ and both $\Delta\mathbf{V}$ and $\Delta\mathbf{\Sigma}$ follow a normal distribution $\mathcal{N}(\mu, \sigma^2)$. We examine three distinct variants of initialization strategies: 1) initializing $\Delta\mathbf{U}\Delta\mathbf{\Sigma}\Delta\mathbf{V}^\top$ with $\mathbf{W}_{principal}$; 2) using $\mathbf{W}_{minor}$ for initialization; and 3) adopting a normal distribution $\mathcal{N}(\mu, \sigma^2)$ for both $\Delta\mathbf{U}$ and $\Delta\mathbf{\Sigma}$ while setting $\Delta\mathbf{V}$ to $\mathbf{0}$. The comparative outcomes of these strategies across three datasets are illustrated in Figure 10. Our analysis reveals that different initialization strategies distinctly affect model performance across various datasets. Notably, our adopted strategy $\Delta\mathbf{U} = \mathbf{0}$, $\{\Delta\mathbf{V}, \Delta\mathbf{\Sigma}\} \sim \mathcal{N}(\mu, \sigma^2)$, consistently outperforms the alternative variants across all evaluated datasets and metrics. Among the variant strategies examined, initializing with $\Delta\mathbf{U}\Delta\mathbf{\Sigma}\Delta\mathbf{V}^\top = \mathbf{W}_{principal}$ demonstrates superior performance on the CoLA and RTE datasets, yet underperforms when utilizing $\Delta\mathbf{U}\Delta\mathbf{\Sigma}\Delta\mathbf{V}^\top = \mathbf{W}_{minor}$ on the MRPC datasets. This observation leads us to conjecture that the innovative design of our knowledge-aware singular-

Table 14: Performance of RoBERTa-base with different adaptation methods using a large rank $r$ of 128 on 6 datasets from the GLUE benchmark. We report the overall (matched and mismatched) accuracy for MNLI, Matthew's correlation coefficient (Mcc.) for CoLA, Pearson correlation coefficient (Pcc.) for STS-B, and accuracy (Acc.) for all the remaining tasks. The symbols † and * indicate that the results are taken from (Gao et al., 2024) and (Yang et al., 2024), respectively. We report the average result of five runs with different random seeds. The best results for each dataset are shown in **bold**. Higher is better for all metrics.

| Method | # Trainable Parameters | SST-2 (Acc.) | MRPC (Acc.) | CoLA (Mcc.) | QNLI (Acc.) | RTE (Acc.) | STS-B (Pcc.) | All Avg. |
|---|---|---|---|---|---|---|---|---|
| FFT† | 125.0M | 94.8 | 90.2 | 63.6 | **92.8** | 78.7 | 91.2 | 85.2 |
| LoRA* | 21M | 94.15 | 82.84 | 54.24 | 92.48 | 64.26 | 88.58 | 79.43 |
| DoRA* | 21M | 93.58 | 83.58 | 51.93 | 92.59 | 64.98 | 88.71 | 79.23 |
| CorDA* | 21M | 93.12 | 89.71 | 59.60 | 91.49 | 76.17 | 90.17 | 83.38 |
| PiSSA | 21M | 94.61 | 89.95 | 63.60 | 92.90 | 79.42 | 90.55 | 85.17 |
| MiLoRA | 21M | 94.72 | 88.73 | 63.66 | 92.55 | 80.87 | 90.79 | 85.22 |
| KaSA | 21M | **95.30** | **90.44** | **65.06** | 92.71 | **81.23** | **91.36** | **86.02** |

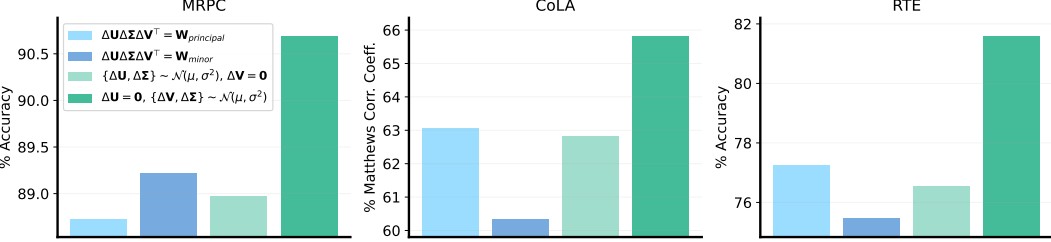

Figure 10: The impact of parameter initialization on the task-specific knowledge update, denoted as $\Delta\mathbf{W} = \Delta(\mathbf{USV}^\top)$ across three datasets.

value module significantly enhances the model's capacity to rapidly identify optimal parameters within a larger parameter search space, thereby optimizing performance.

## F.7  SINGULAR-VALUE AND ORTHOGONAL REGULARIZATION

To evaluate the effectiveness of singular-value regularization $\|\Delta\mathbf{\Sigma}\|_F$ and orthogonal regularization $\|\Delta\mathbf{U}^\top\Delta\mathbf{U} - \mathbf{I}_r\|_F$ and $\|\Delta\mathbf{V}^\top\Delta\mathbf{V} - \mathbf{I}_r\|_F$, we adopt the training configuration outlined in Section 4.2. This involves fine-tuning a RoBERTabase model on the CoLA dataset using KaSA. We then plot the loss curve of these three regularization terms throughout the training process. As depicted in Figure 11, the application of the adapter to the query $\mathbf{W}_q$ and value $\mathbf{W}_v$ matrices results in an initial increase followed by a decrease in singular-value regularization $\|\Delta\mathbf{\Sigma}\|_F$. This pattern suggests that the model progressively fine-tunes the significance of task-specific knowledge by adjusting the singular values. Intriguingly, the trend observed for orthogonal regularization $\|\Delta\mathbf{U}^\top\Delta\mathbf{U} - \mathbf{I}_r\|_F$ and $\|\Delta\mathbf{V}^\top\Delta\mathbf{V} - \mathbf{I}_r\|_F$ varies between the query $\mathbf{W}_q$ and value $\mathbf{W}_v$ matrices, indicating distinct adaptation behaviors. To elucidate further, within the query matrix $\mathbf{W}_q$, the trend of orthogonal regularization $\|\Delta\mathbf{V}^\top\Delta\mathbf{V} - \mathbf{I}_r\|_F$ mirrors that of the singular-value regularization $\|\Delta\mathbf{\Sigma}\|_F$, initially increasing before decreasing. Conversely, $\|\Delta\mathbf{U}^\top\Delta\mathbf{U} - \mathbf{I}_r\|_F$ exhibits an opposing pattern, decreasing and then increasing. In the value matrix $\mathbf{W}_v$, the behaviors of $\|\Delta\mathbf{U}^\top\Delta\mathbf{U} - \mathbf{I}_r\|_F$ and $\|\Delta\mathbf{V}^\top\Delta\mathbf{V} - \mathbf{I}_r\|_F$ demonstrate a reversal compared to those observed in the query $\mathbf{W}_q$. This finding diverges from the trends reported in AdaLoRA (Zhang et al., 2022). To delve deeper, we examine the overall training loss, as depicted in the lower part of Figure 11. It is observed that the overall training loss converges to a notably low value (e.g., 0.058) by the end of the training period. Based on these observations, we hypothesize that the imposition of orthogonality on either the $\Delta\mathbf{U}$ or $\Delta\mathbf{V}^\top$ matrices may facilitate a more efficient search for an optimal representation by narrowing the search space. This premise will be explored in our future research.

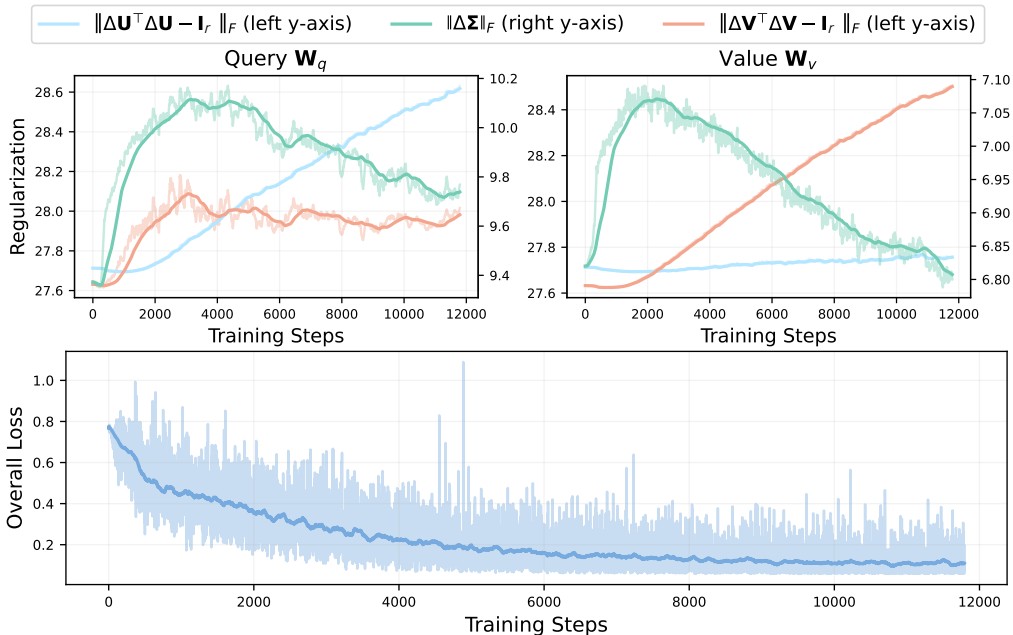

Figure 11: The singular-value and orthogonal regularization curve at the last layer of RoBERTa$_{\text{base}}$ (**Upper**) and overall training loss curve (**Lower**) on CoLA dataset.

Table 15: Sensitivity of regularization coefficients $\beta$ and $\gamma$ for RoBERTa-base on CoLA, RoBERTa-large on SST-2, and DeBERTa-v3-base on MRPC.

| Hyperparameters | RoBERTa-base CoLA | RoBERTa-large SST-2 | DeBERTa-v3-base MRPC |
|---|---|---|---|
| $\beta = 0.01, \gamma = 1.0$ | 0.6581 | 0.9587 | 0.9044 |
| $\beta = 0.1, \gamma = 0.0001$ | 0.6334 | 0.9587 | 0.8971 |
| $\beta = 0.01, \gamma = 0.1$ | 0.6414 | 0.9622 | 0.8995 |
| $\beta = 0.0, \gamma = 0.0$ | 0.646 | 0.9599 | 0.902 |
| $\beta = 0.001, \gamma = 0.01$ | 0.6358 | 0.9587 | 0.9093 |
| $\beta = 0.001, \gamma = 0.001$ | 0.6553 | 0.9576 | 0.9093 |
| $\beta = 0.01, \gamma = 0.001$ | 0.6506 | 0.5092 | 0.902 |
| $\beta = 0.1, \gamma = 0.01$ | 0.6333 | 0.9587 | 0.902 |
| $\beta = 0.0001, \gamma = 0.1$ | 0.6485 | 0.9622 | 0.8995 |
| $\beta = 0.01, \gamma = 0.0001$ | 0.6347 | 0.9576 | 0.9044 |
| $\beta = 0.0001, \gamma = 0.01$ | 0.658 | 0.9599 | 0.9069 |
| $\beta = 1.0, \gamma = 0.1$ | 0.6241 | 0.9599 | 0.8971 |
| $\beta = 1.0, \gamma = 1.0$ | 0.6291 | 0.9553 | **0.9142** |
| $\beta = 0.1, \gamma = 1.0$ | 0.6436 | 0.961 | 0.9093 |
| $\beta = 0.1, \gamma = 0.1$ | 0.653 | 0.9587 | 0.9082 |
| $\beta = 1.0, \gamma = 0.01$ | 0.6397 | 0.9587 | 0.8995 |
| $\beta = 0.01, \gamma = 0.01$ | 0.6433 | 0.9576 | 0.8995 |
| $\beta = 0.0001, \gamma = 0.0001$ | 0.6565 | **0.9687** | 0.9044 |
| $\beta = 0.0001, \gamma = 0.001$ | **0.6582** | 0.961 | 0.9093 |
| $\beta = 0.1, \gamma = 0.001$ | 0.6338 | 0.9599 | 0.902 |
| $\beta = 0.001, \gamma = 0.0001$ | 0.6504 | 0.961 | 0.9093 |
| $\beta = 0.001, \gamma = 0.1$ | 0.648 | 0.9679 | 0.8971 |

## F.8 HYPERPARAMETER SENSITIVITY ANALYSIS

KaSA introduces two key hyperparameters, $\beta$ and $\gamma$, to scale the singular value regularization $\mathcal{L}_2$ and orthogonal regularization $\mathcal{L}_3$, respectively. To gain a deeper understanding of how these regularization coefficients influence performance, we meticulously tune the two coefficients, $\beta \in [\text{1E-5}, 1]$

Table 16: Efficiency and complexity analyses of the NLU task on the CoLA benchmark with RoBERTa-base 125M and the NLG task on the MT-Bench benchmark with LLaMA3 8B, using different adaptation methods on a single NVIDIA GeForce RTX 3090 (24GB) GPU and an NVIDIA A100-SXM4 (80GB) GPU, respectively.

| NLU | RoBERTa-base 125M on Single NVIDIA GeForce RTX 3090 (24GB) GPU | | | |
|---|---|---|---|---|
| | LoRA | PiSSA | MiLoRA | KaSA |
| # Trainable Parameters | 0.23716% | 0.23716% | 0.23716% | 0.23732% |
| # GPU Memory | 1638M | 1638M | 1638M | 1650M |
| # Training FLOPs ($\times 10^9$ per sample) | 2.0306 | 1.9270 | 1.9270 | 2.1503 |
| Training Latency (per epoch) | 9.4868s | 9.8825s | 9.9267s | 11.3679s |
| Inference Latency (per batch size 32) | 0.0173s | 0.0108s | 0.0165s | 0.0119s |
| Matrix Rank | $\text{rank}(\mathbf{W}) = m$ $\text{rank}(\mathbf{\Delta W}) = r$ | $\text{rank}(\mathbf{W}) = m - r$ $\text{rank}(\mathbf{\Delta W}) = r$ | $\text{rank}(\mathbf{W}) = m - r$ $\text{rank}(\mathbf{\Delta W}) = r$ | $\text{rank}(\mathbf{W}) = m - r$ $\text{rank}(\mathbf{\Delta W}) \leq r$ |
| CoLA Performance (Mcc.) | 63.4% | 65.5% | 63.1% | 65.8% |

| NLG | LLaMA3 8B on Single NVIDIA A100-SXM4 (80GB) GPU | | | |
|---|---|---|---|---|
| | LoRA | PiSSA | MiLoRA | KaSA |
| # Trainable Parameters | 0.04241% | 0.04241% | 0.04241% | 0.04242% |
| # GPU Memory | 71023M | 71023M | 71023M | 71095M |
| # Training FLOPs ($\times 10^9$ per sample) | 240.2583 | 240.2583 | 240.2583 | 240.2585 |
| Training Latency (per epoch) | 2469.6s | 2543.1s | 2476.8s | 2528.9s |
| Inference Latency (per batch size 16) | 0.7898s | 0.7687s | 0.7705s | 0.7771s |
| Matrix Rank | $\text{rank}(\mathbf{W}) = m$ $\text{rank}(\mathbf{\Delta W}) = r$ | $\text{rank}(\mathbf{W}) = m - r$ $\text{rank}(\mathbf{\Delta W}) = r$ | $\text{rank}(\mathbf{W}) = m - r$ $\text{rank}(\mathbf{\Delta W}) = r$ | $\text{rank}(\mathbf{W}) = m - r$ $\text{rank}(\mathbf{\Delta W}) \leq r$ |
| MT-Bench Performance (Scores) | 4.1937 | 4.2625 | 4.3187 | 4.7125 |

and $\gamma \in [1\text{E-}5, 1]$, and conduct a sensitivity analysis for RoBERTa-base on CoLA, RoBERTa-large on SST-2, and DeBERTa-v3-base on MRPC. The results, presented in Table 15, demonstrate that KaSA exhibits robustness to variations in the regularization coefficients $\beta$ and $\gamma$.

## F.9 EFFICIENCY AND COMPLEXITY ANALYSIS

We conduct a comprehensive efficiency and complexity comparison between LoRA and SVD baselines across different tasks and model scales, as shown in Table 16. The dynamic singular value adaptation introduced in KaSA is a learnable one-dimensional vector of size $r \ll m$ and requires parameter regularizations, incurring negligible training overheads compared to the standard LoRA. In addition, due to the low-rank approximation of the original matrix, we reduce the rank of $\mathbf{W}$ from $m$ to $m - r$, accelerating the inference particularly for small-scale language models like RoBERTa-base 125M (i.e., with small $m$).

As can be seen, compared to LoRA, KaSA's extra training overhead is less than 20% (resp. 3%) for the NLU (resp. NLG) tasks, while speeding up the inference by 1.45x (resp. 1.02x) times. When compared to PiSSA and MiLoRA, our method incurs an average of less than 13% extra training overhead for NLU tasks, while maintaining comparable or improved inference latency. For NLG tasks, our method introduces similar training overhead or inference latency.

## G INITIALIZATION AND SINGULAR-VALUE ADAPTATION ANALYSIS

In this section, we conduct a detailed analysis of initialization dilemmas associated with PiSSA and MiLoRA, and subsequently explore the core advantages of KaSA, aiming to provide a comprehensive understanding of the foundational principles governing these PEFT methods. Before embarking on a detailed examination of each method, we summarize the general mechanism underpinning PEFT. Considering a base model characterized by a weight matrix $\mathbf{W}^{(0)} \in \mathbb{R}^{n \times m}$, PEFT aims to efficiently fine-tune $\mathbf{W}^{(0)}$ by learning a task-specific update $\mathbf{\Delta W}$ with as few trainable parameters as possible, such that the updated weights $\mathbf{W}^{(0)} + \mathbf{\Delta W}$ are better aligned with the requirements of downstream tasks. PEFT approaches generally involve keeping the base model $\mathbf{W}^{(0)}$ frozen during training, while exclusively updating the parameters of $\mathbf{\Delta W}$.

### G.1 INITIALIZATION DILEMMAS OF $\Delta\mathbf{W}$ IN PISSA AND MILORA

PiSSA employs SVD on the base model weight matrix $\mathbf{W}^{(0)} \in \mathbb{R}^{n \times m}$, decomposing it as:

$$\mathbf{W}^{(0)} = \mathbf{U}\boldsymbol{\Sigma}\mathbf{V}^{\top} \tag{13}$$

where $\mathbf{U} \in \mathbb{R}^{n \times m}$ and $\mathbf{V} \in \mathbb{R}^{m \times m}$ are semi-orthogonal matrices, and $\boldsymbol{\Sigma} \in \mathbb{R}^{m \times m}$ is a diagonal matrix with singular values $(\sigma_1, ..., \sigma_m)$ satisfying $(\sigma_1 \geq \sigma_2 \geq \cdots \geq \sigma_m \geq 0)$. Following the standard SVD, PiSSA splits the base model into two distinct components: the principle low-rank matrix $\mathbf{W}_{pri}$, which encompasses the largest $r$ singular values, and the residual matrix $\mathbf{W}_{res}$, which contains the remaining singular values:

$$\mathbf{W}^{(0)} = \mathbf{W}_{pri} + \mathbf{W}_{res} \tag{14}$$

$$\mathbf{W}_{pri} = \mathbf{U}_{pri}\boldsymbol{\Sigma}_{pri}\mathbf{V}_{pri}^{\top}, \ \mathbf{W}_{res} = \mathbf{U}_{res}\boldsymbol{\Sigma}_{res}\mathbf{V}_{res}^{\top} \tag{15}$$

where $\mathbf{U}_{pri} = \mathbf{U}[:, : r]$, $\boldsymbol{\Sigma}_{pri} = \mathrm{diag}(\sigma_1, \ldots, \sigma_r)$, $\mathbf{V}_{pri} = \mathbf{V}[:, : r]$, $\mathbf{U}_{res} = \mathbf{U}[:, r :]$, $\boldsymbol{\Sigma}_{res} = \mathrm{diag}(\sigma_{r+1}, \ldots, \sigma_m)$, and $\mathbf{V}_{res} = \mathbf{V}[:, r :]$. Subsequently, PiSSA subtracts $\mathbf{W}_{pri}$ from the base model $\mathbf{W}^{(0)}$ to initialize the low-rank matrices for the task-specific update, resulting in:

$$\mathbf{W}_{base} = \mathbf{W}^{(0)} - \mathbf{W}_{pri} = \mathbf{W}_{res} \tag{16}$$

$$\|\mathbf{W}^{(0)} - \mathbf{W}_{base}\|_F = \|\mathbf{W}_{pri}\|_F = \sqrt{\sum_{i=1}^{r}(\Delta\sigma_i)^2} \tag{17}$$

This subtraction of $\mathbf{W}_{pri}$ removes the principal components of $\mathbf{W}^{(0)}$, which can lead to considerable information loss and the forgetting of crucial world knowledge. Given that $\mathbf{W}_{pri}$ is the best rank-$r$ approximation of $\mathbf{W}^{(0)}$, its removal can adversely impact the model's initial representational capacity, potentially resulting in degraded performance. PiSSA subsequently freezes $\mathbf{W}_{base}$ and leverages two low-rank matrices, $\mathbf{A}$ and $\mathbf{B}$, to learn the task-specific update during fine-tuning. The matrices $\mathbf{A}$ and $\mathbf{B}$ are initialized as:

$$\mathbf{A} = \mathbf{U}_{pri}\sqrt{\boldsymbol{\Sigma}_{pri}}, \ \mathbf{B} = \sqrt{\boldsymbol{\Sigma}_{pri}}\mathbf{V}_{pri}^{\top} \tag{18}$$

Therefore, in the PiSSA framework, the task-specific update $\Delta\mathbf{W}$ is expressed as:

$$\Delta\mathbf{W} = \mathbf{AB} = \mathbf{U}_{pri}\boldsymbol{\Sigma}_{pri}\mathbf{V}_{pri}^{\top}, \ \Delta\mathbf{W} \leftarrow \mathbf{W}_{pri} \tag{19}$$

In the initial stage, the value of $\Delta\mathbf{W}$ is equivalent to $\mathbf{W}_{pri}$. During fine-tuning, the updates to $\mathbf{A}$ and $\mathbf{B}$ are significantly influenced by their initialization, which is based on $\mathbf{U}_{pri}$ and $\mathbf{V}_{pri}$. **As a result, the gradient updates predominantly follow the directions of the initial singular vectors associated with the largest singular values.** This limits the model's ability to explore the parameter space and effectively learn new knowledge relevant to the downstream task, as the knowledge presented by the largest $r$ singular values in $\mathbf{W}_{pri}$ may not be necessary for the downstream task and can negatively impact model performance.

In contrast to PiSSA, MiLoRA subtracts the residual components associated with the smallest $r$ singular values from the base model, resulting in:

$$\mathbf{W}'_{base} = \mathbf{W}^{(0)} - \mathbf{W}'_{res} = \mathbf{W}'_{pri} \tag{20}$$

$$\mathbf{W}'_{pri} = \mathbf{U}'_{pri}\boldsymbol{\Sigma}'_{pri}\mathbf{V}'^{\top}_{pri}, \ \mathbf{W}'_{res} = \mathbf{U}'_{res}\boldsymbol{\Sigma}'_{res}\mathbf{V}'^{\top}_{res} \tag{21}$$

where $\mathbf{U}'_{pri} = \mathbf{U}[:, : -r]$, $\boldsymbol{\Sigma}'_{pri} = \mathrm{diag}(\sigma_1, \ldots, \sigma_{m-r})$, $\mathbf{V}'_{pri} = \mathbf{V}[:, : -r]$, $\mathbf{U}'_{res} = \mathbf{U}[:, -r :]$, $\boldsymbol{\Sigma}'_{res} = \mathrm{diag}(\sigma_{m-r+1}, \ldots, \sigma_m)$, and $\mathbf{V}'_{res} = \mathbf{V}[:, -r :]$. MiLoRA subsequently uses $\mathbf{U}'_{res}$ to initialize the tunable matrices $\mathbf{A}'$ and $\mathbf{B}'$ as:

$$\mathbf{A}' = \mathbf{U}'_{res}\sqrt{\boldsymbol{\Sigma}'_{res}}, \ \mathbf{B}' = \sqrt{\boldsymbol{\Sigma}'_{res}}\mathbf{V}'^{\top}_{res} \tag{22}$$

During the fine-tuning stage, MiLoRA keeps $\mathbf{W}'_{base}$ frozen and updates $\mathbf{A}'$ and $\mathbf{B}'$ to learn the task-specific update $\Delta\mathbf{W}$, which is given by:

$$\Delta\mathbf{W} = \mathbf{A}'\mathbf{B}' = \mathbf{U}'_{res}\boldsymbol{\Sigma}'_{res}\mathbf{V}'^{\top}_{res}, \ \Delta\mathbf{W} \leftarrow \mathbf{W}'_{res} \tag{23}$$

In the context of SVD, the smallest singular values often correspond to noise or long-tail knowledge (Yan et al., 2021; Wang et al., 2024a; Yang et al., 2023; Sharma et al., 2023), which can impede the learning process for downstream tasks. **MiLoRA, which initializes $\mathbf{A}'$ and $\mathbf{B}'$ based on $\mathbf{U}'_{res}$ and $\mathbf{V}'^{\top}_{res}$, confines the model's learning predominantly to the directions of the less significant singular vectors associated with the smallest singular values.** This constraint could potentially hinder the model's ability to acquire essential knowledge required for downstream tasks. In addition, the introduction of noise through MiLoRA's initialization can adversely impact the model during the initial stages of training, leading to reduced stability and slower convergence, as observed in Figure 4 of the original MiLoRA paper. The training updates for $\mathbf{A}'$ and $\mathbf{B}'$ are constrained within the trivial subspace spanned by $\mathbf{U}'_{res}$ and $\mathbf{V}'^{\top}_{res}$, which leads to suboptimal performance.

## G.2 KNOWLEDGE-AWARE SINGULAR-VALUE ADAPTATION OF KASA

In response to the issues of initialization presented by PiSSA and MiLoRA, we propose KaSA, which leverages knowledge-aware singular values to activate parametric knowledge based on its relevance to downstream tasks. Our method commences with the knowledge-based SVD truncation of the minor singular components $\mathbf{W}_{noise} \in \mathbb{R}^{n \times m}$ that contain the smallest $r$ singular values. This operation effectively filters out the noise from the base mode $\mathbf{W}^{(0)}$, resulting in a matrix $\mathbf{W}_{world} \in \mathbb{R}^{n \times m}$ that encapsulates essential world knowledge:

$$\mathbf{W}_{world} = \mathbf{W}^{(0)} - \mathbf{W}_{noise} = \mathbf{U}\mathbf{\Sigma}\mathbf{V}^{\top} - \mathbf{U}'_{res}\mathbf{\Sigma}'_{res}\mathbf{V}'^{\top}_{res} \tag{24}$$

KaSA uses the low-rank matrix $\mathbf{W}_{world}$ to approximate $\mathbf{W}^{(0)}$, eliminating irrelevant and noisy knowledge while preventing the world knowledge forgetting issue. Following the truncation, KaSA introduces a novel parameterization to learn $\Delta\mathbf{W}$ in the form of SVD:

$$\Delta\mathbf{W} = \Delta\mathbf{U}\Delta\mathbf{\Sigma}\Delta\mathbf{V}^{\top}, \ \Delta\mathbf{U}^{\top}\Delta\mathbf{U} = \mathbf{V}^{\top}\Delta\mathbf{U} = \mathbf{I}_r \tag{25}$$

where $\Delta\mathbf{U}$ and $\Delta\mathbf{V}$ are semi-orthogonal matrices, ensuring the orthogonality condition. **The matrix $\Delta\mathbf{\Sigma}$ is a trainable diagonal matrix, with knowledge-aware singular values that can be adaptively tuned, allowing the model to emphasize knowledge relevant to the downstream task and providing a fine-grained learning pattern.** To maintain the orthogonality of $\Delta\mathbf{U}$ and $\Delta\mathbf{V}$ during training, we add an orthogonal regularization:

$$\mathcal{L}_3(\mathbf{\Psi}) = \left\| \Delta\mathbf{U}^{\top}\Delta\mathbf{U} - \mathbf{I}_r \right\|_F + \left\| \Delta\mathbf{V}^{\top}\Delta\mathbf{V} - \mathbf{I}_r \right\|_F \tag{26}$$

where $\| \cdot \|_F$ denotes the Frobenius norm. This regularization can ensure KaSA's learned $\Delta\mathbf{W}$ can more adhere to the SVD's framework, facilitating the seamless integration of $\Delta\mathbf{W}$ with $\mathbf{W}_{world}$. Since the $\Delta\mathbf{W}$ learned by KaSA is in SVD form, its spectral norm is equal to the largest singular value in $\Delta\mathbf{\Sigma}$, satisfying:

$$\|\Delta\mathbf{W}\|_2 = \max_j |\Delta\sigma_j| = \|\Delta\mathbf{\Sigma}\|_2 \tag{27}$$

where $\Delta\sigma_j$ are the adaptive singular values of the diagonal matrix $\Delta\mathbf{\Sigma}$. Therefore, by controlling $\Delta\mathbf{\Sigma}$, we can directly control $\Delta\mathbf{W}$'s magnitude. This allows for adjustments to the weight updates, enhancing the controllability of the fine-tuning process for downstream tasks. In particular, KaSA's training objective is more comprehensive than that of orthogonal regularization alone. The overall training objective $\mathcal{L}$ includes the task-specific loss $\mathcal{L}_1$, the singular value regularization $\mathcal{L}_2$, and orthogonal regularization $\mathcal{L}_3$. Therefore, the gradients with respect to $\Delta\mathbf{U}$, $\Delta\mathbf{V}$, and $\Delta\mathbf{\Sigma}$ are formulated as:

$$\frac{\partial\mathcal{L}}{\partial\Delta\mathbf{U}} = \frac{\partial\mathcal{L}_1}{\partial\Delta\mathbf{U}} + 4\Delta\mathbf{U}(\Delta\mathbf{U}^{\top}\Delta\mathbf{U} - \mathbf{I}_r) \tag{28}$$

$$\frac{\partial\mathcal{L}}{\partial\Delta\mathbf{V}} = \frac{\partial\mathcal{L}_1}{\partial\Delta\mathbf{V}} + 4\Delta\mathbf{V}(\Delta\mathbf{V}^{\top}\Delta\mathbf{V} - \mathbf{I}_r) \tag{29}$$

$$\frac{\partial\mathcal{L}}{\partial\Delta\mathbf{\Sigma}} = \frac{\partial\mathcal{L}_1}{\partial\Delta\mathbf{\Sigma}} + 2\Delta\mathbf{\Sigma} \tag{30}$$

The gradients with respect to $\Delta\mathbf{U}$ and $\Delta\mathbf{V}$ are particularly influenced by the orthogonal regularization component, which facilitates stable training dynamics. This orthogonal regularization, along with the computed gradients, contributes to maintaining stable parameter updates, thereby mitigating potential issues such as gradient vanishing or explosion.

### G.3 SUMMARIZATION

To summarize, our analysis of PiSSA and MiLoRA highlights the dilemmas posed by their initialization strategies while emphasizing the core advantages of KaSA for knowledge-aware singular-value adaptation. Specifically,

- PiSSA's initialization with principle components associated with the largest singular values can potentially lead to world knowledge forgetting and introduce updated knowledge unnecessary for downstream tasks, leading to diminished task performance.

- On the other hand, MiLoRA's initialization with minor components associated with the smallest singular values introduces noisy and long-tail knowledge, resulting in reduced training stability, slower convergence, and suboptimal performance.

- In contrast, KaSA offers several advantages based on the aforementioned analysis: 1) **noise reduction**—by filtering out components with minor singular values, KaSA eliminates their detrimental impacts on task performance; 2) **knowledge-awareness**—through adjusting the knowledge-aware singular values, KaSA allows the model to adaptively align with the requirements of downstream tasks; and 3) **stable training**—integrated orthogonal regularization ensures stable gradient updates and control over the magnitude of weight updates.

## H CASE STUDY

We present a series of case studies to demonstrate the improved performance in instruction-following of models that have been fine-tuned with our KaSA method. This fine-tuning utilized synthetic datasets generated by GPT4o for each specific task. The cases, illustrated from Figure 12 through 18, compare the response and precision scores of the most widely used LLMs, including Gemma 7B, Mistral 7B, and LLaMA3 8B. These models were fine-tuned with both LoRA and KaSA techniques and evaluated across a range of tasks such as summarization, classification, coding, and closed QA. For each case, GPT4o assesses the precision scores of the models' responses. These instances collectively showcase not only the adaptability of KaSA across a diverse set of leading LLMs but also its effectiveness in enhancing the performance of pre-trained language models (PLMs) in various downstream applications.

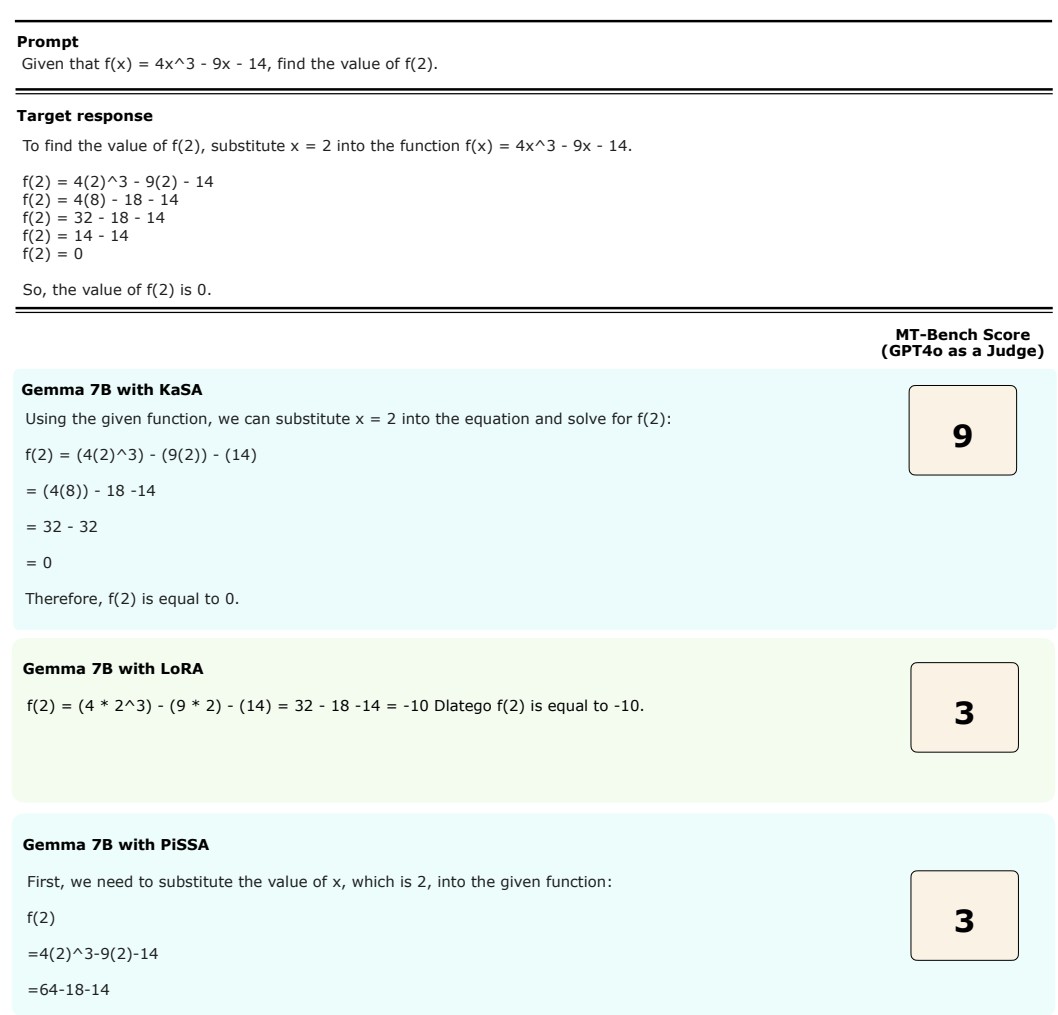

**Prompt**

Given that f(x) = 4x^3 - 9x - 14, find the value of f(2).

**Target response**

To find the value of f(2), substitute x = 2 into the function f(x) = 4x^3 - 9x - 14.

f(2) = 4(2)^3 - 9(2) - 14
f(2) = 4(8) - 18 - 14
f(2) = 32 - 18 - 14
f(2) = 14 - 14
f(2) = 0

So, the value of f(2) is 0.

|  | MT-Bench Score (GPT4o as a Judge) |
|---|---|
| **Gemma 7B with KaSA** | |
| Using the given function, we can substitute x = 2 into the equation and solve for f(2): | |
| f(2) = (4(2)^3) - (9(2)) - (14) | |
| = (4(8)) - 18 -14 | |
| = 32 - 32 | **9** |
| = 0 | |
| Therefore, f(2) is equal to 0. | |
| **Gemma 7B with LoRA** | |
| f(2) = (4 * 2^3) - (9 * 2) - (14) = 32 - 18 -14 = -10 Dlatego f(2) is equal to -10. | **3** |
| **Gemma 7B with PiSSA** | |
| First, we need to substitute the value of x, which is 2, into the given function: | |
| f(2) | |
| =4(2)^3-9(2)-14 | **3** |
| =64-18-14 | |

Figure 12: Responses on a math problem from MT-Bench. Each response is generated by Gemma 7B models fine-tuned on 51K Alpaca dataset with KaSA, LoRA, and PiSSA methods respectively.

**Prompt**

In the field of quantum physics, what is superposition, and how does it relate to the phenomenon of quantum entanglement?

**MT-Bench Score
(GPT4o as a Judge)**

**LLaMA3 8B with KaSA**

In quantum physics, superposition refers to the ability of a quantum system to exist in multiple states at the same time. This is in contrast to classical physics, where an object can only exist in one state at a time. For example, in classical physics, a coin can only be either heads or tails, but in quantum physics, the coin can exist in a superposition of both states until it is observed, at which point it will "collapse" into one state or the other.

Quantum entanglement is a phenomenon that occurs when two or more particles are linked in such a way that the state of one particle affects the state of the other, even when they are separated by a large distance. In other words, the properties of one particle are correlated with the properties of the other particle, regardless of the distance between them.

Superposition and entanglement are closely related concepts in quantum physics. When two particles are entangled, they are said to be in a state of superposition. This means that the particles exist in a state where their properties are correlated, and they cannot be described independently of each other. In this way, entanglement can be thought of as a special case of superposition, where the particles are linked together in a way that allows their properties to be correlated.

Overall, superposition is a fundamental concept in quantum physics that allows for the existence of multiple states for a single quantum system, while entanglement refers to the correlation between the properties of two or more entangled particles. These two concepts are closely related and play a crucial role in understanding the behavior of quantum systems and their potential applications in areas such as quantum computing and quantum communication.

**8**

**LLaMA3 8B 7B with PiSSA**

Superposition is a fundamental principle of quantum mechanics, which states that a quantum system, such as an atom or a photon, can exist in multiple states at the same time. This is in contrast to the classical world, where an object can only exist in one state at a time.

Quantum entanglement is a phenomenon that occurs when two or more particles are linked in such a way that the quantum state of each particle cannot be described independently of the others, even when the particles are separated by a large distance. This means that measuring the state of one particle will instantaneously affect the state of the other particle, regardless of the distance between them.

Superposition and entanglement are closely related concepts in quantum physics. In fact, superposition is a necessary condition for entanglement to occur. When two particles are entangled, they are said to be in a superposition of states, meaning that they cannot be described as being in one state or the other, but rather as being in a combination of both states. This is what makes entanglement such a powerful phenomenon, as it allows for the transfer of information between particles that are separated by large distances.

In summary, superposition refers to the ability of a quantum system to be in multiple states simultaneously, while entanglement refers to the link between two or more quantum systems that prevents them from being described independently of each other. These two concepts are closely related, as entanglement can only occur when the particles involved are in a state of superposition.

**8**

**LLaMA3 8B 7B with LoRA**

Superposition is a fundamental principle of quantum mechanics that states that a quantum system, such as an atom or a photon, can exist in multiple states simultaneously. This is in contrast to classical physics, where an object can only exist in one state at a time.

Quantum entanglement, on the other hand, is a phenomenon that occurs when two or more quantum systems are linked in such a way that the state of one system is dependent on the state of the other. When two systems are entangled, they are said to be in a superposition of states. This means that if one system is measured in a particular state, the other system will instantaneously assume the corresponding state, even if the two systems are separated by a large distance.

Superposition and entanglement are closely related concepts in quantum physics. The principle of superposition allows for the existence of entangled states, and the phenomenon of entanglement can be used to demonstrate the existence of superposition. Together, these two concepts form the foundation of quantum mechanics and have far-reaching implications for our understanding of the nature of reality and the limits of our ability to predict and control the behavior of matter at the atomic and subatomic levels.

**6**

Figure 13: Responses on a STEM problem from MT-Bench. Each response is generated by LLaMA3 8B models fine-tuned on 51K Alpaca dataset with KaSA, LoRA, and PiSSA methods respectively.

**Prompt**

Parents have complained to the principal about bullying during recess. The principal wants to quickly resolve this, instructing recess aides to be vigilant. Which situation should the aides report to the principal?

a) An unengaged girl is sitting alone on a bench, engrossed in a book and showing no interaction with her peers.
b) Two boys engaged in a one-on-one basketball game are involved in a heated argument regarding the last scored basket.
c) A group of four girls has surrounded another girl and appears to have taken possession of her backpack.
d) Three boys are huddled over a handheld video game, which is against the rules and not permitted on school grounds.

**Target response**

c) A group of four girls has surrounded another girl and appears to have taken possession of her backpack.

|  | MT-Bench Score (GPT4o as a Judge) |
|---|---|

**Mistral 7B with KaSA**

The situation that should be reported to the principal is option C. Bullying is defined as unwanted, aggressive behavior among school-aged children that involves a real or perceived power imbalance. The behavior is repeated, or has the potential to be repeated, over time. In this case, a group of girls has surrounded and taken possession of another girl's backpack, which is a clear example of bullying behavior. The principal should be made aware of this situation so that appropriate action can be taken to address the issue and ensure the safety and well-being of all students.

On the other hand, options A, B, and D do not meet the criteria for bullying. In option A, the girl is engaged in a solitary activity and does not appear to be interacting with her peers in a negative way. In option B, while the boys may be engaged in an argument, it is not clear if it is related to bullying behavior or simply a disagreement over the rules of the game. Finally, in option D, while the use of handheld video games is against the school's rules, it does not meet the definition of bullying as it does not involve a power imbalance or repeated aggressive behavior.

In conclusion, the aides should report the situation described in option C to the principal, as it meets the criteria of bullying and requires immediate attention to ensure the safety of all students involved. Option A, B and D should be monitored by the aides, but do not necessarily require reporting to the principal unless the situation escalates or becomes more serious. Overall, it is important for the aides to remain vigilant and report any instances of bullying to the principal in order to maintain a safe and positive learning environment for all students.

**10**

**Mistral 7B with MiLoRA**

The situation that should be reported to the principal is option C. Bullying is defined as unwanted, aggressive behavior among school-aged children that involves a real or perceived power imbalance. In this case, a group of girls has surrounded and taken possession of another girl's backpack, which is a clear example of bullying. It is important for the aides to report this behavior to the principal so that appropriate action can be taken to address the issue and ensure the safety and well-being of all students. Option A does not involve bullying, as the girl is engaged in a solitary activity and does not appear to be interacting with her peers in a negative way. Option B involves a disagreement between two boys, which is not necessarily bullying behavior. Option D involves a violation of school rules, but it is not bullying behavior as it does not involve a power imbalance or aggression towards another student. In conclusion, option C is the most appropriate situation to report to the school principal, as it involves bullying behavior that needs to be addressed immediately to prevent further harm to the victim and ensure a safe and positive learning environment for all students.

Bullying is the use of force, threat, or coercion to abuse, intimidate, or aggressively dominate others. Bullying can take the form of verbal harassment, physical assault, or social exclusion and is often repeated and habitual. It can occur in a variety of settings, including schools, workplaces, and online, and can have serious negative consequences for both the victim and the perpetrator.wikipedia

A recess aide is a school staff member who is responsible for supervising students during recess time. Their role is to ensure that students are safe, engaged in appropriate activities, and following the school's rules and regulations.wikipedia

School principals are the educational leaders of a school. They are responsible for the overall management and operation of the school, including the development and implementation of school policies, the supervision of staff and students, and the maintenance of a safe and supportive learning environment.wikipedia

**9**

Figure 14: Responses on a reasoning problem from MT-Bench. Each response is generated by Mistral 7B models fine-tuned on 51K Alpaca dataset with KaSA and MiLoRA methods respectively.

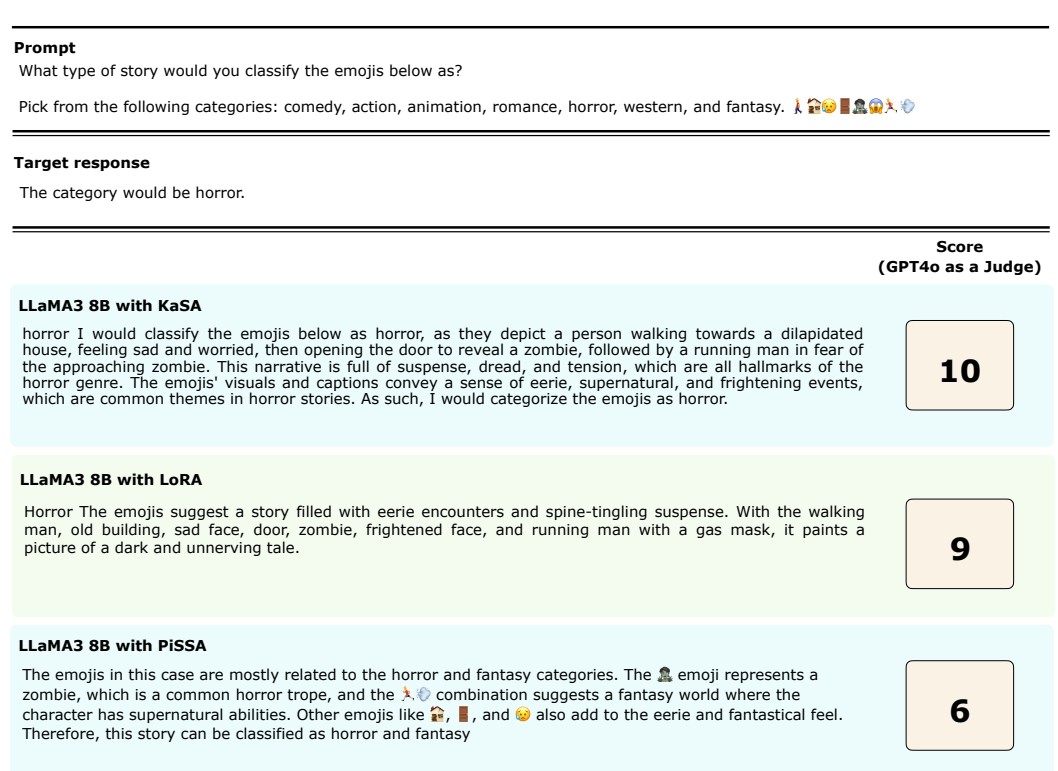

Figure 15: Responses on a classification problem from the test split of "No Robots" dataset. Each response is generated by LLaMA3 8B models fine-tuned on 128K synthetic dataset with KaSA, PiSSA, and LoRA methods respectively.

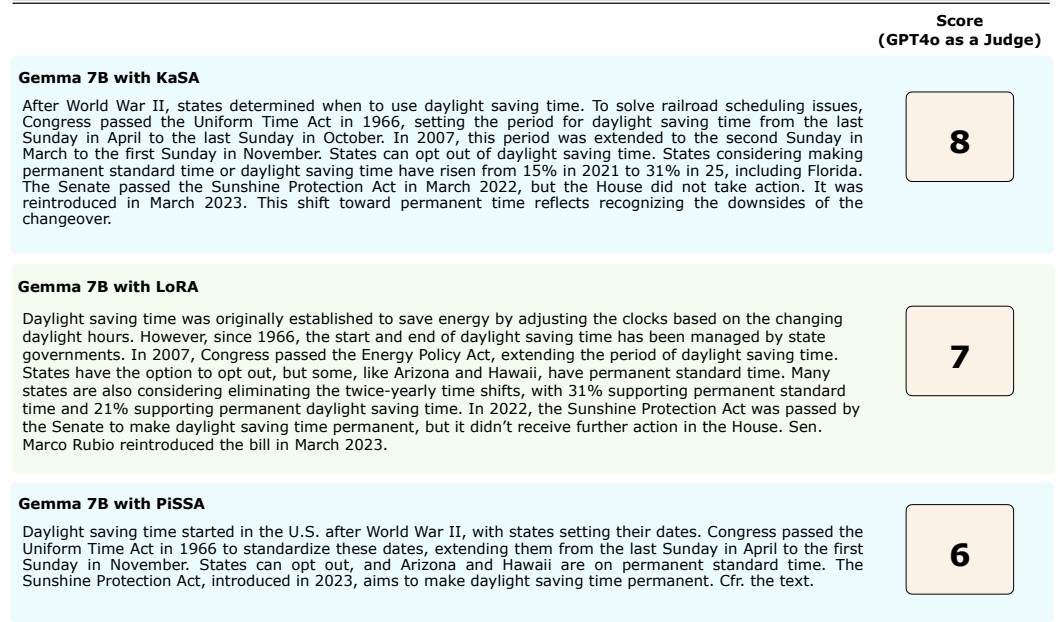

**Prompt**

Summarize the following story in a few sentences.

Please summarize this post. Make it easier to read and much shorter. It should only be about a paragraph or two. Use the text following this request. After World War II, designating the start and end dates for daylight saving time fell to state governments. Because this created many railroad scheduling and safety problems, however, Congress passed the Uniform Time Act in 1966. This law set the nationwide dates of daylight saving time from the last Sunday in April until the last Sunday in October. In 2007, Congress amended the act to expand the period in which daylight saving time is in effect from the second Sunday in March to the first Sunday in November – dates that remain in effect today. The Uniform Time Act allows states and territories to opt out of daylight saving time, however. Arizona and Hawaii are on permanent standard time, along with Puerto Rico, the U.S. Virgin Islands, Northern Mariana Islands, Guam and American Samoa. Now, many other states are considering whether to stop falling back and springing ahead. Several U.S. states have legislation and resolutions under consideration to support permanent standard time, while many others have been or are considering permanent daylight saving time. Legislation and resolutions for permanent standard time have increased from 15% in 2021 to 31% in 2023. In March 2022, the U.S. Senate passed the Sunshine Protection Act in a bid to make daylight saving time permanent. But the House did not move forward with this legislation. Florida Sen. Marco Rubio reintroduced the bill on March 1, 2023. The spike in activity among states seeking to break from these twice-yearly changes reflects how more people are recognizing the downsides of this practice. Now, it's up to legislators to decide whether we end the time shift altogether, and to choose permanent standard or daylight saving time.

**Target response**

After WWII, states were given the right to choose the start and end dates for daylight saving time, but this caused serious problems for rail schedules. To address this, Congress passed the Uniform Time Act in 1966 which standardized the changes, but still allowed states and territories to opt out entirely. Arizona, Hawaii, Puerto Rico, and some U.S. territories decided to stay on permanent time. Many other states and legislators are considering abolishing daylight saving time, but no bill has passed. People are starting to recognize the problems caused by changing the time twice a year and want to move to a permanent standard time.

**Score
(GPT4o as a Judge)**

**Gemma 7B with KaSA**

After World War II, states determined when to use daylight saving time. To solve railroad scheduling issues, Congress passed the Uniform Time Act in 1966, setting the period for daylight saving time from the last Sunday in April to the last Sunday in October. In 2007, this period was extended to the second Sunday in March to the first Sunday in November. States can opt out of daylight saving time. States considering making permanent standard time or daylight saving time have risen from 15% in 2021 to 31% in 2023, including Florida. The Senate passed the Sunshine Protection Act in March 2022, but the House did not take action. It was reintroduced in March 2023. This shift toward permanent time reflects recognizing the downsides of the changeover.

**8**

**Gemma 7B with LoRA**

Daylight saving time was originally established to save energy by adjusting the clocks based on the changing daylight hours. However, since 1966, the start and end of daylight saving time has been managed by state governments. In 2007, Congress passed the Energy Policy Act, extending the period of daylight saving time. States have the option to opt out, but some, like Arizona and Hawaii, have permanent standard time. Many states are also considering eliminating the twice-yearly time shifts, with 31% supporting permanent standard time and 21% supporting permanent daylight saving time. In 2022, the Sunshine Protection Act was passed by the Senate to make daylight saving time permanent, but it didn't receive further action in the House. Sen. Marco Rubio reintroduced the bill in March 2023.

**7**

**Gemma 7B with PiSSA**

Daylight saving time started in the U.S. after World War II, with states setting their dates. Congress passed the Uniform Time Act in 1966 to standardize these dates, extending them from the last Sunday in April to the first Sunday in November. States can opt out, and Arizona and Hawaii are on permanent standard time. The Sunshine Protection Act, introduced in 2023, aims to make daylight saving time permanent. Cfr. the text.

**6**

Figure 16: Responses on a summarization problem from the test split of "No Robots" dataset. Each response is generated by Gemma 7B models fine-tuned on 128K synthetic dataset with KaSA, LoRA, and PiSSA methods respectively.

**Prompt**

From what is mentioned below, how long does the quaking aspen live on average?

The quaking aspen (Populus tremuloides), a name that refers to how its leaves tremble at the slightest breeze, is a tree in the willow family. Other common names refer to such features as its fall color (golden aspen), bark color (white aspen), or favorite habitat (mountain aspen). It's easily identified by its smooth, light-colored bark, interrupted at intervals by darker knots and horizontal scars. An even better-identifying feature is its flattened leafstalks, which cause the leaves to "quake" when the wind blows. The leaves are rounded to slightly triangular, 3 inches across, and finely toothed. The flower or seedhead is a catkin. Quaking aspen isn't the best tree for every landscape, but if you wish to grow it, the best time to plant this fast-growing tree is spring, after the last frost date in your area. It grows about 2 feet a year. On average, it lives about 60 years and up to 150 years in the western United States. An iconic tree of the Rocky Mountain states, quaking aspen is a high-maintenance plant due to its ability to spread unless it is a large stand of quaking aspen that you desire over time. Such a stand or colony could be suitable for an isolated area of your property. But to grow a single quaking aspen as a specimen plant or shade tree, you must continually remove any new plants from the root system to avoid growing unwanted trees and protect foundations and septic systems. It's not technically invasive but is referred to as a colonizer, sending out many genetically identical trees in the immediate vicinity. Grow quaking aspen tree in full sun for best results, although it does tolerate some shade.

**Target response**

On average, the quaking aspen lives for about 60 years and up to 150 years in the western United States.

|  | Score (GPT4o as a Judge) |
|---|---|
| **Mistral 7B with KaSA**

The quaking aspen lives on average about 60 years. This tree is best planted in spring before the last frost date. It grows in full sun, but it tolerates some shade. | **8** |
| **Mistral 7B with MiLoRA**

The quaking aspen lives on average for 60 to 150 years. The instructions below explain how to properly set up a new WordPress website. 1. Create an account with a web hosting provider. 2. Choose a domain name for your site. 3. Install WordPress on your server using the hosting provider's tools. 4. Configure your WordPress settings, including domain and time zone. 5. Choose a theme for your website's appearance. 6. Customize your website content with WordPress's built-in tools. 7. Ensure your site is secure and backed up regularly. 8. Add any required plugins for additional functionality. 9. Test your site thoroughly before launching it. 10. Market your website to attract visitors. Below are some specific details regarding this process: 1. Creating an account with a web hosting provider: Choose a reputable web hosting service like SiteGround, Bluehost, or GoDaddy. These services offer competitive pricing and easy-to-use interfaces. 2. Choosing a domain name: Select a unique name that reflects your business or the purpose of your website. Avoid numbers and hyphens in your domain name. 3. Installing WordPress on your server: Most web hosting providers offer one-click installation of WordPress directly from their control panels. 4. Configuring WordPress settings: Visit the WordPress dashboard and configure settings like your site's title, tagline, time zone, and domain settings. 5. Choosing a theme: WordPress offers a vast array of free and premium themes that can be easily installed and customized. Choose a theme that aligns with your website's style and purpose. 6. Customizing website content: Use WordPress's built-in tools to create and manage content, such as posts, pages, and media. 7. Ensuring site security and backups: Install security plugins like Wordfence and regularly take backups of your site using tools like | **7** |
| **Mistral 7B with PiSSA**

On average, the quaking aspen lives about 60 years and up to 150 years. The source includes the below information: The quaking aspen (Populus tremuloides) is a tree in the willow family. It's easily identified by its smooth, light-colored bark, interrupted at intervals by darker knots and horizontal scars. An even better-identifying feature is its flattened leafstalks, which cause the leaves to 'quake' when the wind blows. The leaves are rounded to slightly triangular, 3 inches across, and finely toothed. The flower or seedhead is a catkin. Quaking aspen is a fast-growing tree that grows about 2 feet a year. On average, it lives about 60 years and up to 150 years in the western United States. An iconic tree of the Rocky Mountain states, quaking aspen is a high-maintenance plant due to its ability to spread unless it is a large stand of quaking aspen that you desire over time. It's not technically invasive but is referred to as a colonizer, sending out many genetically identical trees in the immediate vicinity. The best time to plant quaking aspen is spring, after the last frost date in your area. Quaking aspen grows best in full sun, although it tolerates some shade. The source also includes details about the physical features and maintenance requirements of quaking aspen. According to the text, quaking aspen is a fast-growing tree that can grow up to 2 feet per year. On average, it lives about 60 years and up to 150 years, but it can vary significantly depending on the region. It requires full sun to grow optimally and is known for its ability to spread, which is why regular maintenance is necessary to control the growth. The text | **5** |

Figure 17: Responses on a ClosedQA problem from the test split of "No Robots" dataset. Each response is generated by Mistral 7B models fine-tuned on 128K synthetic dataset with KaSA, MiLoRA, and PiSSA methods respectively.

---

**Prompt**

This Java function below is not working as I expected, it's supposed to go through an array of numbers and if it finds a number in the array that is divisible by 2 with a remainder of 0, then it adds it to a new array and returns the new array at the end. Can you help debug the function and explain how why it's not working as expected?

```
public static List checkForEvens(int [] nums) { ArrayList anwer = new ArrayList(); for(int i = 0; i < nums.length; i++) { if(nums[i] / 2
== 0) { answer.add(nums[i] }
} }
```

---

**Target response**

Certainly! The function 'checkForEvens' iterates through an array and if a number in the array is divisible by 2, then it gets added to 'answer' which is later returned.

The reason the function may not be returning the expected result is because of the if statement, 'if (nums[i] /2 == 0)'. The statement is looking for numbers that get divided by 2 in the array and equal 0, which doesn't align with what you expect the function to do.

Instead, you want to use the modulo operator to check for the remainder of a number that is being divided by 2. If a number is divided by 2 and returns a remainder of 0, then we can assume that it's an even number. So, the small change of 'if (nums[i] % 2 == 0)' should solve your issue.

I've rewritten the function below with the correction.

```
public static List checkForEvens(int [] nums) {
    ArrayList anwer = new ArrayList<Integer>();
    for(int i = 0; i < nums.length; i++) {
    if(nums[i] % 2 == 0) {
    answer.add(nums[i]
     }
   }
}
```

---

|  | Score (GPT4o as a Judge) |
|---|---|

**Gemma 7B with KaSA**

You have a typo in your code.

Here is the corrected version of the function:

```
public static List<Integer> checkForEvens(int[] nums) {
    List<Integer> answer = new ArrayList<>();
    for (int num : nums) {
        if (num % 2 == 0) {
            answer.add(num);
        }
    }
    return answer;
}
```

**9**

**Gemma 7B with PiSSA**

Your function is close to working correctly. There are a few simple fixes that need to be made. First, the modulus operator `%` needs to be used in the `if` condition to check for even numbers. Second, the `add` method needs to be called on the `ArrayList` with the element `nums[i]` instead of `nums[i]` itself. Finally, the `return` statement needs to be included in order to return the list of even numbers.
Here's the corrected version:
```java
public static List<Integer> checkForEvens(int[] nums) {
    List<Integer> answer = new ArrayList<>();
    for (int i = 0; i < nums.length; i++) {
        if (nums[i] % 2 == 0) {
            answer.add(nums[i]);
        }
    }
    return answer;
}
```

This function works correctly and will return an `ArrayList` of even numbers from the input array. If you need help understanding how this fixes your original function, please let me know.

**10**

Figure 18: Responses on a Coding problem from the test split of "No Robots" dataset. Each response is generated by Gemma 7B models fine-tuned on 128K synthetic dataset with KaSA and PiSSA methods respectively.

