# OpenReview forum: "KaSA: Knowledge-Aware Singular-Value Adaptation of Large Language Models"
_ICLR.cc/2025/Conference — ICLR 2025 Poster_

### Official Review · Reviewer_Xp8u · 2024-10-27

**Soundness:** 3
**Presentation:** 3
**Contribution:** 3
**Rating:** 6
**Confidence:** 2

**Summary:**

The paper presents Knowledge-aware Singular-Value Adaptation (KaSA), an innovative PEFT method aimed at overcoming the computational and memory challenges of adapting LLMs to specific tasks. KaSA employs singular value decomposition with knowledge-aware singular values to dynamically activate relevant parametric knowledge for downstream tasks. Through extensive experiments, the authors show that KaSA consistently outperforms full fine-tuning and other popular PEFT methods across a range of tasks, including NLU, NLG, and instruction following.

**Strengths:**

1. The method is novel, integrating SVD with knowledge-aware singular values. This method allows for the dynamic activation of relevant knowledge, which is a significant advancement over existing PEFT techniques.
2. Extensive experiments (various tasks with different LLMs, a wide range of baselines) consistently validate the efficacy and adaptability of the proposed method.
3. The paper is well written with good clarity.

**Weaknesses:**

My main concern -- most experiments, except experiments in Table 4 on instruction following, only use a small model (less than 1B). Need more experiments and fair comparison with the baselines on larger models. And the improvements on instruction following evaluation results are not significant, and there are even drops in performance on LLaMA3 8B.

**Questions:**

How well does KaSA scale to even larger models?

---

> ### Author Response · Authors · 2024-11-24
> **Rebuttal (Part 1/2)**
>
> Thank you for your comprehensive and insightful review of our paper. We greatly appreciate your recognition of the novelty of our work.
>
> > My main concern -- most experiments, except experiments in Table 4 on instruction following, only use a small model (less than 1B). Need more experiments and fair comparison with the baselines on larger models.
> >
>
> For fair comparison, we adopt the same experimental setups of existing PEFT methods `[1, 2, 3, 4]`that utilizes the small language model (less than 1B) for NLU and NLG tasks. For example, VeRA `[1]` and FourierFT `[2]` utilize RoBERTa-base (125M) and RoBERTa-large (355M) for NLU tasks (refer to Table 2), and GPT-2 medium (354.92M) and GPT-2 large (774.03M) for NLG tasks (refer to Table 3). PiSSA `[3]` employs RoBERTa-large (355M) and DeBERTaV3-base (184M) for NLU tasks. Similarly, the latest SARA `[4]` uses GPT-2 medium (354.92M) for NLG tasks (see Table 3).
>
> > And the improvements on instruction following evaluation results are not significant, and there are even drops in performance on LLaMA3 8B.
> >
>
> To demonstrate the significant improvements of our method, we perform an independent two-sample t-test to evaluate the statistical significance of the differences between our method and the baselines (excluding FFT due to its high computational demands). This evaluation is conducted over five runs with different random seeds to ensure robustness. All tests are carried out at a significance level of $\alpha = 0.05$. The performance comparison between our KaSA and key baselines is illustrated in the following Table.
>
> As evidenced, KaSA significantly outperforms all baselines, achieving p-values < 0.05 in 51 out of 60 experimental settings across the Gemma 7B, Mistral 7B, and LLaMA 3 8B on 5 datasets. Therefore, the significance tests substantiate the significant improvement of our method over the baselines, verifying the superiority and robustness of our approach.
>
> ### Gemma 7B
>
> | Method | # Trainable Parameters | Classification | Summarization | Coding | Closed QA |
> | --- | --- | --- | --- | --- | --- |
> | FFT | 8.54B | 5.58 | 7.78 | 7.61 | **8.88** |
> | LoRA | 3.21M | 5.98±0.3 (p=0.001) | 7.29±0.2 (p=0.002) | 7.75±0.2 (p=0.049) | 8.18±0.2 (p=0.002) |
> | PiSSA | 3.21M | 6.23±0.2 (p=0.002) | 7.88±0.1 (p=0.730) | 7.80±0.1 (p=0.018) | 8.22±0.2 (p=0.003) |
> | MiLoRA | 3.21M | 6.30±0.1 (p=0.001) | 7.62±0.2 (p=0.067) | 7.71±0.2 (p=0.028) | 8.27±0.3 (p=0.029) |
> | **KaSA** | 3.22M | **6.88±0.2** | **7.92±0.2** | **8.01±0.1** | 8.69±0.1 |
>
> ### Mistral 7B
>
> | Method | # Trainable Parameters | Classification | Summarization | Coding | Closed QA |
> | --- | --- | --- | --- | --- | --- |
> | FFT | 7.25B | **6.73** | **7.18** | **7.53** | **8.75** |
> | LoRA | 3.40M | 5.07±0.3 (p=0.007) | 5.72±0.2 (p=0.000) | 6.17±0.4 (p=0.034) | 7.39±0.2 (p=0.034) |
> | PiSSA | 3.40M | 5.46±0.2 (p=0.103) | 5.86±0.3 (p=0.002) | 6.41±0.2 (p=0.048) | 7.24±0.2 (p=0.007) |
> | MiLoRA | 3.40M | 5.33±0.2 (p=0.025) | 5.89±0.4 (p=0.006) | 6.52±0.2 (p=0.158) | 7.28±0.3 (p=0.031) |
> | **KaSA** | 3.41M | 5.72±0.2 | 6.82±0.3 | 6.74±0.2 | 7.75±0.2 |
>
> ### LLaMA3 8B
>
> | Method | # Trainable Parameters | Classification | Summarization | Coding | Closed QA |
> | --- | --- | --- | --- | --- | --- |
> | FFT | 8.03B | 5.44 | 7.80 | 7.59 | **8.90** |
> | LoRA | 3.40M | 6.12±0.3 (p=0.019) | 7.20±0.4 (p=0.016) | 7.37±0.2 (p=0.006) | 6.02±0.2 (p=0.002) |
> | PiSSA | 3.40M | 6.35±0.1 (p=0.028) | 7.31±0.3 (p=0.011) | 7.59±0.1 (p=0.028) | 6.18±0.3 (p=0.018) |
> | MiLoRA | 3.40M | 6.37±0.2 (p=0.083) | 7.61±0.1 (p=0.014) | 7.65±0.2 (p=0.128) | 6.39±0.1 (p=0.029) |
> | **KaSA** | 3.41M | **6.55±0.2** | **7.83±0.1** | **7.89±0.2** | 6.81±0.3 |
>
> ### LLaMA2 13B
>
> | Method | # Trainable Parameters | Classification | Summarization | Coding | Closed QA |
> | --- | --- | --- | --- | --- | --- |
> | FFT | 13.02B | 5.86 | **7.93** | 7.88 | **8.97** |
> | LoRA | 6.55M | 6.23±0.4 (p=0.023) | 7.38±0.2 (p=0.005) | 7.54±0.2 (p=0.005) | 6.25±0.3 (p=0.001) |
> | PiSSA | 6.55M | 6.47±0.3 (p=0.062) | 7.45±0.3 (p=0.031) | 7.83±0.1 (p=0.049) | 6.54±0.3 (p=0.006) |
> | MiLoRA | 6.55M | 6.45±0.2 (p=0.020) | 7.63±0.1 (p=0.032) | 7.85±0.1 (p=0.064) | 6.82±0.2 (p=0.028) |
> | **KaSA** | 6.56M | **6.86±0.2** | 7.92±0.2 | **8.09±0.2** | 7.12±0.1 |
>
> | Model | Method | MT-Bench |
> | --- | --- | --- |
> | Gemma 7B | FFT | 4.69 |
> |  | LoRA | 4.32±0.4 (p=0.032) |
> |  | PiSSA | 4.66±0.3 (p=0.182) |
> |  | MiLoRA | 4.53±0.2 (p=0.041) |
> |  | **KaSA** | **4.97±0.3** |
> | Mistral 7B | FFT | 4.22 |
> |  | LoRA | 4.18±0.3 (p=0.057) |
> |  | PiSSA | 4.24±0.2 (p=0.043) |
> |  | MiLoRA | 4.29±0.2 (p=0.074) |
> |  | **KaSA** | **4.58±0.2** |
> | LLaMA3 8B | FFT | 4.11 |
> |  | LoRA | 4.19±0.3 (p=0.020) |
> |  | PiSSA | 4.26±0.2 (p=0.013) |
> |  | MiLoRA | 4.32±0.2 (p=0.025) |
> |  | **KaSA** | **4.71±0.2** |
> | LLaMA2 13B | FFT | 4.37 |
> |  | LoRA | 4.43±0.3 (p=0.011) |
> |  | PiSSA | 4.39±0.2 (p=0.001) |
> |  | MiLoRA | 4.51±0.3 (p=0.024) |
> |  | **KaSA** | **4.95±0.1** |

---

> ### Author Response · Authors · 2024-11-24
> **Rebuttal (Part 2/2)**
>
> > How well does KaSA scale to even larger models?
> >
>
> To showcase the scalability of our KaSA model with even larger architectures, we conducted an instruction-following experiment using the Llama 2 13B model. Unfortunately, we are unable to test the LLaMA 3 70B model due to computational resource constraints, which require approximately 140GB of GPU memory.
>
> As indicated in the last row of the Tables in Part 1, KaSA consistently and significantly outperforms all baseline models in 13 out of 15 experimental settings, with p-values less than 0.05 when scaled to the larger 13B model. This demonstrates the scalability and robustness of our proposed method.
>
> ### Reference
>
> `[1]` Kopiczko, Dawid Jan, Tijmen Blankevoort, and Yuki Markus Asano. "Vera: Vector-based random matrix adaptation." ICLR, 2024.
>
> `[2]` Gao, Ziqi, Qichao Wang, Aochuan Chen, Zijing Liu, Bingzhe Wu, Liang Chen, and Jia Li. "Parameter-Efficient Fine-Tuning with Discrete Fourier Transform." ICML, 2024.
>
> `[3]` Meng, Fanxu, Zhaohui Wang, and Muhan Zhang. "Pissa: Principal singular values and singular vectors adaptation of large language models." NeurIPS, 2024.
>
> `[4]` Gu, Jihao, Shuai Chen, Zelin Wang, Yibo Zhang, and Ping Gong. "Sara: Singular-value based adaptive low-rank adaption." arXiv preprint arXiv:2408.03290 (2024).

---

> > ### Comment · Reviewer_Xp8u · 2024-11-27
> >
> > Thanks for your detailed results! I'll raise my score.

---

### Official Review · Reviewer_uEtH · 2024-11-03

**Soundness:** 2
**Presentation:** 3
**Contribution:** 2
**Rating:** 5
**Confidence:** 4

**Summary:**

This paper proposes a novel PEFT method called KaSA. The main idea is to leverage singularvalue decomposition with knowledge-aware singular values to dynamically activate knowledge based on its relevance to the task at hand. Experiments on NLU, NLG and instruction tuning tasks demonstrate the effectiveness of the proposed method.

**Strengths:**

- The paper is well written and easy to follow.
- The performance is good.

**Weaknesses:**

- The proposed method has a more complex training objective compared to baselines like LoRA, PiSSA, and MiLoRA. Consequently, the training time for KaSA may be longer than that of the baselines. However, the paper does not provide detailed information on this.
- The training objective introduces new hyperparameters, including \beta and \gamma, but there is no analysis regarding their impact.
- How does KaSA perform at different ranks? Can it consistently exceed the baselines across various ranks? This aspect also needs to be validated.
- Some important baselines are missing, like [1]

[1] CorDA: Context-Oriented Decomposition Adaptation of Large Language Models for Task-Aware Parameter-Efficient Fine-tuning

**Questions:**

- The PEFT methods are sensitive to hyperparameters like the learning rate. How does the paper ensure a fair comparison with the baseline models? For instance, in Section 4.2, a hyperparameter search is conducted for KaSA for each dataset. Is a similar hyperparameter search performed for all the baseline methods? For Section 4.3 and 4.4, do KaSA and baselines share the same hyperparameters?
- The setup for the instruction-following experiments is questionable. Given the availability of numerous instruction tuning datasets, why not conduct experiments using these existing datasets? Since other researchers may not have access to the synthetic dataset, reproducing the results could be challenging.

---

> ### Author Response · Authors · 2024-11-24
> **Rebuttal (Part 1/4)**
>
> Thank you for your comprehensive and insightful review of our paper. We appreciate your valuable feedback and suggestions.
>
> > The proposed method has a more complex training objective compared to baselines like LoRA, PiSSA, and MiLoRA. Consequently, the training time for KaSA may be longer than that of the baselines. However, the paper does not provide detailed information on this.
> >
>
> Thank you for bringing this to our attention. We have conducted a comprehensive efficiency and complexity comparison between LoRA and SVD variants across different tasks and model scales, as shown in the table below. We clarify that the dynamic singular value adaptation introduced in our KaSA is a learnable one-dimensional vector of size $r \ll m$ and requires parameter regularizations, incurring negligible training overheads compared to the standard LoRA. In addition, due to low-rank approximation of the original matrix, we reduce the rank of $\mathbf{W}$ from $m$ to $m-r$, accelerating the inference particularly for small-scale language models like RoBERTa-base 125M (i.e., with small $m$). We show the experimental results as follows.
>
> As can be seen, compared to LoRA, the extra training overhead is less than 20% (resp. 3%) for the NLU (resp. NLG) task, while speeding up the inference by 1.45x (resp. 1.02x) times.  When compared to PiSSA and MiLoRA, our approach incurs an average of less than 13% extra training overhead for NLU tasks, while maintaining comparable or improved inference latency. For NLG tasks, our method introduces similar training overhead or inference latency.
>
> **RoBERTa-base 125M on Single NVIDIA GeForce RTX 3090 (24GB) GPU for NLU task**
>
> | Method | LoRA | PiSSA | MiLoRA | KaSA |
> | --- | --- | --- | --- | --- |
> | \# Trainable Parameters | 0.23716% | 0.23716% | 0.23716% | 0.23732% |
> | \# GPU Memory | 1638M | 1638M | 1638M | 1650M |
> | \# Training FLOPs ($\times 10^9$ per sample) | 2.0306 | 1.9270 | 1.9270 | 2.1503 |
> | Training Latency (per epoch) | 9.4868s | 9.8825s | 9.9267s | 11.3679s |
> | Inference Latency (per batch size 32) | 0.0173s | 0.0108s | 0.0165s | 0.0119s |
> | Matrix Rank | $\text{rank}(\mathbf{W})=m$  $\text{rank}(\mathbf{\Delta W})=r$ | $\text{rank}(\mathbf{W})=m-r$  $\text{rank}(\mathbf{\Delta W})=r$ | $\text{rank}(\mathbf{W})=m-r$  $\text{rank}(\mathbf{\Delta W})=r$ | $\text{rank}(\mathbf{W})=m-r$  $\text{rank}(\mathbf{\Delta W})\leq r$ |
> | CoLA Performance (Mcc.) | 63.4% | 65.5% | 63.1% | **65.8%** |
>
> **LLaMA3 8B on Single NVIDIA A100-SXM4 (80GB) GPU for NLG task**
>
> | Method | LoRA | PiSSA | MiLoRA | KaSA |
> | --- | --- | --- | --- | --- |
> | \# Trainable Parameters | 0.04241% | 0.04241% | 0.04241% | 0.04242% |
> | \# GPU Memory | 71023M | 71023M | 71023M | 71095M |
> | \# Training FLOPs ($\times 10^9$ per sample) | 240.2583 | 240.2583 | 240.2583 | 240.2585 |
> | Training Latency (per epoch) | 2469.6s | 2543.1s | 2476.8s | 2528.9s |
> | Inference Latency (per batch size 16) | 0.7898s | 0.7687s | 0.7705s | 0.7771s |
> | Matrix Rank | $\text{rank}(\mathbf{W})=m$  $\text{rank}(\mathbf{\Delta W})=r$ | $\text{rank}(\mathbf{W})=m-r$  $\text{rank}(\mathbf{\Delta W})=r$ | $\text{rank}(\mathbf{W})=m-r$  $\text{rank}(\mathbf{\Delta W})=r$ | $\text{rank}(\mathbf{W})=m-r$  $\text{rank}(\mathbf{\Delta W})\leq r$ |
> | MT-Bench Performance (Scores) | 4.1937 | 4.2625 | 4.3187 | **4.7125** |

---

> ### Author Response · Authors · 2024-11-24
> **Rebuttal (Part 2/4)**
>
> > The training objective introduces new hyperparameters, including \beta and \gamma, but there is no analysis regarding their impact.
> >
>
> We appreciate the reviewer's insightful feedback on this matter. We put great effort into meticulously tuning the two trade-off loss coefficients, $\beta \in [\text{1E-5}, 1]$ and $\gamma \in [\text{1E-5}, 1]$, to explore their impacts. The sensitivity analysis is presented in the Table below. The results demonstrate that KaSA is not sensitive to variations in $\beta$ and $\gamma$.
>
> | Hyperparameters | RoBERTa-base CoLA | RoBERTa-large SST-2 | DeBERTa-v3-base MRPC |
> | --- | --- | --- | --- |
> | $\beta=0.01,\gamma=1.0$ | 0.6581 | 0.9587 | 0.9044 |
> | $\beta=0.1,\gamma=0.0001$ | 0.6334 | 0.9587 | 0.8971 |
> | $\beta=0.01,\gamma=0.1$ | 0.6414 | 0.9622 | 0.8995 |
> | $\beta=0.0,\gamma=0.0$ | 0.646 | 0.9599 | 0.902 |
> | $\beta=0.001,\gamma=0.01$ | 0.6358 | 0.9587 | 0.9093 |
> | $\beta=0.001,\gamma=0.001$ | 0.6553 | 0.9576 | 0.9093 |
> | $\beta=0.01,\gamma=0.001$ | 0.6506 | 0.5092 | 0.902 |
> | $\beta=0.1,\gamma=0.01$ | 0.6333 | 0.9587 | 0.902 |
> | $\beta=0.0001,\gamma=0.1$ | 0.6485 | 0.9622 | 0.8995 |
> | $\beta=0.01,\gamma=0.0001$ | 0.6347 | 0.9576 | 0.9044 |
> | $\beta=0.0001,\gamma=0.01$ | 0.658 | 0.9599 | 0.9069 |
> | $\beta=1.0,\gamma=0.1$ | 0.6241 | 0.9599 | 0.8971 |
> | $\beta=1.0,\gamma=1.0$ | 0.6291 | 0.9553 | **0.9142** |
> | $\beta=0.1,\gamma=1.0$ | 0.6436 | 0.961 | 0.9093 |
> | $\beta=0.1,\gamma=0.1$ | 0.653 | 0.9587 | 0.9082 |
> | $\beta=1.0,\gamma=0.01$ | 0.6397 | 0.9587 | 0.8995 |
> | $\beta=0.01,\gamma=0.01$ | 0.6433 | 0.9576 | 0.8995 |
> | $\beta=0.0001,\gamma=0.0001$ | 0.6565 | **0.9687** | 0.9044 |
> | $\beta=0.0001,\gamma=0.001$ | **0.6582** | 0.961 | 0.9093 |
> | $\beta=0.1,\gamma=0.001$ | 0.6338 | 0.9599 | 0.902 |
> | $\beta=0.001,\gamma=0.0001$ | 0.6504 | 0.961 | 0.9093 |
> | $\beta=0.001,\gamma=0.1$ | 0.648 | 0.9679 | 0.8971 |
>
> > How does KaSA perform at different ranks? Can it consistently exceed the baselines across various ranks? This aspect also needs to be validated.
> >
>
> It would be very helpful to have a performance comparison at different ranks. We examine the impact of rank across LoRA and SVD variant baselines, with the range of ranks set to $r = \\{1,2,4,8,16,32,64,128 \\}$. The experimental results are presented in the Table below.
>
> We observe that KaSA consistently surpasses LoRA and the SVD variant baselines, including PiSSA and MiLoRA, across various ranks except in 4 out of 96 cases on the MRPC, CoLA, and RTE datasets, highlighting the efficacy and robustness of our method.
>
> | Dataset | Rank | 1 | 2 | 4 | 8 | 16 | 32 | 64 | 128 |
> | --- | --- | --- | --- | --- | --- | --- | --- | --- | --- |
> | **CoLA** | LoRA | 60.08 | 61.17 | 63.14 | 63.77 | 63.58 | 63.82 | 62.70 | 63.45 |
> |  | MiLoRA | 60.84 | 61.36 | 63.10 | 63.07 | 63.57 | 64.56 | 63.60 | 63.66 |
> |  | PiSSA | 59.56 | 62.68 | 60.57 | 65.54 | 61.32 | 63.31 | 63.35 | 63.60 |
> |  | KaSA | **63.32** | **65.58** | **63.56** | **65.82** | **64.39** | **65.05** | **64.82** | **65.06** |
> | **MRPC** | LoRA | 88.73 | 87.74 | 88.97 | 88.73 | 89.46 | 89.95 | 88.97 | 88.97 |
> |  | MiLoRA | **89.71** | **89.22** | 88.48 | 88.73 | 88.73 | 90.20 | 88.73 | 88.73 |
> |  | PiSSA | 87.25 | 87.99 | 88.24 | 88.24 | 89.46 | 89.71 | 88.97 | 89.95 |
> |  | KaSA | 89.46 | 87.99 | **90.20** | **90.69** | **89.95** | **90.44** | **90.20** | **90.44** |
> | **RTE** | LoRA | 71.84 | 72.56 | 75.45 | 78.70 | 77.26 | 77.98 | 79.78 | 78.70 |
> |  | MiLoRA | 75.09 | **80.14** | **79.42** | 80.51 | 79.06 | 79.81 | 81.59 | 80.87 |
> |  | PiSSA | 68.95 | 73.29 | 76.17 | 75.09 | 76.90 | 78.34 | 76.53 | 79.42 |
> |  | KaSA | **77.62** | 77.62 | 78.70 | **81.59** | **80.51** | **81.23** | **82.67** | **81.23** |

---

> ### Author Response · Authors · 2024-11-24
> **Rebuttal (Part 3/4)**
>
> > Some important baselines are missing, like [1] CorDA: Context-Oriented Decomposition Adaptation of Large Language Models for Task-Aware Parameter-Efficient Fine-tuning
> >
>
> It is a good suggestion to include the comparison with this important baseline in our study. To ensure a fair comparison, we have reached out to the authors of CorDA to verify the implementation details on the GLUE benchmark and have reproduced their results successfully. (We would like to express our sincere gratitude to the authors of CorDA for their generous assistance.)
>
> In accordance with the experimental settings outlined in the CorDA paper, we present the experimental results for our KaSA with a rank of $r=128$. The results are displayed in the Table below. We observe that KaSA consistently outperforms all baselines across 6 datasets, with a slight exception for the QNLI dataset, where it performs marginally worse than FFT (92.71 vs. 92.8).
>
> | Method | # Trainable Parameters | SST-2  (Acc.) | MRPC   (Acc.) | CoLA   (Mcc.) | QNLI   (Acc.) | RTE  (Acc.) | STS-B   (Pcc.) | All Avg. |
> | --- | --- | --- | --- | --- | --- | --- | --- | --- |
> | FFT | 125.0M | 94.8 | 90.2 | 63.6 | **92.8** | 78.7 | 91.2 | 85.2 |
> | LoRA | 21M | 94.15 | 82.84 | 54.24 | 92.48 | 64.26 | 88.58 | 79.43 |
> | DoRA | 21M | 93.58 | 83.58 | 51.93 | 92.59 | 64.98 | 88.71 | 79.23 |
> | CorDA | 21M | 93.12 | 89.71 | 59.60 | 91.49 | 76.17 | 90.17 | 83.38 |
> | KaSA | 21M | **95.30** | **90.44** | **65.06** | 92.71 | **81.23** | **91.36** | **86.02** |
>
> > The PEFT methods are sensitive to hyperparameters like the learning rate. How does the paper ensure a fair comparison with the baseline models? For instance, in Section 4.2, a hyperparameter search is conducted for KaSA for each dataset. Is a similar hyperparameter search performed for all the baseline methods? For Section 4.3 and 4.4, do KaSA and baselines share the same hyperparameters?
> >
>
> To ensure a fair comparison among various PEFT methods, we adhere to a common practice to directly utilize the experimental results reported in their original papers. For instance, VeRA `[2]`, FourierFT `[3]`, and LoRA `[5]` all incorporate experimental results from prior works for NLU tasks on the GLUE benchmark and NLG tasks on the E2E benchmark.
>
> Specifically,
>
> (1) In our paper, Section 4.2, Table 1, we adopt the baseline results—excluding PiSSA and MiLoRA—from Table 2 of FourierFT `[3]`. In Table 2, we use baseline results—excluding PiSSA and MiLoRA—from Table 2 of BiLoRA `[6]`. We follows `[2]` and `[3]` to perform a hyperparameter search to determine optimal parameters.
>
> (2) In Section 4.3, Table 3, baseline model results are sourced from SARA `[4]` and BiLoRA `[6]`. Our KaSA employs the same hyperparameters as LoRA that is specified at [https://github.com/microsoft/LoRA/tree/main/examples/NLG](https://github.com/microsoft/LoRA/tree/main/examples/NLG).
>
> (3) In Section 4.4, for fair comparison, we make extensive efforts to meticulously tune the hyperparameters to achieve optimal performance for each baseline. To avoid randomness in grid searching, we refer to the LoRA NLG configuration available on the official GitHub repository at [https://github.com/microsoft/LoRA/tree/main/examples/NLG](https://github.com/microsoft/LoRA/tree/main/examples/NLG).
>
> (4) Regarding PiSSA and MiLoRA, the PiSSA implementation has been integrated into the official PEFT packages. Thus, we use PiSSA from PEFT, following the guidelines at [https://huggingface.co/docs/peft/v0.13.0/en/developer_guides/lora#pissa](https://huggingface.co/docs/peft/v0.13.0/en/developer_guides/lora#pissa:). For the MiLoRA implementation, we modify the PiSSA implementation with a few lines of code, as they are complementary initialization approaches. Since PiSSA and MiLoRA are parameter initialization methods for LoRA, their configurations should be strictly consistent with the original LoRA configuration. Thus, for NLU tasks, we employ the LoRA configuration for PiSSA and MiLoRA, following the guidelines at [https://github.com/microsoft/LoRA/tree/main/examples/NLU](https://github.com/microsoft/LoRA/tree/main/examples/NLU). However, for instruction-following tasks, we adhere to the same LoRA configuration. Unlike LoRA, the rank $r$ should be equal to $\alpha$, as mentioned in their paper.

---

> ### Author Response · Authors · 2024-11-24
> **Rebuttal (Part 4/4)**
>
> > The setup for the instruction-following experiments is questionable. Given the availability of numerous instruction tuning datasets, why not conduct experiments using these existing datasets? Since other researchers may not have access to the synthetic dataset, reproducing the results could be challenging.
> >
>
> Thank you for highlighting this important aspect.
>
> **Reason**: Many existing instruction datasets are synthetic dataset generated by LLMs. For example, Alpaca 51k and WizardLM-Evol-Instruct 70k (building upon Alpaca) are generated by OpenAI's text-davinci-003 and gpt-3.5-turbo models, respectively. However, as highlighted in numerous studies, the original Alpaca 51k dataset has several issues, including poor data quality, hallucinations, merged instructions, empty outputs, and empty code examples. Detailed issues can be found at [https://github.com/gururise/AlpacaDataCleaned](https://github.com/gururise/AlpacaDataCleaned). Additionally, the model used for data synthesis is somewhat outdated, which limits the quality of the instruction dataset. Due to these limitations, we did not employ these synthetic datasets in our experiments.
>
> In this work, we contribute a high-quality synthetic instruction datasets by GPT4o to to enhance the functionality of PEFT and support future research endeavors. As acknowledged by **Reviewer 9ooo**, the `'no robots'` dataset is a good choice for seeding synthetic data, as it was generated by human annotators. To maintain anonymity, we cannot provide you with the specific dataset link on Hugging Face at this time. We are committed to publicly releasing the dataset soon.
>
> Moreover, as per your advice, we have followed the VeRA `[2]` and FourierFT `[3]` to fine-tune on Alpaca 51k and report the evaluation results on MT-Bench, with GPT4 serving as the judge, yielding scores within 10. The results are presented in the Table below.
>
> As can be seen, our KaSA consistently and significantly outperforms the LoRA and SVD variant baselines, achieving p-values < 0.05 in 9 out of 12 experimental settings across Gemma 7B, Mistral 7B, LLaMA 3 8B, and LLaMA 2 13B. These findings align with experiments conducted on our synthetic datasets.
>
> | Model | Method | MT-Bench |
> | --- | --- | --- |
> | Gemma 7B | FFT | 4.69 |
> |  | LoRA | 4.32±0.4 (p=0.032) |
> |  | PiSSA | 4.66±0.3 (p=0.182) |
> |  | MiLoRA | 4.53±0.2 (p=0.041) |
> |  | **KaSA** | **4.97±0.3** |
> | Mistral 7B | FFT | 4.22 |
> |  | LoRA | 4.18±0.3 (p=0.057) |
> |  | PiSSA | 4.24±0.2 (p=0.043) |
> |  | MiLoRA | 4.29±0.2 (p=0.074) |
> |  | **KaSA** | **4.58±0.2** |
> | LLaMA3 8B | FFT | 4.11 |
> |  | LoRA | 4.19±0.3 (p=0.020) |
> |  | PiSSA | 4.26±0.2 (p=0.013) |
> |  | MiLoRA | 4.32±0.2 (p=0.025) |
> |  | **KaSA** | **4.71±0.2** |
> | LLaMA2 13B | FFT | 4.37 |
> |  | LoRA | 4.43±0.3 (p=0.011) |
> |  | PiSSA | 4.39±0.2 (p=0.001) |
> |  | MiLoRA | 4.51±0.3 (p=0.024) |
> |  | **KaSA** | **4.95±0.1** |
>
> ### Reference
>
> `[1]` Yang, Yibo, Xiaojie Li, Zhongzhu Zhou, Shuaiwen Leon Song, Jianlong Wu, Liqiang Nie, and Bernard Ghanem. "CorDA: Context-Oriented Decomposition Adaptation of Large Language Models for Task-Aware Parameter-Efficient Fine-tuning." NeurIPS, 2024.
>
> `[2]` Kopiczko, Dawid Jan, Tijmen Blankevoort, and Yuki Markus Asano. "Vera: Vector-based random matrix adaptation." ICLR, 2024.
>
> `[3]` Gao, Ziqi, Qichao Wang, Aochuan Chen, Zijing Liu, Bingzhe Wu, Liang Chen, and Jia Li. "Parameter-Efficient Fine-Tuning with Discrete Fourier Transform." ICML, 2024.
>
> `[4]` Gu, Jihao, Shuai Chen, Zelin Wang, Yibo Zhang, and Ping Gong. "Sara: Singular-value based adaptive low-rank adaption." arXiv preprint arXiv:2408.03290 (2024).
>
> `[5]` Hu, Edward J., Yelong Shen, Phillip Wallis, Zeyuan Allen-Zhu, Yuanzhi Li, Shean Wang, Lu Wang, and Weizhu Chen. "Lora: Low-rank adaptation of large language models." ICLR, 2021.
>
> `[6]` Qiang, Rushi, Ruiyi Zhang, and Pengtao Xie. "BiLoRA: A Bi-level Optimization Framework for Overfitting-Resilient Low-Rank Adaptation of Large Pre-trained Models." arXiv preprint arXiv:2403.13037 (2024).

---

> > ### Comment · Reviewer_uEtH · 2024-11-26
> >
> > Thank you for providing such detailed responses. As my concerns regarding training efficiency and the new hyperparameters \beta and \gamma  have been addressed, I have raised the score to 5. However, several major concerns remain:
> >
> > - Ensuring a Fair Comparison Between PEFTs: While the authors offer some explanations, I still find it difficult to conclude that KaSA is superior to PiSSA, MiLoRA, or CorDA due to the hyperparameter setups.
> > - Redesign of Experimental Sections: As suggested by Reviewer Xp8u, the authors have not conducted comprehensive experiments on large language models and well-known LLM benchmarks. Although additional results were provided during the rebuttal, these are not sufficient to comprehensively demonstrate that KaSA is a better choice compared to existing baselines.
> >
> > In conclusion, while KaSA holds promise, the current experiment parts are insufficient and require further refinement.

---

> ### Author Response · Authors · 2024-12-02
> **Fair Comparison on 8 New LLM Benchmarks**
>
> Thank you again for your feedback. We are very pleased to know that most of your concerns have been addressed. To further address your concerns regarding experiments on fair comparison and on large language models and well-known LLM benchmarks, we have conducted new experiments with relentless efforts so that the response is slightly delayed.
>
> > Ensuring a Fair Comparison Between PEFTs: While the authors offer some explanations, I still find it difficult to conclude that KaSA is superior to PiSSA, MiLoRA, or CorDA due to the hyperparameter setups.
> >
>
> To ensure a fair comparison, we have reached out to the authors of MiLoRA to verify the implementation details on the eight commonsense reasoning benchmarks and have reproduced their results successfully. (We would like to express our sincere gratitude to the authors of MiLoRA for their generous assistance and guidance.)
>
> Specifically, to maintain fairness in our comparison, we undertook the following measures:
>
> 1. All methods are implemented within the open-source framework LLM-Adapters `[1]`, available at [https://github.com/AGI-Edgerunners/LLM-Adapters](https://github.com/AGI-Edgerunners/LLM-Adapters), following MiLoRA `[2]`.
> 2. We adhered strictly to the hyperparameter configurations for training and evaluation as specified in papers `[1]` and `[2]`, `without any tuning`, as advised by the authors of MiLoRA to ensure fair implementation.
> 3. We have anonymously uploaded the training and evaluation logs, including hyperparameter configurations and performance results, for your verification. Please check the link at [train_llama-2-7b-kasa.log](https://anonymous.4open.science/r/KaSA-ICLR-Submission-A533/commonsense_reasoning_logs/train_llama-2-7b-kasa.log) and [eval_llama-2-7b-kasa-benchmarks.log](https://anonymous.4open.science/r/KaSA-ICLR-Submission-A533/commonsense_reasoning_logs/eval_llama-2-7b-kasa-boolq.log).
>
> The results of LLaMA2 7B are presented in the Table below. With identical hyperparameter setups, we observe that KaSA consistently outperforms all strong baselines across the 8 commonsense reasoning datasets.
>
> | **Method** | **BoolQ** | **PIQA** | **SIQA** | **HellaSwag** | **WinoGrande** | **ARC-e** | **ARC-c** | **OBQA** | **Avg.** |
> | --- | --- | --- | --- | --- | --- | --- | --- | --- | --- |
> | LoRA$*$ | 69.8 | 79.9 | 79.5 | 83.6 | 82.6 | 79.8 | 64.7 | 81.0 | 77.6 |
> | PiSSA$*$ | 67.6 | 78.1 | 78.4 | 76.6 | 78.0 | 75.8 | 60.2 | 75.6 | 73.8 |
> | MiLoRA$*$ | 67.6 | 83.8 | 80.1 | 88.2 | 82.0 | 82.8 | 68.8 | 80.6 | 79.2 |
> | KaSA | **73.6** | **84.4** | **80.2** | **91.5** | **84.5** | **84.7** | **72.1** | **81.2** | **81.5** |
>
> The symbol $*$ indicates that the results are taken from the original MiLoRA paper `[2]`.
>
> > Redesign of Experimental Sections: As suggested by Reviewer Xp8u, the authors have not conducted comprehensive experiments on large language models and well-known LLM benchmarks. Although additional results were provided during the rebuttal, these are not sufficient to comprehensively demonstrate that KaSA is a better choice compared to existing baselines.
> >
>
> To date, our experiments have been conducted in NLU (RoBERTa-Base, RoBERTa-Large, and DeBERTaV3-base), NLG (GPT-2 Large and GPT-2 Medium), instruction tuning (Gemma 7B, Mistral 7B, Llama 3 8B, and Llama 2 13B), and commonsense reasoning (Llama 2 7B). We utilize the E2E NLG Challenge benchmark, four 128k synthetic datasets, the 51k Alpaca dataset, an MT-Bench benchmark, a commonsense170k dataset, and 8 commonsense reasoning benchmarks. We believe these comprehensive experiments sufficiently verify the superiority of our KaSA compared to existing baselines. For ease of reference, in what follows, we list the positions of the aforementioned experiments in our revised paper.
>
> | **Experiments** | **LLMs** | **Benchmarks** | **Placement** |
> | --- | --- | --- | --- |
> | NLU | RoBERTa (Base / Large), DeBERTaV3-base | SST-2 / MRPC / CoLA / QNLI / RTE / STS-B  | Table 1, Table 2, Table 12 |
> | NLG | GPT-2 (Large / Medium) | E2E NLG Challenge | Table 3 |
> | Instruction Tuning | Gemma 7B / Mistral 7B / LLaMA 3 8B / LLaMA 2 13B | Four 128k Synthetic Datasets / 51k Alpaca / MT-Bench | Table 4 |
> | Commonsense Reasoning | LLaMA 2 7B | Commonsense170K / BoolQ / PIQA / SIQA / HellaSwag / WinoGrande / ARC-e / ARC-c / OBQA | Table Above |
>
> ### Reference
>
> `[1]` Hu, Zhiqiang, Lei Wang, Yihuai Lan, Wanyu Xu, Ee-Peng Lim, Lidong Bing, Xing Xu, Soujanya Poria, and Roy Lee. "LLM-Adapters: An Adapter Family for Parameter-Efficient Fine-Tuning of Large Language Models." EMNLP, 2023.
>
> `[2]` Wang, Hanqing, Yixia Li, Shuo Wang, Guanhua Chen, and Yun Chen. "Milora: Harnessing minor singular components for parameter-efficient llm finetuning." *arXiv preprint arXiv:2406.09044* (2024).

---

> > ### Author Response · Authors · 2024-12-03
> > **Looking forward to hearing from you**
> >
> > Dear Reviewer,
> >
> > As the deadline for the rebuttal period approaches, we look forward to hearing your feedback on our responses. We would be happy to address any remaining concerns that you may still have.
> >
> > Thanks,
> >
> > Authors

---

> ### Author Response · Authors · 2024-12-04
> **Performance of LLaMA3 8B on 8 commonsense reasoning datasets**
>
> Dear Reviewer,
>
> Thank you again for your feedback.
>
> The results of **LLaMA3 8B**, which has just completed its training and evaluation, are presented in the Table below. With identical hyperparameter setups, we observe that KaSA consistently outperforms all strong baselines by a large margin across the 8 commonsense reasoning datasets, with a slight exception for the WinoGrande dataset, where it performs marginally worse than MiLoRA (85.5 vs. 85.6).
>
> | **Method**  | **BoolQ** | **PIQA** | **SIQA** | **HellaSwag** | **WinoGrande** | **ARC-e** | **ARC-c** | **OBQA** | **Avg.** |
> |-------------|-----------|----------|----------|---------------|----------------|-----------|-----------|----------|----------|
> | LoRA$*$       | 70.8      | 85.2     | 79.9     | 91.7          | 84.3           | 84.2      | 71.2      | 79.0     | 80.8     |
> | PiSSA$*$      | 67.1      | 81.1     | 77.2     | 83.6          | 78.9           | 77.7      | 63.2      | 74.6     | 75.4     |
> | MiLoRA$*$     | 68.8      | 86.7     | 77.2     | 92.9          | **85.6**       | 86.8      | 75.5      | 81.8     | 81.9     |
> | KaSA        | **73.6**  | **88.1** | **80.4** | **94.7**      | 85.5           | **89.7**  | **79.4**  | **85.6** | **84.6** |
>
> The symbol $*$ indicates that the results are taken from the original MiLoRA paper `[2]`.
>
> The training and evaluation logs, including hyperparameter configurations and performance results, have also been uploaded for your verification. Please check the link to [train_llama-3-8b-kasa.log](https://anonymous.4open.science/r/KaSA-ICLR-Submission-A533/commonsense_reasoning_logs/train_llama-3-8b-kasa.log) and [eval_llama-3-8b-kasa-benchmarks.log](https://anonymous.4open.science/r/KaSA-ICLR-Submission-A533/commonsense_reasoning_logs/eval_llama-3-8b-kasa-boolq.log).
>
> Thank you.
>
> Authors

---

### Official Review · Reviewer_dg7a · 2024-11-05

**Soundness:** 3
**Presentation:** 3
**Contribution:** 4
**Rating:** 8
**Confidence:** 4

**Summary:**

The paper introduces KaSA (Knowledge-aware Singular-value Adaptation), a parameter-efficient fine-tuning (PEFT) method that adapts large language models (LLMs) using knowledge-aware singular values derived from singular value decomposition (SVD). KaSA aims to selectively activate task-relevant knowledge, discarding irrelevant or noisy information to improve model performance on specific tasks. The method consists of knowledge-based SVD truncation to filter noise and knowledge-aware singular value adaptation to dynamically manage task-specific knowledge activation. Experiments across natural language understanding, generation, and instruction-following tasks demonstrate that KaSA consistently outperforms other PEFT approaches, including LoRA and its variants.

**Strengths:**

KaSA combines SVD truncation with a knowledge-aware adaptation process, enabling it to achieve impressive performance on various NLP tasks. KaSA often outperforms other PEFT methods on benchmarks, showing its effectiveness in balancing performance and efficiency. The experiments are thorough, covering natural language understanding, generation, and instruction-following.
The ablation studies break down how each component (like the knowledge-aware adaptation) contributes to the overall performance, giving readers a clearer picture of why KaSA works so well.

**Weaknesses:**

The method adds some complexity, especially with the dynamic singular value adaptation, which might make it harder to implement compared to simpler PEFT methods. A potential limitation is the risk that SVD truncation might discard useful knowledge, especially if it’s not fully adapted to a task. It would also help to see this method applied to a wider range of LLM architectures to confirm its generalizability. While KaSA does well on the tested models, applying it to newer transformer architectures or other types of large models could demonstrate its versatility even more.

**Questions:**

The SVD truncation stage may discard knowledge components deemed noisy or irrelevant. However, there’s a risk that some truncated components might contain task-relevant information. Did you experiment with different truncation thresholds, or analyze the potential trade-offs of truncation in various tasks? A discussion here would enhance understanding of KaSA’s robustness.

The SVD rank and regularization coefficients play an important role in KaSA’s performance. Could you provide practical guidelines or a sensitivity analysis to help practitioners choose these values for new tasks or datasets? This could make the method more accessible to those less familiar with PEFT or SVD.

---

> ### Author Response · Authors · 2024-11-24
> **Rebuttal (Part 1/2)**
>
> Thank you for your thorough and insightful review. We greatly appreciate your recognition of the contributions of our work.
>
> > The method adds some complexity, especially with the dynamic singular value adaptation, which might make it harder to implement compared to simpler PEFT methods.
> >
>
> Compared to LoRA, KaSA requires a small amount of additional efforts. Specifically, we have provided a PyTorch-style pseudocode block illustrating our KaSA implementation, which requires only three additional operations:
>
> 1. Knowledge-based SVD truncation, which involves just 4 additional lines of code.
> 2. Knowledge-aware singular-value adaptation, necessitating only 1 additional line of code.
> 3. Parameters regularization losses, which require 20 additional lines of code.
>
> ```python
> class KaSA(nn.Module):
>     def __init__(self,
> 	    rank: int = 8, # kasa rank
> 	    alpha: int = 16, # kasa alpha
> 	    base_layer: nn.Module # pre-trained layer
>     )
>         # definitions
>         self.r = rank
>         self.alpha = alpha
>         self.scaling = alpha / rank
>         self.in_features, self.out_features = base_layer.in_features, base_layer.out_features
>
>         # Step 1: knowledge-based SVD truncation
>         self.svd_rank = self.in_features - self.r
>         U, S, Vh = torch.linalg.svd(base_layer.weight.data, full_matrices=False)
>         base_layer.weight.data = U[:, :self.svd_rank] @ torch.diag(S[:self.svd_rank]) @ Vh[:self.svd_rank, :]
>         self.base_layer = base_layer
>
>         # Step 2: knowledge-aware singular-value adaptation
>         self.delta_v = nn.Linear(self.in_features, self.r, bias=False)
>         self.delta_sigma = torch.diag(nn.Parameter(torch.randn(self.r), requires_grad=True))
>         self.delta_u = nn.Linear(self.r, self.out_features, bias=False)
>
>     def forward(self, x: torch.Tensor):
>     	# Step 3: merge W + Delta_W (Eq.7)
>         Delta_W = self.delta_u @ self.delta_sigma @ self.delta_v
>         result = self.base_layer(x)
>         result = result + torch.einsum('ijk,kl->ijl', x, Delta_W) * self.scaling
>         return result
>
> def regularization_loss(model, beta, gamma):
>     l2_loss, l3_loss = 0.0, 0.0
>     num_param = 0
>     for name, param in model.named_parameters():
>         if param.requires_grad:
>             if 'delta_sigma' in name:
>                 num_param += 1
>                 diag_norm = torch.sum(param ** 2)
>                 l2_loss += diag_norm
>             elif 'delta_v' in name or 'delta_u' in name:
>                 if 'delta_v' in name:
>                     matmul_result = torch.matmul(param.T, param)
>                 else:
>                     matmul_result = torch.matmul(param, param.T)
>
>                 I = torch.eye(matmul_result.size(0), device=matmul_result.device)
>                 diff_I = matmul_result - I
>                 matrix_loss = torch.norm(diff_I, p='fro')
>                 l3_loss += matrix_loss
>     auxi_loss = (beta * l2_loss/num_p + gamma * l3_loss) / num_param if num_param > 0 else 0.0
>     return auxi_loss
> ```
>
> > It would also help to see this method applied to a wider range of LLM architectures to confirm its generalizability. While KaSA does well on the tested models, applying it to newer transformer architectures or other types of large models could demonstrate its versatility even more.
> >
>
> Good suggestion. In this study, we examine two types of LLM architectures: encoder-only language models, which include RoBERTa-base, RoBERTa-large, and DeBERTaV3-base, and decoder-only language models, such as GPT-2, Gemma, Mistral, and LlaMA3. We agree that applying our KaSA method to newer Transformer architectures, such as Mamba `[1]`, can further enhance our work. Due to time limitations, we promise to explore this later.

---

> ### Author Response · Authors · 2024-11-24
> **Rebuttal (2/2)**
>
> > The SVD truncation stage may discard knowledge components deemed noisy or irrelevant. However, there’s a risk that some truncated components might contain task-relevant information. Did you experiment with different truncation thresholds, or analyze the potential trade-offs of truncation in various tasks? A discussion here would enhance understanding of KaSA’s robustness.
> >
>
> Nice comments. Based on the formulation in Equation (7), the reconstruction error between the original and truncated weights is given by $\Vert \mathbf{W} - \mathbf{W} _ {truncated} \Vert_F=\sum_{j=m-r}^{m}{\Delta\sigma_j^2}$.
>
> In PEFT research, choosing a rank ($r \ll m$) ensures that the approximate matrix ($\Vert \mathbf{W} - \mathbf{W}_{truncated} \Vert_F$) closely resembles the original matrix ($\mathbf{W}$).
>
> However, as the rank ($r$) increases, there is a potential risk that some truncated components may still contain task-relevant information. To investigate this, we examine the impact of varying the SVD rank (denoted as $k \in \\{1, 2, 4, 8, 16, 32, 64, 128\\}$) on model performance. This analysis uses RoBERTa-base across the MRPC, CoLA, and RTE tasks, as illustrated in Figure 8 of Appendix D.1 in our manuscript. As shown in Figure 8, model performance improves as $k$ increases from 1 to 8. This demonstrates that truncating the long-tail noise or less relevant information, associated with much smaller singular values, can positively enhance task-relevant knowledge adaptation. Conversely, increasing $k$ from 8 to 128 results in a performance decline, indicating that some truncated components may still contain valuable task-relevant information.
>
> These findings emphasize the importance of identifying an optimal SVD truncation rank that balances incorporating world knowledge with large singular values and excluding disruptive noise with smaller singular values. Achieving this balance is crucial for optimizing model performance.
>
> > The SVD rank and regularization coefficients play an important role in KaSA’s performance. Could you provide practical guidelines or a sensitivity analysis to help practitioners choose these values for new tasks or datasets? This could make the method more accessible to those less familiar with PEFT or SVD.
> >
>
> According to our empirical results, we recommend that when there is a significant disparity between the distribution of the pre-training tasks and the downstream fine-tuning task, opting for a larger SVD rank can effectively filter out knowledge that is not pertinent to the downstream task. Conversely, a smaller SVD rank should be selected when the distributions are more aligned. A comprehensive sensitivity analysis of the SVD rank is provided in Figure 8 of Appendix D.1 in our manuscript.
>
> Additionally, for more complex fine-tuning tasks, we advise using larger regularization coefficients to enhance the stability of model training and address potential issues such as gradient vanishing or explosion. The sensitivity analysis of these regularization coefficients for RoBERTa-base on CoLA, RoBERTa-large on SST-2, and DeBERTa-v3-base on MRPC is presented in the Table below. The findings indicate that KaSA exhibits robustness to variations in the regularization coefficients $\beta$ and $\gamma$.
>
> | Hyperparameters | RoBERTa-base CoLA | RoBERTa-large SST-2 | DeBERTa-v3-base MRPC |
> | --- | --- | --- | --- |
> | $\beta=0.01,\gamma=1.0$ | 0.6581 | 0.9587 | 0.9044 |
> | $\beta=0.1,\gamma=0.0001$ | 0.6334 | 0.9587 | 0.8971 |
> | $\beta=0.01,\gamma=0.1$ | 0.6414 | 0.9622 | 0.8995 |
> | $\beta=0.0,\gamma=0.0$ | 0.646 | 0.9599 | 0.902 |
> | $\beta=0.001,\gamma=0.01$ | 0.6358 | 0.9587 | 0.9093 |
> | $\beta=0.001,\gamma=0.001$ | 0.6553 | 0.9576 | 0.9093 |
> | $\beta=0.01,\gamma=0.001$ | 0.6506 | 0.5092 | 0.902 |
> | $\beta=0.1,\gamma=0.01$ | 0.6333 | 0.9587 | 0.902 |
> | $\beta=0.0001,\gamma=0.1$ | 0.6485 | 0.9622 | 0.8995 |
> | $\beta=0.01,\gamma=0.0001$ | 0.6347 | 0.9576 | 0.9044 |
> | $\beta=0.0001,\gamma=0.01$ | 0.658 | 0.9599 | 0.9069 |
> | $\beta=1.0,\gamma=0.1$ | 0.6241 | 0.9599 | 0.8971 |
> | $\beta=1.0,\gamma=1.0$ | 0.6291 | 0.9553 | **0.9142** |
> | $\beta=0.1,\gamma=1.0$ | 0.6436 | 0.961 | 0.9093 |
> | $\beta=0.1,\gamma=0.1$ | 0.653 | 0.9587 | 0.9082 |
> | $\beta=1.0,\gamma=0.01$ | 0.6397 | 0.9587 | 0.8995 |
> | $\beta=0.01,\gamma=0.01$ | 0.6433 | 0.9576 | 0.8995 |
> | $\beta=0.0001,\gamma=0.0001$ | 0.6565 | **0.9687** | 0.9044 |
> | $\beta=0.0001,\gamma=0.001$ | **0.6582** | 0.961 | 0.9093 |
> | $\beta=0.1,\gamma=0.001$ | 0.6338 | 0.9599 | 0.902 |
> | $\beta=0.001,\gamma=0.0001$ | 0.6504 | 0.961 | 0.9093 |
> | $\beta=0.001,\gamma=0.1$ | 0.648 | 0.9679 | 0.8971 |
>
> ### Reference
>
> `[1]` Gu, Albert, and Tri Dao. "Mamba: Linear-time sequence modeling with selective state spaces." arXiv preprint arXiv:2312.00752 (2023).

---

> > ### Comment · Reviewer_dg7a · 2024-11-26
> >
> > Thank you for your efforts to address my concerns. I'm still a little bit unsure about the robustness of KaSA. I will maintain my score.

---

### Official Review · Reviewer_9ooo · 2024-11-08

**Soundness:** 4
**Presentation:** 4
**Contribution:** 4
**Rating:** 8
**Confidence:** 5

**Summary:**

This study proposes a new technique for low-rank finetuning and inference. “KaSA” reparameterizes finetuned weights in two steps: task-specific SVD truncation, and “knowledge-aware singular- value adaptation” that selects particular singular values based on their relevance to particular tasks.

This study shows that KaSA is competitive with popular low rank approaches such as LoRA across multiple models and tasks. In particular, they report extensive results for RoBERTa evaluated on the GLUE tasks, GPT2 evaluated on the E2E NLG challenge, and three 7-8B LLMs evaluated on four custom synthetic instruction following datasets.

**Strengths:**

KaSA seems like a promising technique, and is competitive with LoRA and SVD style techniques like PiSSA and MiLoRA. The authors do extensive experiments and benchmarking, and the paper is very well written.

Minor notes:
* The paper has extensive citations that are very helpful to the reader
* Figure 2 with “ablations” is very informative - the trends are quite clear
* The “no robots” dataset is a good choice for seeding synthetic data, since it was generated by human annotators.

**Weaknesses:**

The experimental results are overall quite strong. The main weakness of the manuscript is that it lacks a straightforward explanation of the method. I’m not sure I fully understand the KaSA method here based on the description, and the diagram is slightly confusing as well. It would be very helpful to have a pseudocode block/section.

1. My current understanding is that the method requires (1) LoRA finetuning followed by (2) SVD truncation followed by (3) “knowledge aware singular-value adaptation.” Is this correct? If so, doesn’t this mean that it will require more training/compute than vanilla LoRA? Can you do a direct comparison of training compute/FLOPs for LoRA vs. KaSA?

2. “KaSA employs a small set of parameters Ψ to reparameterize the task-specific update △Θ” is the task specific update full finetuning or LoRA? I assume the task specific update is vanilla LoRA.

3. How do you calculate $W_{fft}$ for $\mathcal{L}_2$? Do you have to do full finetuning? I assume not…

There are a few smaller details that I think are worth mentioning:

1. What is the rank for instruction following? I don’t see it in the manuscript

2. It feels like important baselines are missing from Table 4 - how well do the models do out of the box? Since these datasets are created specifically for this study, it is hard to get a sense of task difficulty or to compare to other papers

3. The authors claim no extra inference latency due to the technique. It would be helpful to show these numbers. If I understand correctly, however, there will be extra _training_ latency. Can you expand on this in the paper?

4. The most similar methods to KaSA are PiSSA and MiLoRA. In Table 4 it could be helpful to do a comparison with PiSSA and/or MiLoRA

**Questions:**

1. What exactly is the reader supposed to take away from the visualizations in Figure 4? It seems somewhat unclear. It might be helpful to have a comparison with and without KaSA

2. Why scale between 0 and 100 when using GPT4o as a judge? It seems a bit arbitrary/unnecessary. Do you have strong motivation for using this scaling?

Some small nits:
* Typo: line 043 “gets popular”
* Typo: line 426 “four synthetic datasets using GPT-4”; is it GPT-4 or GPT4o?

---

> ### Author Response · Authors · 2024-11-24
> **Rebuttal (Part 1/4)**
>
> Thank you for your comprehensive and insightful review of our paper. We appreciate your valuable feedback and suggestions.
>
> > The experimental results are overall quite strong. The main weakness of the manuscript is that it lacks a straightforward explanation of the method. I’m not sure I fully understand the KaSA method here based on the description, and the diagram is slightly confusing as well. It would be very helpful to have a pseudocode block/section.
> >
>
> It is a nice suggestion to incorporate a pseudocode block for a better explanation of our method. In what follows, we provide the PyTorch-style pseudocode of KaSA. Specifically, KaSA consists of three main steps, including (Step 1) Knowledge-Based SVD Truncation, (Step 2) Knowledge-Aware Singular-Value Adaptation, and (Step 3) Integration of Fine-tuned Parameters into the Base Model.
>
> ```python
> class KaSA(nn.Module):
>     def __init__(self,
> 	    rank: int = 8, # kasa rank
> 	    alpha: int = 16, # kasa alpha
> 	    base_layer: nn.Module # pre-trained layer
>     )
>         # definitions
>         self.r = rank
>         self.alpha = alpha
>         self.scaling = alpha / rank
>         self.in_features, self.out_features = base_layer.in_features, base_layer.out_features
>
>         # Step 1: knowledge-based SVD truncation
>         self.svd_rank = self.in_features - self.r
>         U, S, Vh = torch.linalg.svd(base_layer.weight.data, full_matrices=False)
>         base_layer.weight.data = U[:, :self.svd_rank] @ torch.diag(S[:self.svd_rank]) @ Vh[:self.svd_rank, :]
>         self.base_layer = base_layer
>
>         # Step 2: knowledge-aware singular-value adaptation
>         self.delta_v = nn.Linear(self.in_features, self.r, bias=False)
>         self.delta_sigma = torch.diag(nn.Parameter(torch.randn(self.r), requires_grad=True))
>         self.delta_u = nn.Linear(self.r, self.out_features, bias=False)
>
>     def forward(self, x: torch.Tensor):
>     	# Step 3: merge W + Delta_W (Eq.7)
>         Delta_W = self.delta_u @ self.delta_sigma @ self.delta_v
>         result = self.base_layer(x)
>         result = result + torch.einsum('ijk,kl->ijl', x, Delta_W) * self.scaling
>         return result
> ```

---

> ### Author Response · Authors · 2024-11-24
> **Rebuttal (Part 2/4)**
>
> > My current understanding is that the method requires (1) LoRA finetuning followed by (2) SVD truncation followed by (3) “knowledge aware singular-value adaptation.” Is this correct? If so, doesn’t this mean that it will require more training/compute than vanilla LoRA? Can you do a direct comparison of training compute/FLOPs for LoRA vs. KaSA?
> The authors claim no extra inference latency due to the technique. It would be helpful to show these numbers. If I understand correctly, however, there will be extra training latency. Can you expand on this in the paper?
> >
>
> Thank you for your insightful feedback regarding the training compute/FLOPs comparison between KaSA and other baselines. We clarify that the dynamic singular value adaptation introduced in our KaSA is a learnable one-dimensional vector of size $r \ll m$ and requires parameter regularizations, incurring negligible training overheads compared to the standard LoRA. In addition, due to low-rank approximation of the original matrix, we reduce the rank of $\mathbf{W}$ from $m$ to $m-r$, accelerating the inference particularly for small-scale language models like RoBERTa-base 125M (i.e., with small $m$). We show the experimental results as follows. As can be seen, compared to LoRA, the extra training overhead is less than 20% (resp. 3%) for the NLU (resp. NLG) task, while speeding up the inference by 1.45x (resp. 1.02x) times.
>
> **RoBERTa-base 125M on Single NVIDIA GeForce RTX 3090 (24GB) GPU for NLU task**
>
> | Method | LoRA | PiSSA | MiLoRA | KaSA |
> | --- | --- | --- | --- | --- |
> | \# Trainable Parameters | 0.23716% | 0.23716% | 0.23716% | 0.23732% |
> | \# GPU Memory | 1638M | 1638M | 1638M | 1650M |
> | \# Training FLOPs ($\times 10^9$ per sample) | 2.0306 | 1.9270 | 1.9270 | 2.1503 |
> | Training Latency (per epoch) | 9.4868s | 9.8825s | 9.9267s | 11.3679s  |
> | Inference Latency (per batch size 32) | 0.0173s | 0.0108s | 0.0165s | 0.0119s |
> | Matrix Rank | $\text{rank}(\mathbf{W})=m$  $\text{rank}(\mathbf{\Delta W})=r$ | $\text{rank}(\mathbf{W})=m-r$  $\text{rank}(\mathbf{\Delta W})=r$ | $\text{rank}(\mathbf{W})=m-r$  $\text{rank}(\mathbf{\Delta W})=r$ | $\text{rank}(\mathbf{W})=m-r$  $\text{rank}(\mathbf{\Delta W})\leq r$ |
> | CoLA Performance (Mcc.) | 63.4% | 65.5% | 63.1% | **65.8%** |
>
> **LLaMA3 8B on Single NVIDIA A100-SXM4 (80GB) GPU for NLG task**
>
> | Method | LoRA | PiSSA | MiLoRA | KaSA |
> | --- | --- | --- | --- | --- |
> | \# Trainable Parameters | 0.04241% | 0.04241% | 0.04241% | 0.04242% |
> | \# GPU Memory | 71023M | 71023M | 71023M | 71095M |
> | \# Training FLOPs ($\times 10^9$ per sample) | 240.2583 | 240.2583 | 240.2583 | 240.2585 |
> | Training Latency (per epoch) | 2469.6s | 2543.1s | 2476.8s | 2528.9s  |
> | Inference Latency (per batch size 16) | 0.7898s | 0.7687s | 0.7705s | 0.7771s |
> | Matrix Rank | $\text{rank}(\mathbf{W})=m$ $\text{rank}(\mathbf{\Delta W})=r$ | $\text{rank}(\mathbf{W})=m-r$ $\text{rank}(\mathbf{\Delta W})=r$ | $\text{rank}(\mathbf{W})=m-r$ $\text{rank}(\mathbf{\Delta W})=r$ | $\text{rank}(\mathbf{W})=m-r$  $\text{rank}(\mathbf{\Delta W})\leq r$ |
> | MT-Bench Performance (Scores) | 4.1937 | 4.2625 | 4.3187 | **4.7125** |
>
> > "KaSA employs a small set of parameters $\psi$ to reparameterize the task-specific update $\Delta \Theta$" is the task specific update full finetuning or LoRA? I assume the task specific update is vanilla LoRA.
> >
>
> To clarify, the task-specific update $\Delta \Theta$ does not require full fine-tuning. Similar to LoRA, our KaSA method employs low-rank matrices, parameterized by $\psi$, to effectively learn $\Delta \Theta$. This approach allows the model to efficiently learn task-specific updates without the need for extensive parameter updates.
>
> > How do you calculate $W_{fft}$ for $L_2$? Do you have to do full finetuning? I assume not…
> >
>
> We confirm that we do not perform full fine-tuning. In fact, instead of directly calculating $W_{fft}$, we compute L2 regularization loss without involving $W_{fft}$ as a lower bound of $||W_{fft}-W_{world}||_F$.
>
> > What is the rank for instruction following? I don’t see it in the manuscript
> >
>
> We set the ranks $r_q$ and $r_v$ to 8 for instruction following. Due to page constraints, we provide the detailed implementation configurations for instruction-following tasks in Table 10 of the Appendix.

---

> ### Author Response · Authors · 2024-11-24
> **Rebuttal (Part 3/4)**
>
> > It feels like important baselines are missing from Table 4 - how well do the models do out of the box? Since these datasets are created specifically for this study, it is hard to get a sense of task difficulty or to compare to other papers.
> The most similar methods to KaSA are PiSSA and MiLoRA. In Table 4 it could be helpful to do a comparison with PiSSA and/or MiLoRA.
> >
>
> Following your valuable suggestion, we have incorporated important baselines, such as the models' performance out of the box and SVD variant methods, into the updated Table 4 below.
>
> As can be seen, KaSA consistently and significantly outperforms LoRA and the SVD variant baselines across Gemma 7B, Mistral 7B, LLaMA 3 8B, and LLaMA 2 13B models. Notably, the baseline models, when used out of the box, tend to generate random and poor-quality responses for specific downstream tasks. This highlights the base model's lack of specific domain knowledge.
>
> ### Gemma 7B
>
> | Method | # Trainable Parameters | Classification | Summarization | Coding | Closed QA |
> | --- | --- | --- | --- | --- | --- |
> | w/o FT | - | 2.41 | 2.28 | 3.07 | 2.95 |
> | FFT | 8.54B | 5.58 | 7.78 | 7.61 | **8.88** |
> | LoRA | 3.21M | 5.98±0.3 (p=0.001) | 7.29±0.2 (p=0.002) | 7.75±0.2 (p=0.049) | 8.18±0.2 (p=0.002) |
> | PiSSA | 3.21M | 6.23±0.2 (p=0.002) | 7.88±0.1 (p=0.730) | 7.80±0.1 (p=0.018) | 8.22±0.2 (p=0.003) |
> | MiLoRA | 3.21M | 6.30±0.1 (p=0.001) | 7.62±0.2 (p=0.067) | 7.71±0.2 (p=0.028) | 8.27±0.3 (p=0.029) |
> | **KaSA** | 3.22M | **6.88±0.2** | **7.92±0.2** | **8.01±0.1** | 8.69±0.1 |
>
> ### Mistral 7B
>
> | Method | # Trainable Parameters | Classification | Summarization | Coding | Closed QA |
> | --- | --- | --- | --- | --- | --- |
> | w/o FT | - | 2.31 | 2.81 | 2.32 | 3.02 |
> | FFT | 7.25B | **6.73** | **7.18** | **7.53** | **8.75** |
> | LoRA | 3.40M | 5.07±0.3 (p=0.007) | 5.72±0.2 (p=0.000) | 6.17±0.4 (p=0.034) | 7.39±0.2 (p=0.034) |
> | PiSSA | 3.40M | 5.46±0.2 (p=0.103) | 5.86±0.3 (p=0.002) | 6.41±0.2 (p=0.048) | 7.24±0.2 (p=0.007) |
> | MiLoRA | 3.40M | 5.33±0.2 (p=0.025) | 5.89±0.4 (p=0.006) | 6.52±0.2 (p=0.158) | 7.28±0.3 (p=0.031) |
> | **KaSA** | 3.41M | 5.72±0.2 | 6.82±0.3 | 6.74±0.2 | 7.75±0.2 |
>
> ### LLaMA3 8B
>
> | Method | # Trainable Parameters | Classification | Summarization | Coding | Closed QA |
> | --- | --- | --- | --- | --- | --- |
> | w/o FT | - | 2.04 | 2.03 | 2.86 | 3.33 |
> | FFT | 8.03B | 5.44 | 7.80 | 7.59 | **8.90** |
> | LoRA | 3.40M | 6.12±0.3 (p=0.019) | 7.20±0.4 (p=0.016) | 7.37±0.2 (p=0.006) | 6.02±0.2 (p=0.002) |
> | PiSSA | 3.40M | 6.35±0.1 (p=0.028) | 7.31±0.3 (p=0.011) | 7.59±0.1 (p=0.028) | 6.18±0.3 (p=0.018) |
> | MiLoRA | 3.40M | 6.37±0.2 (p=0.083) | 7.61±0.1 (p=0.014) | 7.65±0.2 (p=0.128) | 6.39±0.1 (p=0.029) |
> | **KaSA** | 3.41M | **6.55±0.2** | **7.83±0.1** | **7.89±0.2** | 6.81±0.3 |
>
> ### LLaMA2 13B
>
> | Method | # Trainable Parameters | Classification | Summarization | Coding | Closed QA |
> | --- | --- | --- | --- | --- | --- |
> | w/o FT | - | 1.00 | 1.08 | 1.01 | 1.27 |
> | FFT | 13.02B | 5.86 | **7.93** | 7.88 | **8.97** |
> | LoRA | 6.55M | 6.23±0.4 (p=0.023) | 7.38±0.2 (p=0.005) | 7.54±0.2 (p=0.005) | 6.25±0.3 (p=0.001) |
> | PiSSA | 6.55M | 6.47±0.3 (p=0.062) | 7.45±0.3 (p=0.031) | 7.83±0.1 (p=0.049) | 6.54±0.3 (p=0.006) |
> | MiLoRA | 6.55M | 6.45±0.2 (p=0.020) | 7.63±0.1 (p=0.032) | 7.85±0.1 (p=0.064) | 6.82±0.2 (p=0.028) |
> | **KaSA** | 6.56M | **6.86±0.2** | 7.92±0.2 | **8.09±0.2** | 7.12±0.1 |

---

> ### Author Response · Authors · 2024-11-24
> **Rebuttal (Part 4/4)**
>
> > What exactly is the reader supposed to take away from the visualizations in Figure 4? It seems somewhat unclear. It might be helpful to have a comparison with and without KaSA
> >
>
> Figure 4 shows that different scales of singular values are allocated across positions in various layers, suggesting a dynamic prioritization of knowledge across parameters. This confirms the effectiveness of the innovative knowledge-aware singular-value adaptation in KaSA. Note that the knowledge-aware singular-value adaptation is a core component of KaSA, which cannot be trivially applied to other models.
>
> > Why scale between 0 and 100 when using GPT4o as a judge? It seems a bit arbitrary/unnecessary. Do you have strong motivation for using this scaling?
> >
>
> We appreciate the reviewer's insightful feedback on this matter. Following the common setting in previous work `[1]`, `[2]`, and `[3]`, we have conducted additional experiments using a 0 to 10 score range. The updated results are presented in the Tables in Part 3. The general trends are similar, demonstrating the superiority of our KaSA method.
>
> > Some small nits:
> > * Typo: line 043 “gets popular”
> > * Typo: line 426 “four synthetic datasets using GPT-4”; is it GPT-4 or GPT4o?
> >
>
> Thank you for bringing these typos to our attention. We have addressed and corrected them in the revised version of our manuscript.
>
> ### Reference
>
> `[1]` Zheng, Lianmin, Wei-Lin Chiang, Ying Sheng, Siyuan Zhuang, Zhanghao Wu, Yonghao Zhuang, Zi Lin et al. "Judging llm-as-a-judge with mt-bench and chatbot arena." NeurIPS, 2023.
>
> `[2]` Kopiczko, Dawid Jan, Tijmen Blankevoort, and Yuki Markus Asano. "Vera: Vector-based random matrix adaptation." ICLR, 2024.
>
> `[3]` Gao, Ziqi, Qichao Wang, Aochuan Chen, Zijing Liu, Bingzhe Wu, Liang Chen, and Jia Li. "Parameter-Efficient Fine-Tuning with Discrete Fourier Transform." ICML, 2024.

---

> > ### Comment · Reviewer_9ooo · 2024-11-25
> > **Thank you for the rigorous updates**
> >
> > I would like to thank the authors for their rigorous rebuttal/updates. They have addressed all my concerns with lots of detail and new experiments. I strongly recommend that this paper gets accepted, and I hope that these extra details will be included in the appendix of the final version!

---

> > > ### Author Response · Authors · 2024-11-25
> > > **Thank you for the prompt response and recognition**
> > >
> > > Dear Reviewer,
> > >
> > > Thank you for your prompt response and recognition of our work. We have included these extra details in the Appendix of our revised manuscript.
> > >
> > > Thanks,
> > >
> > > Best regards

---

### Official Review · Reviewer_pwGj · 2024-11-10

**Soundness:** 3
**Presentation:** 3
**Contribution:** 2
**Rating:** 6
**Confidence:** 3

**Summary:**

This paper presents a novel parameter-efficient learning algorithm for LLMs. It addresses two issues of the prior research: i) a subset of model updates are associated with the noise encoded in the base model; and ii) the low-rank matrices A and B are initialized using the knowledge from the base model that is not relevant to the downstream tasks. The new method tackles the two issues with four components: knowledgebased SVD truncation, knowledge-aware singular value adaptation, singular value regularization L2, and orthogonal regularization. The ablation studies show that all four of them are effective for the selected evaluation tasks.

**Strengths:**

* The evaluated tasks are comprehensive, including NLU, NLG and instruction-following.
* The ablation study provides evidence to demonstrate the effectiveness of all four design choices on selected NLP tasks.

**Weaknesses:**

* The improvement of the proposed method over the baselines is marginal. The variances and the results of the significance tests may help justify the improvement.
* It is unclear why GPT-2 is used for NLG tasks, while Gemma, Mistral and LlaMA3 are used for instruction following tasks. Hence, the experimental design is not systematic.

**Questions:**

It would be great to justify why only MRPC, CoLA and RTE are chosen for the ablation studies.

---

> ### Author Response · Authors · 2024-11-24
> **Rebuttal (Part 1/2)**
>
> Thank you for your comprehensive and insightful review of our paper. We appreciate your valuable feedback and suggestions.
>
> > The improvement of the proposed method over the baselines is marginal. The variances and the results of the significance tests may help justify the improvement.
> >
>
> Thank you for your valuable suggestion. We agree that it is a common phenomenon that the average performance improvement of recent SOTA PEFT methods (e.g., PiSSA `[1]` and FourierFT `[2]` as illustrated in Table 2) over existing approaches is relatively marginal. However, as per your suggestion, we show that our KaSA method clearly outperforms baselines in terms of significance tests.
>
> Specifically, we present the results as mean $\pm$ standard deviation and utilize an independent two-sample t-test to evaluate the statistical significance of the differences between our method and the baseline. This evaluation is conducted over five runs with different random seeds to ensure robustness. All tests are carried out at a significance level of $\alpha = 0.05$. The performance comparison between our KaSA and key baselines on NLU tasks is illustrated in the following Table.
>
> | Method | # Trainable Param. | SST-2 | MRPC | CoLA | QNLI | RTE | STS-B | Avg. |
> | --- | --- | --- | --- | --- | --- | --- | --- | --- |
> | RoBERTa-base (125M) |  |  |  |  |  |  |  |  |
> | LoRA | 0.3M | 95.1$\pm$0.2(p=0.397) | 89.7$\pm$0.8(p=0.030) | 63.4$\pm$1.3(p=0.004) | **93.3$\pm$0.3(p=1.000)** | 78.4$\pm$0.9(p=0.000) | **91.5$\pm$0.2(p=0.057)** | 85.2 |
> | PiSSA | 0.3M | 95.0$\pm$0.1(p=0.022) | 88.2$\pm$0.7(p=0.000) | 65.5$\pm$0.3(p=0.135) | 92.0$\pm$0.4(p=0.000) | 75.1$\pm$0.8(p=0.000) | 90.4$\pm$0.1(p=0.002) | 84.4 |
> | MiLoRA | 0.3M | 94.6$\pm$0.2(p=0.001) | 88.7$\pm$0.6(p=0.000) | 63.1$\pm$0.4(p=0.000) | 92.8$\pm$0.3(p=0.024) | 80.5$\pm$0.9(p=0.048) | 91.3$\pm$0.1(p=0.242) | 85.2 |
> | KaSA | 0.3M | **95.2$\pm$0.1** | **90.7$\pm$0.3** | **65.8$\pm$0.2** | **93.3$\pm$0.2** | **81.6$\pm$0.6** | 91.1$\pm$0.3 | **86.3** |
> | DeBERTaV3-base (184M) |  |  |  |  |  |  |  |  |
> | LoRA | 0.3M | 95.0$\pm$0.4(p=0.000) | 89.7$\pm$1.2(p=0.031) | 68.7$\pm$0.3(p=0.000) | 94.0$\pm$0.2(p=0.020) | 85.6$\pm$0.6(p=0.000) | **91.7$\pm$0.3(p=0.714)** | 87.44 |
> | PiSSA | 0.3M | 95.3$\pm$0.2(p=0.000) | 91.4$\pm$0.6(p=1.000) | 70.3$\pm$0.1(p=0.128) | 93.6$\pm$0.3(p=0.002) | 84.8$\pm$0.2(p=0.000) | 91.4$\pm$0.1(p=0.008) | 87.80 |
> | MiLoRA | 0.3M | 96.0$\pm$0.1(p=0.012) | 89.7$\pm$0.9(p=0.012) | 70.3$\pm$0.2(p=0.549) | 94.1$\pm$0.1(p=0.032) | 85.9$\pm$0.4(p=0.000) | 90.3$\pm$0.2(p=0.000) | 87.73 |
> | KaSA | 0.3M | **96.2$\pm$0.1** | **91.4$\pm$0.8** | **70.4$\pm$0.1** | **94.5$\pm$0.3** | **88.1$\pm$0.4** | 91.6$\pm$0.1 | **88.72** |
>
> As evidenced, our proposed KaSA significantly outperforms all baselines, achieving p-values < 0.05 in 13 out of 18 experimental settings with RoBERTa-base (125M) and 14 out of 18 settings with DeBERTaV3-base (184M). Therefore, the significance tests substantiate the significant improvement of our method over the baselines, verifying the superiority and robustness of our approach.

---

> > ### Author Response · Authors · 2024-11-24
> > **Rebuttal (Part 2/2)**
> >
> > > It is unclear why GPT-2 is used for NLG tasks, while Gemma, Mistral and LlaMA3 are used for instruction following tasks. Hence, the experimental design is not systematic.
> > >
> >
> > We appreciate your suggestion of underscoring the selection of models for tasks with varying levels of difficulty. To ensure easy and fair comparisons with existing PEFT methods, we aligned our experimental setups with those used in previous studies.
> >
> > For instance, both VeRA `[3]` and FourierFT `[2]` utilize GPT-2 for the E2E benchmark (refer to Table 3 in both studies) and LLaMA and LLaMA 2 for instruction-following tasks (see Table 4). Similarly, SARA `[4]` employs GPT-2 for the E2E benchmark, while using LLaMA 7B and 13B for commonsense inference (refer to Table 2) and mathematical reasoning (see Table 4).
> >
> > Unlike the E2E NLG Challenge task, instruction-following tasks present greater challenges due to the complexity and variety of task instructions, which are often framed in diverse natural language forms. Many PEFT works adopt large-scale language models, such as Gemma 7B, Mistral 7B, and LlaMA 7B, for instruction-following tasks. However, for the easier E2E benchmark, GPT-2 is often evaluated despite the lack of a systematic experimental design.
> >
> > > It would be great to justify why only MRPC, CoLA and RTE are chosen for the ablation studies.
> > >
> >
> > We appreciate the reviewer's feedback. Due to page constraints, we have included ablation results only for the MRPC, CoLA, and RTE datasets, as their smaller sizes make it easier to validate our findings.
> > In fact, our experiments reveal consistent results across other datasets as well. For instance, the following table shows the ablation studies conducted on the SST-2, QNLI, and STS-B datasets.
> >
> > | RoBERTa-base | Base | + SVD | + Adaptive Singular-Value | + Singular-Value Regularization | + Orthogonal Regularization |
> > | --- | --- | --- | --- | --- | --- |
> > | SST-2 | 94.38 | 94.84 | 94.95 | 95.07 | **95.18** |
> > | QNLI | 92.17 | 92.35 | 92.71 | 92.82 | **93.31** |
> > | STS-B | 90.19 | 90.52 | 90.77 | 90.92 | **91.14** |
> >
> > We observe the same trends: (i) the model's performance consistently improves with the inclusion of additional components during fine-tuning; (ii) excluding any of these components leads to a decline in performance, emphasizing the effectiveness of the designed principal components in KaSA.
> >
> > ### Reference
> >
> > `[1]` Meng, Fanxu, Zhaohui Wang, and Muhan Zhang. "Pissa: Principal singular values and singular vectors adaptation of large language models." NeurIPS, 2024.
> >
> > `[2]` Gao, Ziqi, Qichao Wang, Aochuan Chen, Zijing Liu, Bingzhe Wu, Liang Chen, and Jia Li. "Parameter-Efficient Fine-Tuning with Discrete Fourier Transform." ICML, 2024.
> >
> > `[3]` Kopiczko, Dawid Jan, Tijmen Blankevoort, and Yuki Markus Asano. "Vera: Vector-based random matrix adaptation." ICLR, 2024.
> >
> > `[4]` Gu, Jihao, Shuai Chen, Zelin Wang, Yibo Zhang, and Ping Gong. "Sara: Singular-value based adaptive low-rank adaption." arXiv preprint arXiv:2408.03290 (2024).

---

### Meta-Review · Area_Chair_pkiP · 2024-12-20

**Metareview:**

This paper presents an approach to parameter-efficient finetuning based on SVD, where SVD is applied on the original model to get rid of the "noisy" components, and the "knowledge-aware" components are adapted during finetuning. The approach is found to outperform other PEFT methods, including approaches such as PiSSA which also use SVD to enable better LoRA-style finetuning.

The main strength of the paper is its comprehensive experiments across a bunch of tasks and models (both encoder/decoder models). However, while the method seems to result in empirical gains (that seem small but statistically significant), I found the paper quite difficult to read, despite being quite familiar with PEFT methods. I was also not a huge fan of the rather laissez faire use of the term "knowledge" in the paper.

Despite these drawbacks, the reviewers were generally positive of the paper, especially with the new results with LLaMA that were conducted with the rebuttal. I am thus recommending that this paper be accepted. (However, please consider rewriting section 3 so that it is much clearer for the camera-ready).

**Additional Comments On Reviewer Discussion:**

Many reviewers were initially on the fence about the paper due the the limited scope of initial experiments, but became positive after the authors conducted an impressive set of experiments in the rebuttal, which was a significant factor in my final recommendation.

---

### Decision · Program_Chairs · 2025-01-22

Accept (Poster)